# Remodeling of maternal mRNA through poly(A) tail orchestrates human oocyte-to-embryo transition

Yusheng Liu [1,2,11], Han Zhao[3,11], Fanghong Shao[4,5,11], Yiwei Zhang [2,6,11], Hu Nie[2,7], Jingye Zhang[3], Cheng Li[3], Zhenzhen Hou[3], Zi-Jiang Chen [3,8], Jiaqiang Wang [6] ✉, Bing Zhou[4,9,10] ✉, Keliang Wu [3] ✉ & Falong Lu[2,7] ✉

Poly(A)-tail-mediated post-transcriptional regulation of maternal mRNAs is vital in the oocyte-to-embryo transition (OET). Nothing is known about poly(A) tail dynamics during the human OET. Here, we show that poly(A) tail length and internal non-A residues are highly dynamic during the human OET, using poly(A)-inclusive RNA isoform sequencing (PAIso-seq). Unexpectedly, maternal mRNAs undergo global remodeling: after deadenylation or partial degradation into 3′-UTRs, they are re-polyadenylated to produce polyadenylated degradation intermediates, coinciding with massive incorporation of non-A residues, particularly internal long consecutive U residues, into the newly synthesized poly(A) tails. Moreover, TUT4 and TUT7 contribute to the incorporation of these U residues, BTG4-mediated deadenylation produces substrates for maternal mRNA re-polyadenylation, and TENT4A and TENT4B incorporate internal G residues. The maternal mRNA remodeling is further confirmed using PAIso-seq2. Importantly, maternal mRNA remodeling is essential for the first cleavage of human embryos. Together, these findings broaden our understanding of the post-transcriptional regulation of maternal mRNAs during the human OET.

The OET is the process by which a fully grown oocyte undergoes maturation and fertilization, resulting in a totipotent embryo that can support the full development of a new organism[1–4]. The OET features the absence of transcription until zygotic genome activation (ZGA)[5–7], during which diverse events are controlled by post-transcriptional regulation of maternal mRNAs[1,8–13]. Poly(A) tails are added to the 3′-ends of most eukaryotic mRNAs, where they are essential for mRNA stability and translation[14,15]. Poly(A)-tail-mediated post-transcriptional regulation has essential roles in the OET in several species[1,8–14,16]. In *Drosophila* oocytes, global poly(A) tail elongation, catalyzed by Wispy during late

[1]College of Life Science, Northeast Forestry University, Harbin, China. [2]State Key Laboratory of Molecular Developmental Biology, Institute of Genetics and Developmental Biology, Innovative Academy of Seed Design, Chinese Academy of Sciences, Beijing, China. [3]Center for Reproductive Medicine, Shandong University, The Key laboratory of Reproductive Endocrinology, Ministry of Education, Shandong University, Jinan, China. [4]State Key Laboratory of Stem Cell and Reproductive Biology, Institute of Zoology, Chinese Academy of Sciences, Beijing, China. [5]Division of Life Sciences and Medicine, University of Science and Technology of China, Hefei, China. [6]College of Life Science, Northeast Agricultural University, Harbin, China. [7]University of Chinese Academy of Sciences, Beijing, China. [8]Shanghai Key Laboratory for Assisted Reproduction and Reproductive Genetics, Shanghai, China. [9]Institute for Stem Cell and Regeneration, Chinese Academy of Sciences, Beijing, China. [10]Beijing Institute for Stem Cell and Regenerative Medicine, Beijing, China. [11]These authors contributed equally: Yusheng Liu, Han Zhao, Fanghong Shao, Yiwei Zhang. ✉e-mail: wangjiaqiang@neau.edu.cn; zhoubing@ioz.ac.cn; wukeliang_527@163.com; fllu@genetics.ac.cn

oogenesis, promotes global translation[10,12,17,18]. Zebrafish (*Danio rerio*) miR-430 promotes maternal mRNA clearance by facilitating deadenylation[19]. Uridylation by TUT4 and TUT7 is required for maternal mRNA clearance in early embryos to secure embryonic development in both zebrafish and *Xenopus*[13]. In addition, TUT4 and TUT7 are required for mouse oogenesis[9]. In mouse oocytes, deletion of the maternal *Btg4* or *Cnot6l*, which encode an adapter protein and a core component of the deadenylase complex, respectively, leads to developmental arrest of early embryos due to failed deadenylation of maternal mRNAs[20–24]. Interestingly, one-cell embryos from women with *BTG4* mutations also showed failed first zygotic cleavage[25].

Methods for analyzing transcriptome-wide poly(A) tails enable a global view of poly(A)-tail-mediated post-transcriptional regulation. For example, TAIL-seq (or its modified version, mTAIL-seq) and poly(A)-tail-length profiling by sequencing (PAL-seq and PAL-seq2) are two main technologies, based on the Illumina platform, that can measure poly(A) tail length[10,11,26,27]. In addition, TAIL-seq reveals the existence of non-A residues at the 3′-ends of poly(A) tails[26]. Nanopore sequencing can measure the poly(A) tail length through machine learning of the signal from poly(A) sequence[28–30]. Full-length poly(A) and mRNA sequencing (FLAM-seq) can quantify the length of poly(A) tails and measure the non-A residues in the body of poly(A) tails using PacBio HiFi sequencing[31]. Full-length elongating and polyadenylated RNA sequencing (FLEP-seq) can measure poly(A) tail length on both the PacBio and Nanopore platforms[32,33].

Taking advantage of these methods, profiles of the transcriptome-wide mRNA poly(A) tails during the OET have been revealed in zebrafish, *Xenopus*, and *Drosophila*[10–13]. However, the transcriptome-wide poly(A) tail landscape during mammalian OET remains unknown, because the methods mentioned above require micrograms of input RNA, which cannot be obtained from oocytes or embryos from mammals[10,11,26–34]. The poly(A)-tail-length changes during the mouse OET are known for only a handful of genes[20–24,35–38], whereas they are completely unknown for even a single gene during the human OET.

Recently, we developed PAIso-seq on the PacBio platform to accurately measure the poly(A) tail length and non-A residues within the body of poly(A) tails at single-mouse-oocyte-level sensitivity[39,40]. Here, using single-oocyte/embryo PAIso-seq, we investigated poly(A) tail profiles in human oocytes and early embryos. To study the potential regulatory mechanism of poly(A) tails during the human OET, we performed short interfering RNA (siRNA)-mediated knockdown of *BTG4*, *TUT4*, *TUT7*, *TENT4A*, and *TENT4B*, followed by PAIso-seq analysis. PAIso-seq has limitations: it cannot detect mRNA with very short poly(A) tails, or mRNA with non-A residues at the 3′-ends[39]. Therefore, we additionally analyzed human oocytes and embryos with PAIso-seq2 (ref. [41]), which can detect transcripts with very short or no poly(A) tails, and transcripts with non-A residues at their 3′-ends, although at lower sensitivity. The PAIso-seq2 dataset well-validates our observations

from the PAIso-seq dataset and provides additional insights into the dynamic changes of 3′-end non-A residues during the human OET. Interestingly, blocking maternal mRNA remodeling leads to failed first cleavage of human embryos. Together, the results of our study reveal extensive remodeling of maternal mRNA poly(A) tails and provide an important resource for further study of the oocyte maturation and preimplantation development in humans.

## Results

### PAIso-seq analysis of human oocytes and early embryos

We applied single-oocyte/embryo PAIso-seq to donated human oocytes at the germinal vesicle, metaphase I, and metaphase II stages, as well as pre-implantation embryos at the one-cell, two-cell, four-cell, eight-cell, morula, and blastocyst stages (Fig. 1a). In total, we analyzed 24 oocytes and 31 pre-implantation embryos (Fig. 1b). In addition, to gain insight into the regulation of poly(A) tails during the human OET, we performed PAIso-seq on 18 human one-cell embryos with siRNA-mediated knockdown of *BTG4* (*siBTG4*), *TUT4* and *TUT7* simultaneously (*siTUT4/7*), or *TENT4A* and *TENT4B* simultaneously (*siTENT4A/B*), which encode candidate regulators of poly(A) tails, or a negative control (*siNC*) (Fig. 1b).

We obtained a total of 16 million poly(A)-tail-inclusive full-length complementary DNA reads mapped to the human genome from the 73 oocytes and embryos. All the PAIso-seq experiments were successful, except for one MI oocyte (MI-7), for which very few reads were recovered, indicating loss of this oocyte during the experiment (Supplementary Table 1). Because the same barcode sequence was shared for the following pairs of oocytes, they were combined as one replicate in the subsequent analyses, except for the uniform manifold approximation and projection (UMAP) analysis of the gene expression level: GV-1 and GV-6, GV-2 and GV-7, GV-3 and GV-8, GV-4 and GV-9, MI-1 and MI-7, and MI-2 and MI-8, resulting in five germinal vesicle replicates and six metaphase I replicates. For the other PAIso-seq samples, each oocyte/embryo was used as one replicate in the analyses. Samples from the same stage clustered together for the 72 successful PAIso-seq datasets (MI-7 excluded) using UMAP analysis[42], with the exception that two metaphase I oocytes clustered close to germinal vesicle oocytes (Fig. 1c).

### Dynamics of mRNA poly(A) tail length during the human oocyte-to-embryo transition

The global transcriptome undergoes drastic changes in poly(A) tail lengths during the human OET (Fig. 1d and Supplementary Table 2). In the germinal vesicle stage, most transcripts have poly(A) tail lengths between 15 and 100 nt, with two peaks at around 18 nt and 36 nt. During oocyte maturation, the relative abundance of transcripts with 25- to 60-nt poly(A) tails decreased in metaphase I stage, and further decreased in metaphase II stage, suggesting global deadenylation of maternal mRNAs. After fertilization, the relative abundance of transcripts with 12- to 60-nt poly(A) tails increased in one-, two-, and

**Fig. 1 | Profiles of mRNA poly(A) tail length in human oocytes and early embryos, measured by PAIso-seq. a**, Microscopy imaging of human oocytes and early embryos. Scale bar, 50 µm. Germinal vesicle, GV; metaphase I, MI; metaphase II, MII; one-cell, 1C; two-cell, 2C; four-cell, 4C; eight-cell, 8C; morula, MO; blastocyst, BL. **b**, Numbers of human oocytes and early embryos analyzed. **c**, UMAP analysis of 72 single oocytes and embryos, based on gene expression. Different oocyte or embryo groups are shown as symbols with the same shape and color. **d**, Distributions of the poly(A) tail lengths of transcripts. Median length (nt): GV, 37; MI, 41; MII, 63; 1C, 33; 2C, 37; 4C, 42; 8C, 61; MO, 57; BL, 74. **e**, Box plots for poly(A) tail length distribution of *BTG4* (reads: GV, 4,326; MI, 1,878; MII, 1,159; 1C, 921; 2C, 608; 4C, 174) and *NLRP5* (reads: GV, 432; MI, 170; MII, 78; 1C, 143; 2C, 188; 4C, 103). **f**, Box plots for geometric means of poly(A) tail length of alternative polyadenylation (APA) isoforms with a proximal polyadenylation site (pPAS) or a distal polyadenylation site (dPAS) in oocytes. Genes (GV, 817; MI,

173; MII, 81) with two PASs have enough coverage (≥20 poly(A)+ reads for both dPAS and pPAS isoforms) to be analyzed here. **g**, Gene expression level change in *siBTG4*-transfected (*n* = 4) compared with *siNC*-transfected (*n* = 4) 1C embryos. The upregulated and downregulated genes are shown in pink and blue (|log$_2$(fold change)| ≥ 0.5, *P* < 0.05). **h**, Distributions of poly(A) tail lengths of transcripts in *siNC* and *siBTG4* 1C embryos. **i**, Transcriptional abundance of *PADI6*, *MOS*, and *ZP2* in *siBTG4* (*n* = 4) and *siNC* (*n* = 4) 1C embryos. Error bars indicate the s.e.m. **j**, Histogram of poly(A) tail lengths of transcripts of *PADI6*, *MOS*, and *ZP2* in *siNC* and *siBTG4* 1C embryos. Data lists for **d** and **f**–**i** are provided in Supplementary Tables 2–4. CPM, counts per million. The *P* values were calculated by one-tailed Student's *t*-test. For all box plots, the '×' indicates the mean value, the horizontal bars show the median value, and the top and bottom of the box represent the value of 25th and 75th percentiles, respectively.

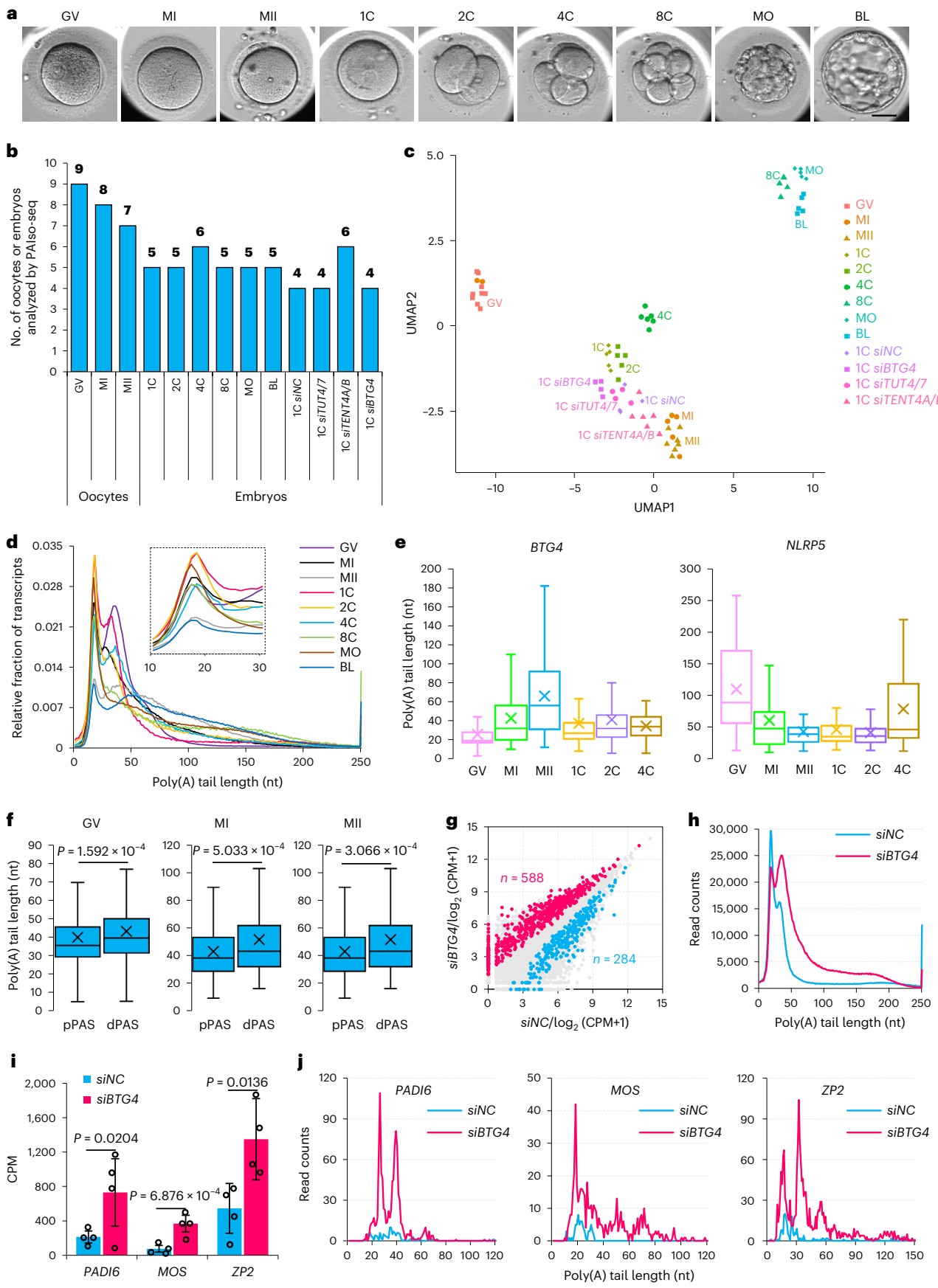

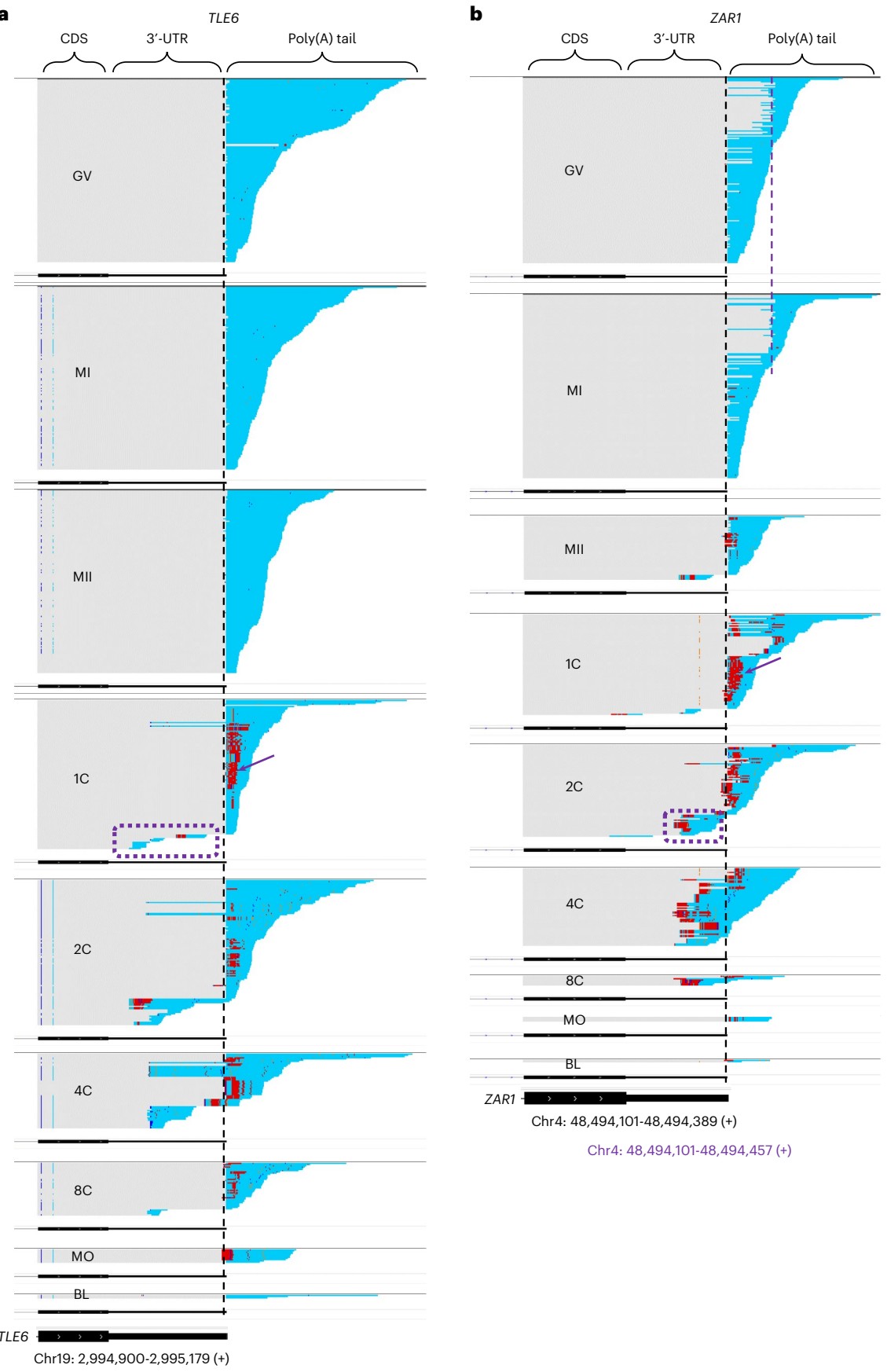

**Fig. 2 | Genome browser view of aligned PAIso-seq reads for two representative genes. a,b**, IGV tracks showing reads (including part of the CDS, 3′-UTR and poly(A) tail) of *TLE6* (**a**) and *ZAR1* (**b**) in oocytes and early embryos. For *TLE6*, the vertical dotted lines in black mark the position of annotated polyadenylation sites. For *ZAR1*, the vertical dotted line in black marks the position of an annotated proximal polyadenylation site, while the vertical dotted line in violet marks the position of an unannotated distal polyadenylation site, and gray lines between the black and violet dotted line represent alternative 3′-UTR sequences included in the long isoform. The sequences that match the reference genome are shown in gray (the dispersed colored regions in the genomic regions indicated unmatched residues caused by single-nucleotide polymorphisms or sequencing errors); the poly(A) tails that cannot map to the genome are shown in colors (A residues, cyan; U residues, magenta; C residues, blue; G residues, dark yellow). Violet dotted rectangle highlights polyadenylated reads showing a 3′-UTR shorter than the canonical 3′-UTR. The violet arrow indicates the presence of a large amount of U residues within the poly(A) tails. There are very few detected *TLE6* reads in MO and BL embryos, and there are very few detected *ZAR1* reads in 8C, MO, and BL embryos. We combined all replicates of each stage for analysis.

four-cell embryos, suggesting global polyadenylation. In eight-cell and morula embryos, in which ZGA has already taken place[43–48], most transcripts have poly(A) tail lengths between 15 and 200 nt, with an ~18-nt peak. In blastocysts, the relative abundance of transcripts with 15- to 40-nt poly(A) tail is decreased.

There is no information about poly(A) tail length for any gene in human oocytes and early embryos. Therefore, we turned to two genes that are known to be of conserved translational regulation in human and mouse oocytes. In both mice and humans, BTG4 protein is absent in germinal vesicle oocytes, and its translation begins after germinal vesicle breakdown[21,23,25], while NLRP5 protein is already present in the germinal vesicle oocytes[49–51]. *BTG4* mRNA harbored a shorter poly(A) tail than did *NLRP5* in human germinal vesicle oocytes (Fig. 1e), and its length increased gradually during human oocyte maturation. In contrast, the poly(A) tail length of *NLRP5* decreased gradually (Fig. 1e). These results suggest that *BTG4*, but not *NLRP5*, is also dormant maternal mRNA in human germinal vesicle oocytes and is similar to that in mice[21,23,49,50]. In addition, the poly(A) tail lengths of *BTG4* and *NLRP5* also showed differential changes in early human embryos (Fig. 1e). These results indicate that poly(A) tail lengths are differentially regulated in human oocytes and early embryos.

PAIso-seq can analyze the mRNA-isoform-specific poly(A) tails[39]. Interestingly, isoforms derived from distal polyadenylation site (dPAS) carried significantly longer poly(A) tails than proximal ones in human germinal vesicle, metaphase I, and metaphase II oocytes (Fig. 1f and Supplementary Table 3), indicating that mRNA isoforms are differentially controlled through poly(A) tail length during human oocyte maturation.

### BTG4 regulates maternal mRNA deadenylation during the human oocyte-to-embryo transition

Among the BTG/Tob family genes, *BTG4* was highly expressed in mouse[21,23] and human oocytes and early embryos (Extended Data Fig. 1a). In *BTG4*-knockdown human one-cell embryos, 588 genes showed significant upregulation, while 284 genes showed significant downregulation (Fig. 1g, Extended Data Fig. 1c,d, and Supplementary Table 4). As the read number for the low-input full-length transcriptome is relatively low, to quantify the level of transcripts of the coding genes measured by PAIso-seq, we compared the counts per million reads mapped (CPM), followed by Student's *t*-test analysis, which showed comparable results to those of the DESeq2 and EdgeR methods (Extended Data Fig. 1g). The global distribution of poly(A) tail lengths shifted right towards the longer side, with more long tails and fewer short tails in human *BTG4*-knockdown one-cell embryos (Fig. 1h and Supplementary Table 2). Decay of *Padi6*, *Mos*, and *Zp2* mRNAs occurs through BTG4-mediated deadenylation in mouse oocytes and one-cell embryos[21,23]. These three genes also showed increased mRNA level with poly(A) tails in the range of 20–40 nt in *BTG4*-knockdown human one-cell embryos (Fig. 1i,j). These results reveal that human BTG4 regulates global maternal mRNA deadenylation.

### Two unique features of mRNA poly(A) tails of maternal mRNA

When examining the poly(A) tails of the maternal mRNAs at different stages, we found two very interesting features, as shown by Integrative Genomics Viewer (IGV) screenshots of *TLE6* and *ZAR1* (Fig. 2), which encode essential regulators of the OET in mice and in humans[50,52–55]. We observed the appearance of many polyadenylated transcripts with shortened 3′-untranslated regions (3′-UTRs) (violet dotted rectangular in Fig. 2) in one-, two-, and four-cell embryos, compared with the amount in the germinal vesicle and metaphase I stages. Many non-A residues appeared (colors other than cyan in the poly(A) tails in Fig. 2), especially in the one-, two-, and four-cell stages. Notably, internal U residues, which refer to U residues within poly(A) tails that are not located at the very 3′-ends, often existed in a consecutive manner, such as UU, UUU and U*n* up to more than 20 U residues (violet arrows in Fig. 2, Extended Data Fig. 2a, and Supplementary Table 5). Similar features were observed for other maternal mRNA species (IGV files available in the 'Data availability' section in the Methods), implying that these features were common for maternal mRNAs during the human OET.

### Polyadenylated degradation intermediates

We called the polyadenylated transcripts with intact 3′-UTRs or shortened 3′-UTRs polyadenylated intact transcripts (PITs) or polyadenylated degradation intermediates (PDIs), respectively (Fig. 3a). In the germinal vesicle, metaphase I, and metaphase II oocytes, around 15% of reads were PDIs. However, the proportion of PDIs increased drastically

**Fig. 3 | There is a high level of PDIs and internal non-A residues in human oocytes and early embryos. a**, Illustration of PITs and PDIs. The dashed trapezoid highlights partially degraded 3′-UTRs, with degraded nucleotides shown as small rectangular in gray. **b**, The PDI level of mRNA transcripts. **c**, Box plots of the PDI level of individual genes (GV, 7,796; MI, 5,133; MII, 3,425; 1C, 4,031; 2C, 3,919; 4C, 3,156; 8C, 1,987; MO, 4,812; BL, 4,951; HeLa S3, 2,020; iPSCs, 259; organoids, 471). **d**, Proportion of the PDIs in the detected reads for *TLE6*, *ZAR1*, *ZAR1L*, *BTG4*, *KHDC3L*, *PADI6*, *NLRP2*, *NLRP7*, *NLRP5*, *TUBB8*, *REC114*, *MOS*, *PATL2*, *WEE2*, and *PANX1*. Stages with coverage of fewer than 20 poly(A)+ reads are left blank in the graph. **e**, Frequency of non-A residues of mRNA transcripts. **f**, Box plots showing frequency of non-A residues of individual genes (GV, 8,298; MI, 5,471; MII, 3,643; 1C, 4,265; 2C, 4,154; 4C, 3,378; 8C, 2,341; MO, 6,131; BL, 6,431; HeLa S3, 2,532; iPSCs, 300; organoids, 594). **g**, Distribution of U length of mRNAs. The U length is the number of the longest consecutive U residues within the poly(A) tail. **h–i**, Frequency of non-A residues of mRNAs grouped by the length of the consecutive non-A residues within poly(A) tails. The number of replicates in **b**, **e**, **h**, and **i**: GV, 5; MI, 6; MII, 7; 1C, 5; 2C, 5; 4C, 6; 8C, 5; MO, 5; BL, 5; HeLa S3, 2; iPSCs, 2; organoids, 2. Data lists for **b–i** are provided in Supplementary Tables 6–10. For gene-level analysis, genes with ≥20 reads (the sum of the number of PDIs and PITs for PDI analysis; the number of poly(A)+ reads for non-A residue analysis) are included. Error bars indicate the s.e.m. The *P* values were calculated by one-tailed Student's *t*-test. For all box plots, the '×' indicates the mean value, the horizontal bars show the median value, and the top and bottom of the box represent the value of 25th and 75th percentiles, respectively.

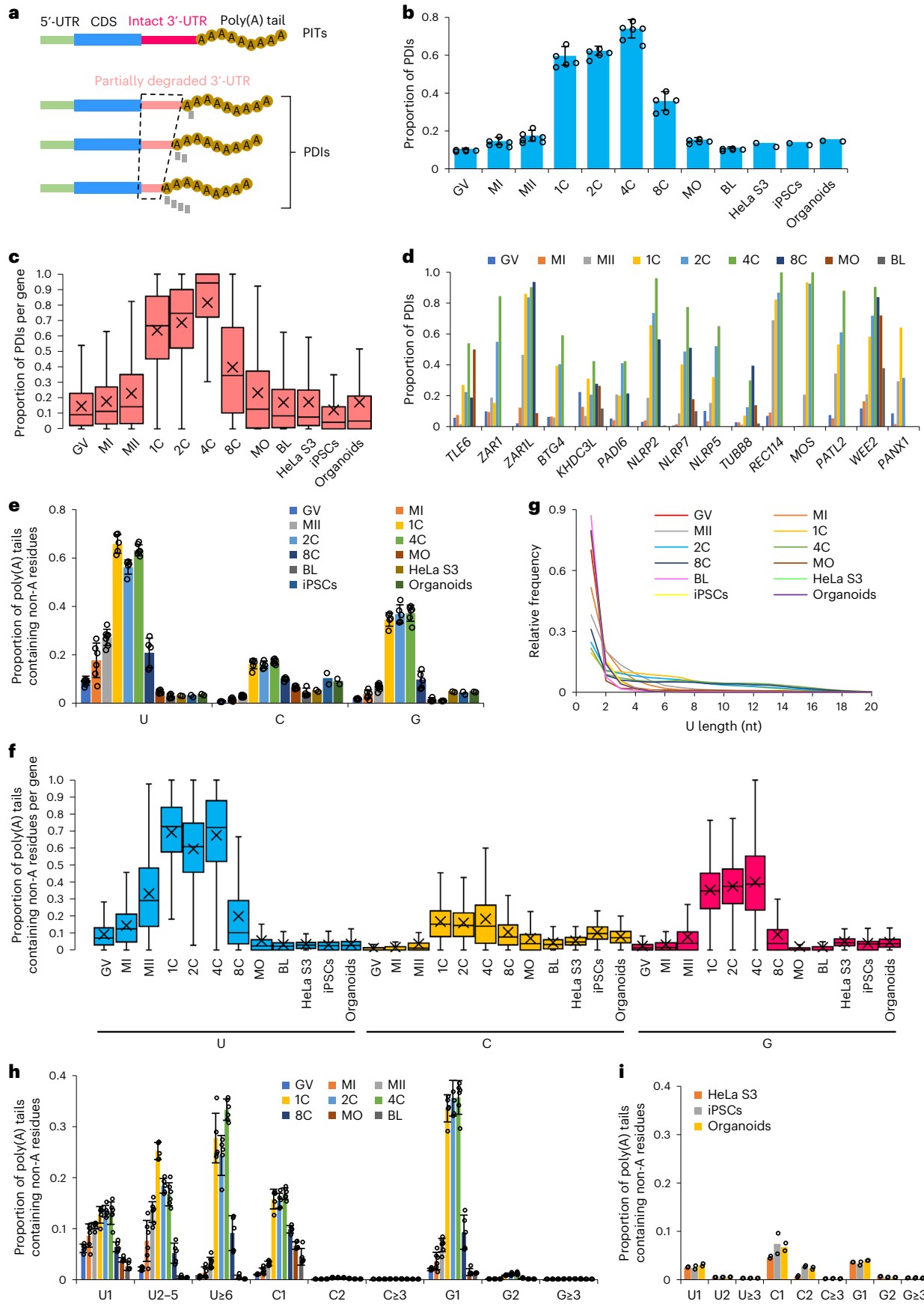

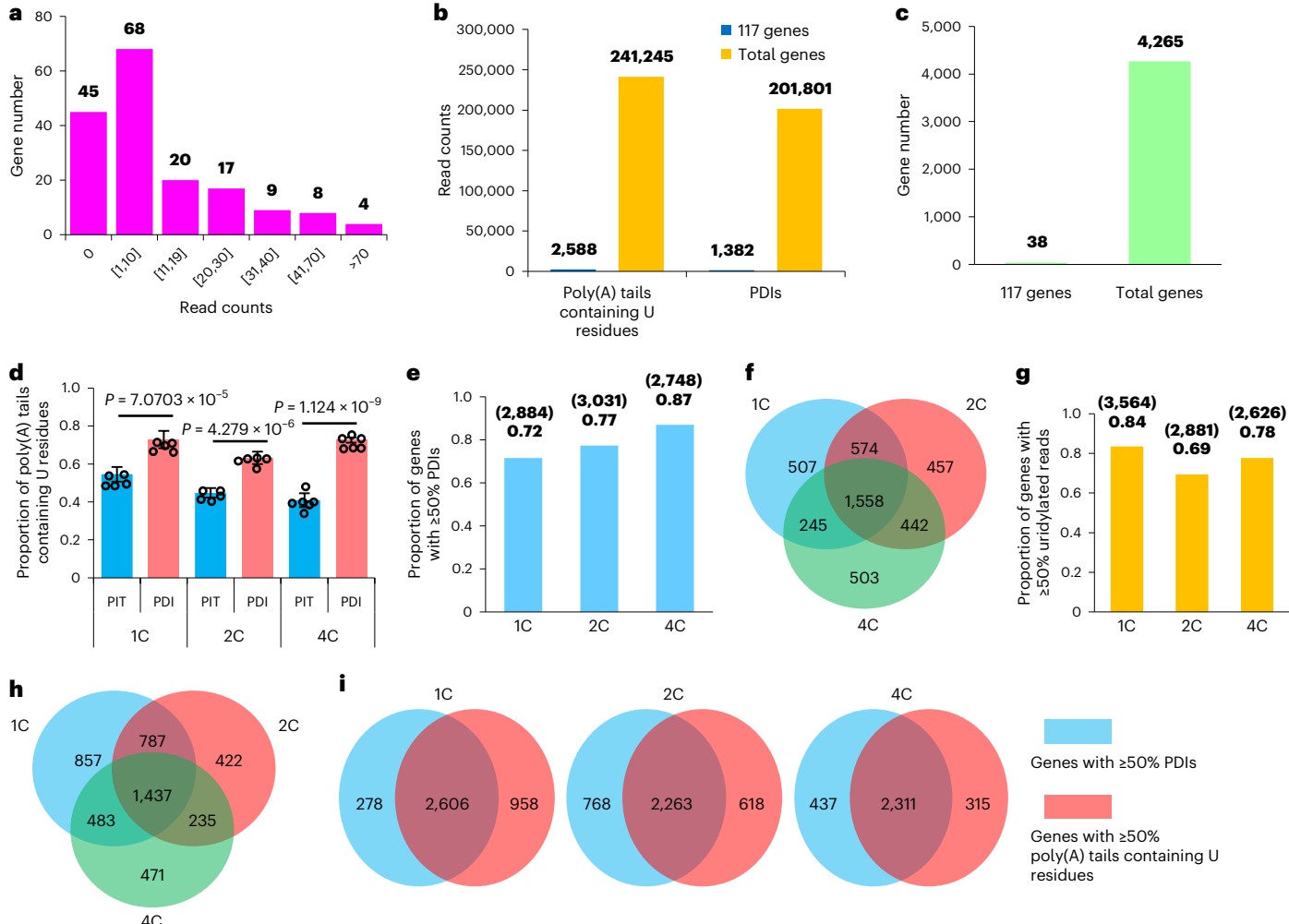

**Fig. 4 | The relationship between PDIs and internal non-A residues in human early embryos, measured by PAIso-seq. a**, The distribution of number of reads detected in 1C embryo PAIso-seq data for the 171 transcriptionally activated genes, as reported in a recent study[67]. The y axis is the number of genes with the given number of detected reads. The numbers in the brackets refer to the minimal (left) and maximal (right) number of read counts for genes included in the groups. **b**, Bar plots of the number of poly(A) tails containing U residues or PDIs for the 171 genes mentioned in **a** or for all the genes in 1C embryo PAIso-seq data. **c**, Bar plots of the number of genes with ≥20 reads among the 171 genes mentioned above or the entire transcriptome in 1C embryo PAIso-seq data. **d**, Frequency of U residues of PITs or PDIs in 1C ($n = 5$), 2C ($n = 5$), and 4C ($n = 6$) embryos. The proportion (y axis) shows the number of PITs or PDIs with U

residues in their poly(A) tails divided by total number of PITs or PDIs for each sample. Error bars indicate the s.e.m. The P values were calculated by one-tailed Student's t test. **e**, Proportion of genes with ≥50% PDIs in 1C, 2C, and 4C embryos. **f**, Venn diagram showing the overlap of genes defined in **e** in 1C, 2C, and 4C embryos. **g**, Proportion of genes with ≥50% uridylated reads in 1C, 2C, and 4C embryos. **h**, Venn diagram showing the overlap of genes defined in **g** in 1C, 2C, and 4C embryos. **i**, Venn diagrams showing the overlap between genes defined in **e** and genes defined in **g** in 1C, 2C, and 4C embryos, respectively. Data lists for **d**, **e**, and **g** are provided in Supplementary Tables 11, 7, and 9. Poly(A)+ reads are included for the analysis. Genes with ≥20 poly(A)+ reads in each stage are included for the analyses in **d**–**i**.

to around 60% in one-cell embryos, which remained high in two- and four-cell embryos, and started to decrease in eight-cell embryos, with the level returning to around 15% in morula and blastocyst (Fig. 3b and Supplementary Table 6). To examine the level of PDIs in non-germ cells, we analyzed HeLa S3, human induced pluripotent stem cell (iPSC), and organoid data generated by FLAM-seq[31]. The proportion of PDIs in these cells was also around 15% (Fig. 3b). Gene-level analysis revealed that the pattern of changes in PDIs across different samples was similar to that seen in the transcript-level analysis (Fig. 3c and Supplementary Table 7), for example for *TLE6*, *ZAR1*, *ZAR1L*, *BTG4*, *KHDC3L*, *PADI6*, *NLRP2*, *NLRP7*, *NLRP5*, *TUBB8*, *REC114*, *MOS*, *PATL2*, *WEE2*, and *PANX1* (refs. [25,53–66]), which encode key regulators of the mouse or human OET (Fig. 3d and Supplementary Table 7). These results reveal that a high level of PDIs is a unique feature for human one-, two-, and four-cell embryos, but not for other stages or somatic cells.

## Internal non-A residues of poly(A) tails during the human oocyte-to-embryo transition

Another feature is the appearance of a large amount of internal non-A residues within poly(A) tails during the human OET. PAIso-seq is not able to capture poly(A) tails with non-A residues at the 3′-end[39]. Owing to the small amount of 3′-end non-A residues that were sequenced (Extended Data Fig. 2b,c and Supplementary Table 8), in the following analysis of non-A residues in the PAIso-seq dataset, we did not separate the internal and 3′-end non-A residues and considered them all as internal non-A residues.

Transcript-level analysis revealed that levels of internal U residues were very high in human one-, two-, and four-cell embryos; these residues were found in around 60% of the mRNA poly(A) tails. The level started to increase along the oocyte maturation, was very high at the one-, two-, and four-cell stages, decreased from the eight-cell stage, and

was lower than that in the germinal vesicle stage in the morula and blastocyst stages (Fig. 3e). The levels of internal C and G residues showed a similar pattern of dynamic changes as internal U residues during the human OET, but the levels of internal C and G residues were lower than that of internal U residues (Fig. 3e). In the FLAM-seq datasets, in which we also could not detect 3′-end non-A residues[31], the transcript-level internal non-A residue abundance was low and was similar to that seen in human germinal vesicle oocytes (Fig. 3e). Gene-level analysis revealed that the pattern of changes of internal non-A residues across different samples was similar to that seen in transcript-level analysis (Fig. 3f and Supplementary Table 9). These results reveal that there is a high level of internal non-A residues in human one-, two-, and four-cell embryos.

We found consecutive internal U residues in the poly(A) tails of maternal mRNAs (Fig. 2 and Supplementary Table 5), with lengths of up to 20 nt; in non-germ cells, the lengths were up to only 5 nt (Fig. 3g and Supplementary Table 10). In contrast, the length of consecutive internal C or G residues was short (Supplementary Table 10). To further quantify the level of consecutive or monomeric internal non-A residues, we separated the poly(A) tails with non-A residues into three groups (U1, U2–5, U≥6 for U; C1, C2, C≥3 for C; and G1, G2, G≥3 for G) on the basis of the maximum length of consecutive non-A residues within a given poly(A) tail. During the human OET, the majority of internal C and G residues were monomeric, and the majority of internal U residues were consecutive (Fig. 3h and Supplementary Table 8). In contrast, the majority of the internal non-A residues in poly(A) tails were monomeric in HeLa S3, iPSCs, and organoids (Fig. 3i and Supplementary Table 8).

## Maternal mRNA remodeling through poly(A) tails during the human oocyte-to-embryo transition

There are two possible mechanisms responsible for global polyadenylation together with the production of PDIs and internal non-A residues in human one-, two-, and four-cell embryos: new transcription or post-transcriptional regulation of maternal mRNAs. Transcription is minor before the eight-cell stage in human embryos[43–48], suggesting that new transcription contributes minimally to the above global changes in poly(A) tails. To further distinguish the contributions from the zygotic transcripts and maternal transcripts, we examined the amount of potentially zygotically transcribed mRNAs. A recent study reported 171 genes ($\log_2$(FC) > 0.5, $P$ < 0.05) with low-level new transcription in human one-cell embryos[67]. There were 366,976 reads in our one-cell embryo poly(A) tail data. However, among the 171 genes, 45 were not detected in our data, and the number of reads for the detected genes was small (Fig. 4a). We detected 201,801 PDIs and 241,245 reads with internal U residues, and only 1,382 PDIs and 2,588 reads with internal U residues belong to the 171 genes (Fig. 4b). We needed at least 20 reads for gene-level analysis; 4,265 genes in our one-cell data and only 38 genes among the 171 genes met this criteria (Fig. 4c). These results reveal that most of the transcripts analyzed

here are maternal mRNAs, and only a few are from potentially newly transcribed mRNAs.

Therefore, maternal mRNAs are globally deadenylated and then re-polyadenylated with a large amount of degradation intermediates being re-polyadenylated to produce PDIs during the human OET, coinciding with incorporation of large numbers of internal non-A residues into the newly synthesized poly(A) tails. We call this process maternal mRNA remodeling.

## Uridylation of polyadenylated degradation intermediates in maternal mRNA remodeling

About 60–70% of PDIs contained internal U residues within their poly(A) tails in human one-, two-, and four-cell embryos (Fig. 4d and Supplementary Table 11), indicating that internal U residues could be incorporated into poly(A) tails during the synthesis of most of the PDIs. We focused on genes in which at least 50% of reads were PDIs and genes in which at least 50% of poly(A) tails contained internal U residues. For the majority of the genes in human one-, two-, and four-cell embryos, at least 50% of transcripts were PDIs or at least 50% of poly(A) tails contained internal U residues (Fig. 4e,g and Supplementary Tables 7 and 9), and they overlapped highly among these three developmental stages (Fig. 4f,h). In addition, genes with at least 50% PDIs and genes in which at least 50% of poly(A) tails contained internal U residues showed good overlap (Fig. 4i). Gene Ontology (GO) analysis showed that these groups of genes were enriched for similar GO terms (Supplementary Table 12), likely because of large overlap of genes among these groups.

## Role of BTG4 in maternal mRNA remodeling

We asked whether active deadenylation affects the production of PDIs and internal non-A residues. BTG4 regulates maternal mRNA deadenylation (Fig. 1h), so we examined the PDI level in the *BTG4*-knockdown one-cell embryos. Both transcript- and gene-level analyses revealed that the level of PDIs decreased significantly after *BTG4* knockdown (Fig. 5a,b and Supplementary Tables 6 and 7), including most of the above mentioned functionally important regulators (Fig. 5c and Supplementary Table 7). Internal U residues in the poly(A) tails usually exist close to the 5′-ends of poly(A) tails (Fig. 2). Hereafter, the length of the sequences between the end of a 3′-UTR and the first U residue is called N length (Fig. 5d). We quantified the global distribution of the N length in human one-, two-, and four-cell embryos. In all U1, U2–5, and U≥6 groups, the N length of the poly(A) tails was very short, with about 30% at 0 and the vast majority having an N length between 1–15 nt (Fig. 5e and Supplementary Table 13), implying that uridylation takes place on the deadenylated maternal mRNAs. Indeed, the N length tended to become longer upon *BTG4* knockdown in human one-cell embryos (Fig. 5f and Supplementary Table 13). Together, our results reveal that BTG4-mediated maternal mRNA deadenylation produces substrates for uridylation and re-polyadenylation to generate internal non-A residues and PDIs during maternal mRNA remodeling.

**Fig. 5 | The rle of BTG4 in maternal mRNA remodeling in one-cell embryos. a,** The proportion of the PDIs in *siNC* ($n$ = 4) and *siBTG4* ($n$ = 4) 1C embryos. **b**, Box plot of the proportion of the PDIs of individual genes (4,643 genes with ≥20 reads in both samples) in *siNC* and *siBTG4* 1C embryos. In box plots, the '×' indicates the mean value, the horizontal bars show the median value, and the top and bottom of the box represent the 25th and 75th percentiles, respectively. **c**, Proportion of the PDIs for *TLE6*, *ZAR1*, *ZAR1L*, *TUBB8*, *PADI6*, *NLRP2*, *NLRP7*, *NLRP5*, *KHDC3L*, *REC114*, *MOS*, *PATL2*, *WEE2*, and *PANX1* in *siNC* and *siBTG4* 1C embryos. **d**, Diagram depicting mRNAs with U residues, grouped by the length of longest stretch of consecutive U residues. The N length represents the number of residues between the end of the 3′-UTR and the first base of the longest consecutive stretch of U residues in a poly(A) tail. **e**, Distribution of N length of mRNAs divided into U1, U2–5, and U≥6 groups in 1C, 2C, and 4C embryos. **f**, Distribution of N length of mRNAs divided into U1, U2–5, and U≥6 groups in *siNC* ($n$ = 4) and *siBTG4* ($n$ = 4)

1C embryos. The $P$ values of N0 to N25 are 0.042, 0.017, 0.324, 0.179, 0.338, 0.375, 0.017, 0.178, 0.012, 0.023, 0.348, 0.214, 0.016, 0.011, $2.744 \times 10^{-3}$, $1.355 \times 10^{-3}$, $1.886 \times 10^{-3}$, $1.779 \times 10^{-4}$, $6.477 \times 10^{-4}$, $2.710 \times 10^{-3}$, $2.103 \times 10^{-3}$, $9.146 \times 10^{-3}$, $2.885 \times 10^{-3}$, $3.757 \times 10^{-3}$, $2.151 \times 10^{-3}$, $9.896 \times 10^{-3}$ for the U1 group; $9.172 \times 10^{-3}$, $7.515 \times 10^{-3}$, 0.260, $2.945 \times 10^{-3}$, 0.395, 0.311, $7.969 \times 10^{-3}$, 0.026, 0.095, $2.454 \times 10^{-3}$, 0.033, 0.028, $5.442 \times 10^{-3}$, $2.917 \times 10^{-3}$, 0.051, $2.314 \times 10^{-3}$, 0.021, $4.852 \times 10^{-3}$, $2.962 \times 10^{-3}$, $8.939 \times 10^{-4}$, 0.031, 0.044, 0.065, $9.049 \times 10^{-3}$, 0.038, $4.591 \times 10^{-4}$ for U2–5 group; and 0.056, 0.044, 0.231, 0.061, 0.098, 0.407, 0.059, 0.129, 0.160, 0.047, 0.161, $6.804 \times 10^{-3}$, 0.010, 0.489, 0.111, 0.098, 0.386, $2.021 \times 10^{-3}$, $4.605 \times 10^{-3}$, 0.046, 0.263, 0.179, 0.207, 0.131, $9.133 \times 10^{-3}$, 0.175 for the U6 group. Data lists for **a**–**c** and **e** and **f** are provided in Supplementary Tables 6, 7, and 13. The data here were measured by PAIso-seq. Error bars indicate the s.e.m. The $P$ values were calculated by one-tailed Student's $t$-test. Genes with ≥20 reads are included for gene-level analysis.

### Role of TUT4 and TUT7 in maternal mRNA remodeling

In HeLa cells and during mouse oogenesis, TUT4 and TUT7 are responsible for the mRNA 3′-end oligo-uridylation, but contribute minimally to the mono-uridylation[9,68]. To test whether TUT4 and TUT7 contribute to poly(A) tail internal oligo-uridylation in human early embryos, we performed knockdown of *TUT4* and *TUT7* (Extended Data Fig. 1c).

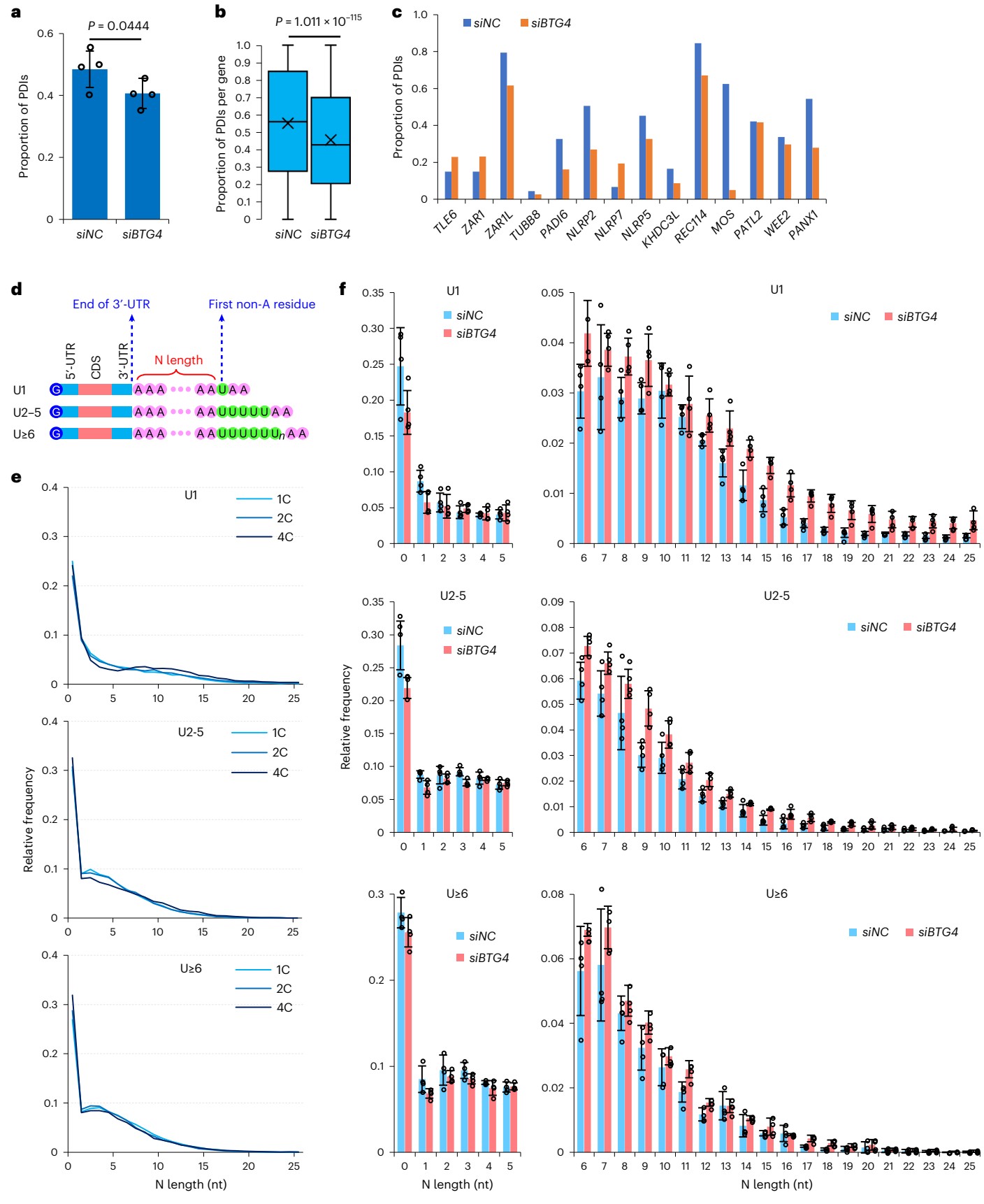

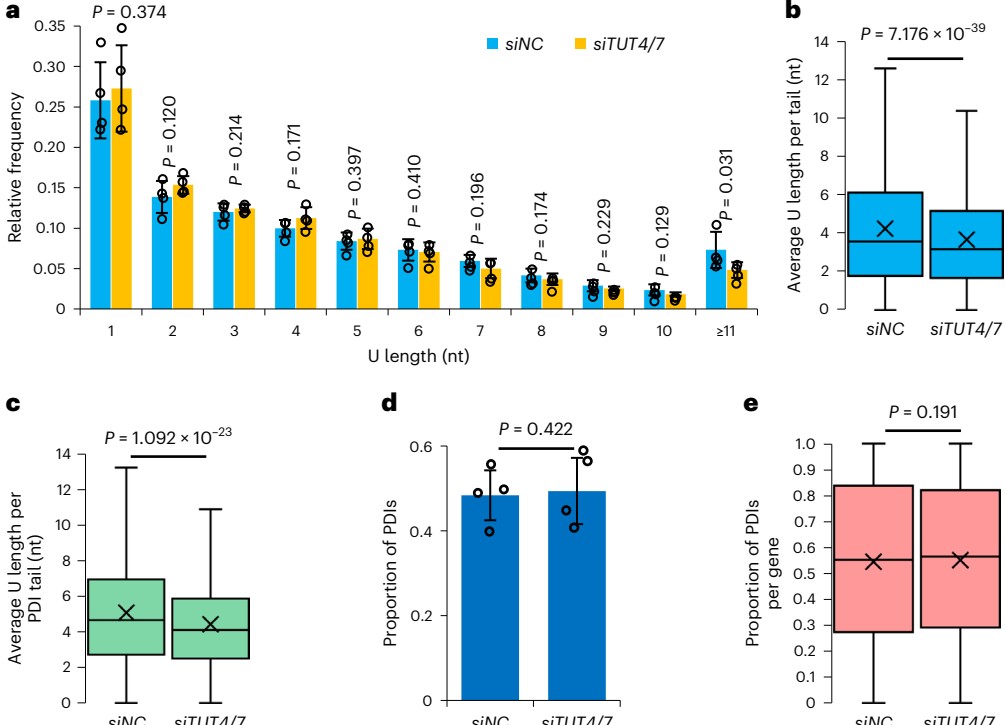

**Fig. 6 | Role of TUT4 and TUT7 in maternal mRNA remodeling in human one-cell embryos. a**, Distribution of U length of mRNAs in *siNC* (*n* = 4) and *siTUT4/7* (*n* = 4) 1C embryos. The U length (*x* axis) is the number of the longest consecutive U residues within the poly(A) tail. The *P* values from U1 to U10 successively are 0.374, 0.120, 0.214, 0.171, 0.397, 0.410, 0.196, 0.174, 0.229, and 0.129. The *P* value for U≥11 is 0.031. **b**, Box plot of the average U length per tail in *siNC* and *siTUT4/7* 1C embryos (4,732 genes). The *y* axis (average U length per tail) is the sum of the number of U residues in all poly(A) tails divided by the total number of reads with poly(A) tails for each gene of each sample, which refers to a published analysis[68]. **c**, Box plots of the average U length per tail of PDIs in *siNC* and *siTUT4/7* 1C embryos (2,804 genes). The *y* axis (average U length per tail) is the sum of the number of U residues in poly(A) tails of PDIs divided by the total number of PDIs for each gene of each sample. **d**, The PDI level of mRNAs in *siNC* (*n* = 4) and *siTUT4/7* (*n* = 4) 1C embryos. **e**, Box plot showing the PDI level of individual genes in *siNC* and *siTUT4/7* 1C embryos (4,402 genes). Data lists for **a**–**e** are provided in Supplementary Tables 10, 14, and 6–7. The data here were measured by PAIso-seq. Error bars indicate the s.e.m. The *P* values were calculated by one-tailed Student's *t*-test. Genes with ≥20 reads are included for gene-level analysis. For all box plots, the '×' indicates the mean value, the horizontal bars show the median value, and the top and bottom of the box represent the value of 25th and 75th percentiles, respectively.

Transcript-level analysis revealed that the level of internal U residues that were no more than 10 nt was minimally affected, whereas the level of those longer than 10 nt decreased significantly in *TUT4*- and *TUT7*-knockdown human one-cell embryos (Fig. 6a and Supplementary Table 10). Gene-level analysis also revealed a significant reduction of uridylation upon *TUT4* and *TUT7* knockdown (Fig. 6b and Supplementary Table 14). These results reveal that long consecutive internal U residues are sensitive to *TUT4* and *TUT7* knockdown in human one-cell embryos.

Of note, the expression levels of *TUT4* and *TUT7* are relatively low (Extended Data Fig. 1b and Supplementary Table 4), and could not be detected in the PAIso-seq data for control and *TUT4*- and *TUT7*-knockdown human one-cell embryos. In addition, the transcriptional changes upon knockdown of *TUT4* and *TUT7* knockdown were moderate (Extended Data Fig. 1h and Supplementary Table 4).

### Internal U residues and re-polyadenylation

Next, we asked whether U residues affected re-polyadenylation. PDIs are produced through re-polyadenylation of maternal transcripts that undergo deadenylation and 3′-UTR partial degradation, and most PDIs contain internal U residues, making PDIs a good system for studying the relationship between U residues and re-polyadenylation. Internal U residues of PDIs were significantly reduced in *TUT4*- and *TUT7*-knockdown one-cell embryos (Fig. 6c and Supplementary Table 14), whereas the level of PDIs was not affected (Fig. 6d,e and Supplementary Tables 6 and 7). These results indicate that reduced uridylation likely does not

affect the re-polyadenylation of degradation intermediates in human early embryos.

### The role of TENT4A and TENT4B in maternal mRNA remodeling

TENT4A and TENT4B catalyze mRNA guanylation that shields mRNA from rapid deadenylation in somatic cells[69]. We performed siRNA-mediated knockdown of *TENT4A* and *TENT4B* simultaneously in human one-cell embryos (Extended Data Fig. 1c,e,f and Supplementary Table 4). Both transcript- and gene-level analyses showed that the number of internal G residues was significantly reduced in *TENT4A*- and *TENT4B*-knockdown human one-cell embryos (Fig. 7a,b and Supplementary Tables 8 and 9). We found that 1,192 genes were downregulated, whereas only 213 genes were upregulated in *TENT4A*- and *TENT4B*-knockdown one-cell embryos (Fig. 7c and Supplementary Table 4).

Extensive maternal mRNA remodeling occurs in the human one-cell stage; thus, the 1,192 genes may be targets of remodeling. The levels of PDIs (Fig. 7d,e and Supplementary Tables 6 and 7) and internal non-A residues (Fig. 7f,g and Supplementary Tables 8 and 9) for these 1,192 genes dramatically increased from metaphase II to the one-cell stage. Moreover, the levels of both PDIs (Fig. 7h and Supplementary Table 7) and internal non-A residues (Fig. 7i and Supplementary Table 9) for these 1,192 genes decreased significantly upon *TENT4A* and *TENT4B* knockdown. At the transcript level, the

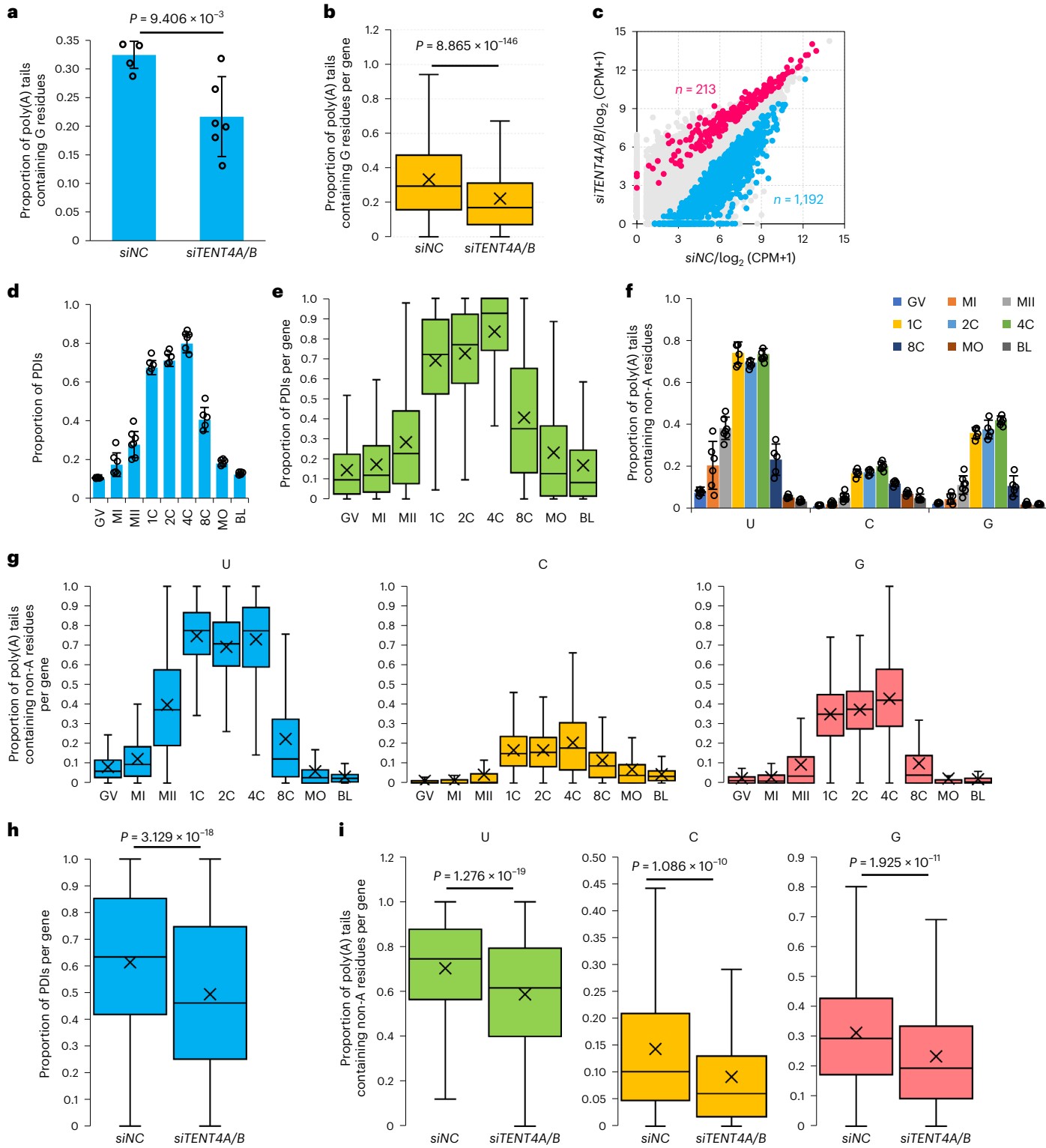

decrease in poly(A) tails with G residues for these 1,192 genes after *TENT4A* and *TENT4B* knockdown (*siNC*, 30.24%; *siTENT4A/B*, 25.44%) was similar to the trend seen at the gene level (Fig. 7i). These results suggest that TENT4A and TENT4B take part in the maternal mRNA remodeling through incorporation of internal G residues to stabilize the newly synthesized poly(A) tails.

### PAIso-seq2 analysis of human oocytes and embryos
We used PAIso-seq2 (ref. [41]) to measure transcripts that could not be or were inefficiently captured by PAIso-seq (Fig. 8a), and analyzed oocytes

at the germinal vesicle and metaphase II stages as well as embryos at the one-, two-, and four-cell stages using three to five oocytes or embryos per replicate, with two replicates (Extended Data Fig. 3a). We obtained 3.32 million poly(A)-inclusive cDNA reads mapped to the genome (Supplementary Table 15). Among all replicates, the second replicates for metaphase II, one-cell, and two-cell samples had a small number of detected reads, leading to detection of only a few genes in these three replicates (Extended Data Fig. 3b and Supplementary Table 4). In addition, the sensitivity of PAIso-seq2 was much lower than that of PAIso-seq (Extended Data Fig. 3d).

**Fig. 7 | Role of TENT4A and TENT4B in maternal mRNA remodeling in human one-cell embryos. a**, The frequency of mRNA poly(A) tails containing G residues in *siNC* (*n* = 4) and *siTENT4A/B* (*n* = 6) 1C embryos. The proportion (*y* axis) shows the number of poly(A) tails with G residues divided by the total number of reads with poly(A) tails for each sample. **b**, Box plot showing the frequency of G residues in poly(A) tails of individual genes in *siNC* and *siTENT4A/B* 1C embryos (4,474 genes). **c**, Gene expression level change in *siTENT4A/B* (*n* = 6) compared with *siNC* (*n* = 4) 1C embryos. The upregulated and downregulated genes upon *TENT4A* and *TENT4B* knockdown are shown in pink and blue (|log$_2$(fold change)| ≥ 0.5, *P* < 0.05). **d**, The PDI level of mRNAs of 1,192 downregulated genes upon *TENT4A* and *TENT4B* knockdown in oocytes and early embryos. **e**, Box plot of the PDI level of individual genes downregulated upon *TENT4A* and *TENT4B* knockdown (number: GV, 978; MI, 666; MII, 266; 1C, 579; 2C, 518; 4C, 425; 8C, 258; MO, 639; BL, 621). **f**, Frequency of non-A residues in mRNAs of 1,192 downregulated genes upon *TENT4A* and *TENT4B* knockdown in oocytes and early embryos. **g**, Box plot of the frequency of poly(A) tails containing non-A residues for genes that are downregulated following *TENT4A* and *TENT4B* knockdown (number: GV, 1,019; MI, 702; MII, 282; 1C, 611; 2C, 556; 4C, 445; 8C, 272; MO, 701; BL, 678). **h**, Box plot of the PDI level of individual genes that are downregulated following *TENT4A* and *TENT4B* knockdown in *siNC* and *siTENT4A/B* 1C embryos (397 genes). **i**, Box plot of frequency of poly(A) tails containing U, C, or G residues in individual genes that are downregulated following *TENT4A* and *TENT4B* knockdown in *siNC* and *siTENT4A/B* 1C embryos (420 genes). Number of replicates in **d** and **f** (GV, 5; MI, 6; MII, 7; 1C, 5; 2C, 5; 4C, 6; 8C, 5; MO, 5; BL, 5). Data lists for **a**–**i** are provided in Supplementary Tables 4 and 6–9. The data here are measured by PAIso-seq. Error bars indicate the s.e.m. The *P* values were calculated by one-tailed Student's *t*-test. Genes with ≥20 reads are included for gene-level analysis. For all box plots, the '×' indicates the mean value, the horizontal bars show the median value, and the top and bottom of the box represent the value of 25th and 75th percentiles, respectively.

In the PAIso-seq2 dataset, most mRNAs had poly(A) tail lengths in the range of 6–100 nt, with two major peaks at around 12 nt and 36 nt in the germinal vesicle stage (Fig. 8b and Supplementary Table 2). There was a large decrease in the relative abundance of transcripts with poly(A) tails in the range of 20–72 nt and a large increase in the abundance of those below 20 nt during oocyte maturation, which was not seen in the PAIso-seq data (Fig. 8b and Supplementary Table 2). After fertilization, the relative abundance of transcripts with poly(A) tails in the range of 12–40 nt increased (Fig. 8b and Supplementary Table 2). Except for the appearance of a large amount of poly(A) tails shorter than 20 nt in the metaphase II stage, the trends in poly(A)-tail-length changes observed in the PAIso-seq2 data were largely consistent with those in the PAIso-seq data (Fig. 1d). Taking advantage of PAIso-seq2 in capturing very short poly(A) tails, we detected a very large amount of poly(A) tails shorter than 20 nt in metaphase II oocytes, which were products of global maternal mRNA deadenylation during oocyte maturation, leading to overall shorter poly(A) tails in the PAIso-seq2 data, expect at the germinal vesicle stage (Figs. 1d and 8b, Extended Data Fig. 3e, and Supplementary Table 16).

Further analysis of PAIso-seq2 data revealed that the patterns of dynamic changes of internal non-A residues (Fig. 8c, Extended Data Fig. 4a,d, and Supplementary Tables 8 and 9), PDIs (Fig. 8d, Extended Data Fig. 4b, and Supplementary Tables 6 and 7), length of consecutive U residues (Extended Data Fig. 4c and Supplementary Table 10), and N length of uridylated poly(A) tails (Extended Data Fig. 4e and Supplementary Table 13) during the human OET were consistent with those observed in PAIso-seq data. PAIso-seq2 can detect the non-A residues at the 3′-ends that PAIso-seq cannot detect. Interestingly, there was a very high level of 3′-end U residues in the metaphase II and four-cell stages, whereas the levels in the one- and two-cell stages were low and comparable to that in the germinal vesicle stage (Fig. 8c, Extended Data Fig. 4a,d (bottom), and Supplementary Tables 8 and 9) which were different from the internal ones, suggesting that there are two waves of uridylation during the human OET. Together, PAIso-seq and PAIso-seq2 data are complementary and mutually supportive, which

cross-validates the findings that maternal mRNA remodeling occurs with global deadenylation followed by re-polyadenylation during the human OET.

**Maternal mRNA remodeling is necessary for the first cleavage**

3′-deoxyadenosine (3′-dA) is an adenosine analog that is converted to 3′-dATP in cells (Extended Data Fig. 5a), which can be incorporated into poly(A) tails to prevent further cytoplasmic polyadenylation owing to the absence of a 3′ hydroxyl group[70–74]. To test the role of re-polyadenylation after fertilization, we treated the embryos with 3′-dA immediately following fertilization through intracytoplasmic sperm injection (ICSI) in five independent experiments (Fig. 8e). There were 11 3′-dA-treated two-pronuclear one-cell embryos in total, all of which were arrested at the one-cell stage, whereas about 95% of non-treated two-pronuclear one-cell embryos could complete the first cleavage. Although 3′-dA can interfere with transcription in actively transcribing cells[75], the phenotype observed here was not due to the transcription interference, because human embryos could develop to the eight-cell stage normally after global transcriptional inhibition by α-amanitin[76,77].

Next, we performed PAIso-seq2 on the 3′-dA-treated human one-cell embryos (Fig. 8e). One of the 3′-dA replicates was lost during library preparation. To minimize the use of human embryos and because one replicate of the 3′-dA-treated sample yielded compelling results, we proceeded with one replicate. The levels of both PDIs (Fig. 8f, Extended Data Fig. 5b, and Supplementary Tables 6 and 7) and internal non-A residues (Fig. 8g (top), Extended Data Fig. 5c, and Supplementary Tables 8 and 9) decreased significantly in 3′-dA-treated human one-cell embryos. Moreover, 3′-dA treatment led to an obvious decrease in the level of poly(A) tails in the range of 15–40 nt and an increase in the range below 15 nt (Fig. 8h and Supplementary Table 2), the opposite of our prior observation of the change from the metaphase II stage to the one-cell stage (Fig. 8b), further confirming that re-polyadenylation is blocked by 3′-dA. In addition, we did not observe a decrease of 3′-end U residues in 3′-dA-treated human one-cell embryos (Fig. 8g (bottom), Extended Data Fig. 5d, and Supplementary Tables 8 and 9), indicating

**Fig. 8 | Human oocytes and early embryos analyzed by PAIso-seq2. a**, Diagram showing the principle and capability of capturing different types of transcripts for PAIso-seq and PAIso-seq2. **b**, Global distribution of poly(A) tail lengths of mRNAs measured by PAIso-seq2. Median poly(A) tail length (nt): GV, 40; MII, 25; 1C, 21; 2C, 17; 4C, 21. **c**, Frequency of non-A residues separated by internal and 3′-end positions in oocytes and early embryos, measured by PAIso-seq2 (two replicates each). **d**, The PDI level of mRNAs in oocytes and early embryos, measured by PAIso-seq2 (two replicates each). **e**, Illustration of 3′-dA treatment experiments in 1C embryos for PAIso-seq2 analysis. **f**, Box plots showing the PDI level of individual genes (29 genes with ≥20 reads in both samples) in control (Con) and 3′-dA-treated 1C embryos. **g**, Box plot of frequency of non-A residues of individual genes (29 genes with ≥20 reads in both samples) in control and 3′-dA-treated 1C embryos. **h**, Global distribution of poly(A) tail lengths of mRNAs in control and 3′-dA-treated 1C embryos. The bottom histogram is zoomed in view of the upper histogram between 1–50 nt. Data lists for **b**–**d** and **f**–**h** are provided in Supplementary Tables 2 and 6–9. The data here were measured by PAIso-seq2. Error bars indicate the s.e.m. The *P* values were calculated by one-tailed Student's *t*-test. Genes with ≥20 reads were included for gene-level analysis. For all box plots, the '×' indicates the mean value, the horizontal bars show the median value, and the top and bottom of the box represent the value of 25th and 75th percentiles, respectively.

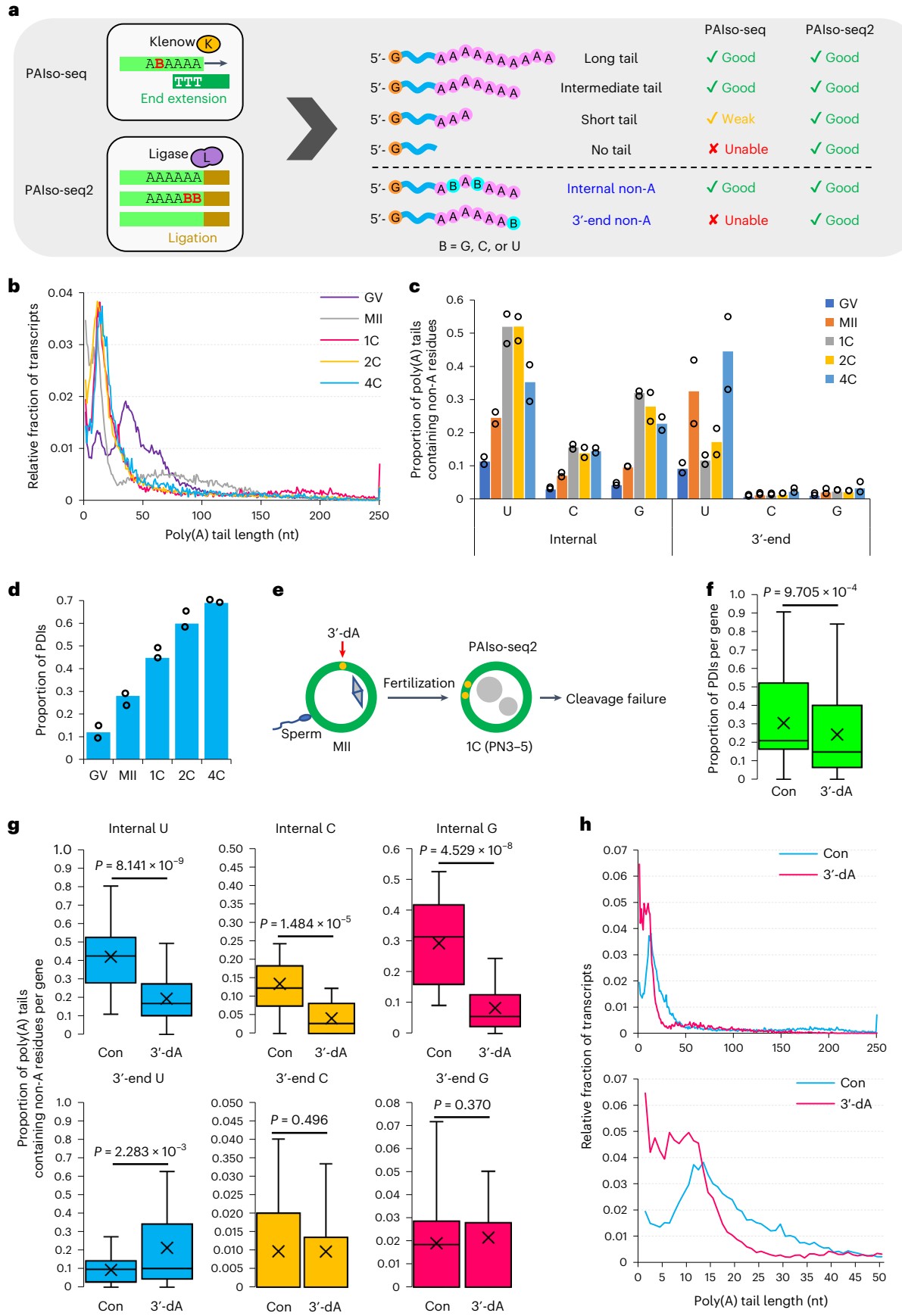

that blockage of re-polyadenylation prevents the conversion of 3′-end U residues to internal U residues. Together, these results suggest that maternal mRNA re-polyadenylation after fertilization has essential roles in early embryo development in humans.

## Discussion

This study reveals unexpected dynamic changes of PDIs and poly(A) tail non-A residues in the maternal transcripts during the human OET (Extended Data Fig. 5e (left)). Maternal mRNAs undergo BTG4-dependent global deadenylation during oocyte maturation, followed by either mRNA body decay or cytoplasmic re-polyadenylation. Interestingly, a large amount of the degradation intermediates are not further degraded, which undergo cytoplasmic re-polyadenylation. The cytoplasmic re-polyadenylation can be uridylation followed by re-polyadenylation or direct re-polyadenylation. These re-polyadenylation events are associated with G residues incorporated by TENT4A and TENT4B, which potentially stabilize the re-polyadenylated tails. Importantly, re-polyadenylation in the one-cell embryos is essential for the first cleavage (Extended Data Fig. 5e (right)).

More than 60% of the transcripts are PDIs in one-, two-, and four-cell embryos. We are confident that these PDIs are generated by polyadenylation on partially degraded transcripts, but not by regular PAS-cleavage-coupled polyadenylation, for the following four reasons. First, the high level of PDIs is with their polyadenylation sites within the coding sequences (CDS) (around 10% in PAIso-seq data for the one-, two-, and four-cell stages, while only around 1% in the germinal vesicle and blastocyst stages). Second, if the new PASs are generated through canonical cleavage-coupled polyadenylation, we would expect that the cleavage sites would cluster near the polyadenylation signal sites; however, we see largely even distribution upstream of the original PAS. Third, new transcription is minimal in human one-, two-, and four-cell embryos[43–48]; therefore, it is very unlikely that there are highly abundant cleavage and polyadenylation events that are generally transcription coupled. Forth, the results from knockdown of *BTG4*, *TUT4* and *TUT7*, or *TENT4A* and *TENT4B* support dynamic post-transcriptional regulation of poly(A) tails by these factors.

The maternal mRNA remodeling brings forward many interesting directions for the future. Are there other factors involved in mRNA deadenylation and decay, such as RNA m⁶A modification, that also contribute to this process? What factors protect these degradation intermediates from further degradation? What poly(A) polymerases or other factors are responsible for the massive re-polyadenylation? What are the roles of global re-polyadenylation? Why does the global poly(A)-tail-length distribution generally have two peaks? A previous study has found mRNAs with non-canonical poly(A) sites in pre-ZGA zebrafish embryos, which were thought to be produced by cytoplasmic polyadenylation of degradation intermediates[78] and were similar to the PDIs described here. Therefore, we'd expect conserved features for the PDIs in vertebrate embryos. Zebrafish embryos may be a good system for exploring the above questions about the mechanism and function of PDIs.

TUT4- and TUT7-mediated uridylation is coupled with rapid mRNA degradation in human somatic cells and during mouse oogenesis[9,68]. However, during the human OET, uridylated transcripts account for up to two-thirds of the maternal mRNAs. Transcripts with internal U residues become drastically increased at the one-cell stage and are maintained stably until the four-cell stage, spanning about two days until degradation at the eight-cell stage, when ZGA takes place. Therefore, mRNAs with uridylation at their 3′-ends do not go through immediate degradation in human oocytes and early embryos, but can be stabilized and further re-polyadenylated to form a new type of poly(A) tail with U residues followed by a stretch of A residues. Identification of the stage-specific biochemical mechanisms responsible for stabilization versus degradation of transcripts with U residues represents an interesting research direction for future studies. Furthermore, the large amount of maternal transcripts with internal U residues may promote their degradation at the eight-cell stage, which warrants further investigation.

Blocking maternal mRNA re-polyadenylation after fertilization led to first cleavage failure of human one-cell embryos. A recent clinical genetic study has revealed that maternal mutation of *BTG4* also leads to the first cleavage failure of human one-cell embryos[25,79]. Therefore, poly(A) tails of maternal mRNAs need to be tightly regulated to ensure successful human embryonic development. The mechanistic link between regulation of poly(A) tails and embryonic development warrants further investigation. The relationship between poly(A) tail dynamics and mRNA translational efficiency[80,81] is an obvious direction to explore.

In conclusion, we reveal extensive dynamic poly(A) tail changes and provide evidence of potential regulatory mechanisms during the human OET. As poly(A) tails are universal in eukaryotic mRNAs, poly(A) tail length and non-A-residue-mediated post-transcriptional regulations can be general mechanisms that control diverse biological or disease processes.

## Online content

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

## Methods

### Human oocytes and embryos

The collection and use of human gametes and embryos in this study follow these guidelines: Human Biomedical Research Ethics Guidelines (set by National Health Commission of the People's Republic of China in 2016), the 2016 Guidelines for Stem Cell Research and Clinical Translation (issued by the International Society for Stem Cell Research, ISSCR), and the Human Embryonic Stem Cell Research Ethics Guidelines (set by China National Center for Biotechnology Development on 24 December 2003). The aim and protocols of this study are in compliance with the above ethical regulations and have been reviewed and approved by the Institutional Review Board of Reproductive Medicine, Shandong University.

Immature oocytes in either the germinal vesicle or metaphase I stage were donated by individuals taking intracytoplasmic sperm injection (ICSI) treatments, and these immature oocytes were not used in regular clinical practice. The donor women are 25–38 years old with tubal-factor infertility, and their partners have healthy semen. In general, immature oocytes obtained in controlled ovarian hyperstimulation cycles were not used for subsequent clinical practice, because the development efficiency of embryos from immature oocytes was low, and participants generally had enough mature oocytes. Therefore, the research purpose was clearly explained to participants with a large number of follicles before oocyte retrieval, to see whether they would be willing to donate immature oocytes for scientific research with no compensation; participants were also assured that the donated oocytes would be used for only research, not any clinical purposes. Written informed consent was obtained from all donors. When obviously immature germinal vesicle or metaphase I oocytes were identified by an embryologist during oocyte denuding, another embryologist would confirm the oocyte maturity and then check whether the participant had signed the informed consent for donation. The oocytes that met the requirements were collected for subsequent scientific research. The sperm is cryopreserved normal semen donated for research purposes from men no older than 35, with written informed consent.

In vitro maturation, ICSI, and other oocyte processing steps were completed in the scientific research laboratory, which is physically separated from the clinical laboratory. The source and destination of all the donated samples were recorded according to the regulations to ensure that they could be tracked.

Metaphase II oocytes were from denuded germinal vesicle or metaphase I oocytes that were kept in in vitro maturation (IVM) medium at 37 °C in an atmosphere with 5% $CO_2$ for 23–27 hours (staring from germinal vesicle stage) or for 18–24 hours (staring from the metaphase I stage)[82]. The IVM medium consists of M199 medium (GIBCO, 11-150-059) with 20% serum substitute supplement (Irvine Scientific, 99193) and 75 mIU/ml of recombinant follicle stimulating hormone (Merck Serono, Gonal-f).

The early embryos at each developmental stage without treatment were prepared as described below, while the early embryos that were treated with an inhibitor or siRNA were prepared as described in the '3'-dA treatment' and 'Gene knockdown by siRNA' sections. For early embryos without treatment, the in vitro-matured oocytes described above were fertilized using donated sperm through ICSI. Then the embryos were cultured in G1.5 medium (Vitrolife, 10128) in a humidified atmosphere at 37 °C with 6% $CO_2$ in air around 17–19 hours at the one-cell stage, 27 hours at the two-cell stage, 48 hours at the four-cell stage, 3 days at the 8-cell stage, 4 days at the morula stage, and 5 days at the blastocyst stage to be vitrified, as described in the previous study[83]. Vitrification was done by incubating the embryos in Vitrification Solution 1 (8% ethylene glycol and 8% dimethyl sulfoxide (DMSO) in Cryobase (10 mM HEPES-buffered media containing 20 mg/ml human serum albumin and 0.01 mg/ml gentamicin)) at room temperature for 11 minutes. After initial shrinkage, embryos with original volume were transferred into Vitrification Solution 2 (16% ethylene glycol, 16% DMSO, and 0.68 M trehalose in Cryobase) for 1–1.5 minutes. Then, the

embryos were transferred onto Cryotop strip in a very small volume of solution (<0.1 μl) and plunged into liquid nitrogen. The Cryotop with the protective cover added was transferred into liquid nitrogen for storage. Thawing of the vitrified embryos was done by removing them from the liquid nitrogen after removal of the protective cover, and then immersing them in 2.5 ml of 37 °C Warming Solution 1 (1 M trehalose in Cryobase) for 1 minute on a heated stage. Embryos were then transferred to 0.5 ml of Warming Solution 2 (0.5 M trehalose in Cryobase) for 3 minutes, and then placed into 0.5 ml Cryobase for 5 minutes, followed by fresh 0.5 ml Cryobase for 1 minute. Embryos were finally transferred to G1.5 or G2 medium (Vitrolife, 10131) for evaluation of embryo quality. Embryos of good quality were washed with 1× phosphate-buffered saline (PBS, Invitrogen, AM9625) containing 0.1% bovine serum albumin (BSA, Sigma-Aldrich, A1933) three times and were collected into PCR tubes with a very small volume of buffer. The oocytes and embryos were randomly assigned to experimental groups. A single oocyte or embryo was used for PAIso-seq analysis with 4–9 replicates for each stage (details in Fig. 1b). Three to five oocytes or embryos were used for each PAIso-seq2 replicate (details in Extended Data Fig. 3a). No statistical methods were used to pre-determine sample sizes, but our sample sizes are similar to those used in previous publications[44,45]. All embryos used in this study were cultured for no more than 7 days and were used only for molecular analyses.

### 3'-dA treatment

3'-dA (Sigma, C3394) was directly dissolved using G1.5 medium to a final concentration of 2 mM. The medium without 3'-dA was used as a control. The in vitro matured metaphase II oocytes described above were fertilized through ICSI and cultured immediately in control medium or medium containing 3'-dA. The embryos were either monitored for development or collected at the PN3–5 stage for PAIso-seq2 library construction.

### Gene knockdown by siRNA

The germinal vesicle oocytes were microinjected with 5–10 pl siRNA, matured in vitro to the metaphase II stage as described above, fertilized through ICSI, and cultured until collection at the PN3–5 stage for PAIso-seq analysis. The siRNAs against *BTG4*, *TUT4*, *TUT7*, *TENT4A*, *TENT4B*, and negative control were purchased from ON-TARGETplus SMARTpool (Dharmacon, https://grcf.jhmi.edu/dna-services/sishrna/dharmacon/). siRNA sequence information is included in Supplementary Table 17. The concentration used for injection was 10 μM for control siRNA and *siBTG4*. For *siTUT4/7* and *siTENT4A/B*, used to knockdown two genes simultaneously, equal amounts of the siRNA against each gene were mixed to a final concentration of 10 μM.

### PAIso-seq library construction

A single human oocyte or embryo (details in Fig. 1b) was washed with 1× phosphate-buffered saline (PBS, Invitrogen, AM9625) containing 0.1% bovine serum albumin (BSA, Sigma-Aldrich, A1933) three times, and transferred into a 0.2-ml thin-walled PCR tube containing 2.5 μl of cell lysis buffer (0.2% Triton X-100 (Sigma-Aldrich, T9284) containing 2 U/μl of RNase inhibitor (TaKaRa, 2313A)) using a micro capillary pipette in the lowest possible volume (around 0.5 μl) to a final volume of around 3 μl. Then samples were incubated at 85 °C for 5 minutes for lysis and denaturation of the RNA, then put on ice immediately. The single-oocyte/embryo PAIso-seq library construction was carried out following our recently published detailed protocol[40]. The libraries were sequenced on a PacBio Sequel or Sequel II System under HiFi mode according to the standard PacBio Iso-Seq procedures at Annoroad (a sequencing service provider in China, http://www.annoroad.com/).

### PAIso-seq2 library construction

**Sample collection and lysis.** Three to five (details in Extended Data Fig. 3a) human oocytes or embryos were washed with 1× PBS containing

0.1% BSA three times, and transferred into a 0.2-ml thin-walled PCR tube containing 2.5 µl of cell lysis buffer using a micro capillary pipette in the lowest possible volume (around 0.5 µl) to a final volume of around 3 µl. Then samples were incubated at 85 °C for 5 minutes for lysis and denaturation of the RNA, and then put on ice immediately. The samples were ready for PAIso-seq2 library preparation as described briefly below.

**3′-end adapter ligation to preserve poly(A) tails.** One microliter of 3′-end adapter (20 µM, Supplementary Table 18), 3 µl of nuclease-free water, and 13 µl adapter ligation mix (final concentration: 1× T4 RNA Ligase 2 truncated KQ reaction buffer (NEB, M0373L), 10 U/µl of T4 RNA Ligase 2 truncated KQ (NEB, M0373L), 2 U/µl of RNase inhibitor, and 15% PEG8000 (NEB, M0373L)) was added to each of the samples, which were incubated at 16 °C for 16 hours. The ligation reaction was stopped by heating at 65 °C for 20 minutes. Then the samples with different barcodes were mixed together into one tube and purified using RNA Clean & Concentrator-5 kit in accordance with the manufacturer's guidelines. Briefly, following binding and washing, the adapter-ligated RNA was eluted with 7 µl nuclease-free water.

**Reverse transcription with template switching.** Each sample was added with 0.4 µl of RT primer (100 µM, Supplementary Table 18) and 1 µl of dNTP mix (10 mM each), incubated at 72 °C for 3 minutes, and put on ice immediately, to anneal the RT primer to the 3′-end adapter of RNAs. After adding 11.6 µl of the RT mix (final concentration: 1× SuperScript II first-strand buffer, 10 U/µl of SuperScript II reverse transcriptase (Invitrogen, 18064-014), 1 U/µl of RNase inhibitor (TaKaRa, 2313A), 5 mM DTT (Invitrogen, 18064-014), 1 M Betaine (Sigma-Aldrich, 61962), 6 mM $MgCl_2$ (Invitrogen, AM9530G), and 0.98 µM TSO (Supplementary Table 18)), the sample was incubated at 42 °C for 90 minutes; 10 cycles of 50 °C for 2 minutes and 42 °C for 2 minutes; 70 °C for 15 minutes; and held at 4 °C.

**cDNA synthesis.** Twenty microliters of nuclease-free water and 1 µl of Ribonuclease H (TaKaRa, 2151) were added to each of the samples, which were incubated at 37 °C for 20 minutes. Then, 100 µl of KAPA HiFi HotStart ReadyMix (2×), 30 µl of IS PCR primer (10 µM, Supplementary Table 18), and 29 µl of nuclease-free water were added to the samples, followed by PCR with the following program: 98 °C for 3 minutes; 3 cycles of 98 °C for 20 secconds, 67 °C for 15 seconds, and 72 °C for 6 minutes; 72 °C for 10 minutes; and held at 4 °C. Then, the double-stranded cDNA was purified using 0.8× SPRIselect beads and eluted with 50 µl of nuclease-free water.

**Ribosomal sequence removal.** PAIso-seq2 uses CRISPR–Cas9-mediated removal of cDNA from rRNA, as described in the previous study with minor modifications[84]. The templates for sgRNA targeting ribosomal sequence were prepared by PCR amplification with forward primers containing a T7 promoter, 20-nt variable protospacer sequences targeting the human ribosomal sequence, a 20-nt sequence paired with the 5′-end of the sgRNA backbone (Supplementary Table 19), and a reverse primer paired with a 30-nt sequence of the 3′-end of the sgRNA backbone (Supplementary Table 19), a plasmid template containing the sgRNA backbone (pX330)[85], and KOD-Plus-Neo DNA Polymerase (TOYOBO, KOD-401). The DNA templates were used for in vitro transcription (IVT) to produce sgRNAs using the HiScribe T7 Quick High Yield RNA Synthesis Kit (New England Biolabs, E2040S). Next, the IVT products were cleaned with RNA Clean & Concentrator-5 Kit according to the manufacturer's instructions. The purified sgRNAs were stored at −80 °C. Each sample was digested with Cas9 digestion mix (final concentration: 1× NEBuffer 3.1 (New England Biolabs, B7203), 200 nM Cas9 nuclease (New England Biolabs, M0386M), and 8–15 ng/µl of the above sgRNA targeting human genes encoding nuclear and mitochondrial rRNA) and incubated at 37 °C for 5 hours. After Cas9 digestion, 2 µl of RNase A (TaKaRa, 2158) was added to digest sgRNAs

with incubation at 37 °C for 30 minutes. Then the mixture was purified using 0.8× SPRIselect beads and eluted with 20 µl of nuclease-free water.

**PCR preamplification.** Twenty-five microliters of KAPA HiFi HotStart ReadyMix (2×, KAPA Biosystems, KK2601) and 5 µl of IS PCR primer (10 µM, Supplementary Table 18) were added to each sample. Preamplification was performed with the following program: 98 °C for 3 minutes; 15 cycles of 98 °C for 20 seconds, 67 °C for 15 seconds, and 72 °C for 6 minutes; 72 °C for 10 minutes; and hold at 4 °C. Then the preamplification product was purified using 0.8× SPRIselect beads (Beckman Coulter, B23318) and eluted with 20 µl of nuclease-free water. The concentration of the purified preamplification product was measured via Fluorometer (DeNovix, DS-11 FX+).

**Large-scale PCR.** Twenty nanograms of purified preamplification product was added with 400 µl of KAPA HiFi HotStart ReadyMix (2×), 80 µl of IS PCR primer (10 µM, Supplementary Table 18), and nuclease-free water to achieve an 800 µl mix, which was then split into 16 × 50-µl tubes for large-scale PCR with the following program: 98 °C for 3 minutes; 10 cycles of 98 °C for 20 seconds, 67 °C for 15 seconds, 72 °C for 6 minutes; 72 °C for 10 minutes; and hold at 4 °C. Then, the large-scale PCR product was purified using 0.8× SPRIselect beads and eluted with 100 µl of nuclease-free water. The concentration of the purified large-scale PCR product was measured via Fluorometer (DeNovix, DS-11 FX+).

**SMRTbell library construction and sequencing.** SMRTbell library construction was performed using SMRTbell Template Prep Kit in accordance with the manufacturer's guidelines with purified double-stranded cDNA from the large-scale PCR. Then the SMRTbell library was sequenced on a PacBio Sequel II System under HiFi mode according to the standard PacBio Iso-Seq procedures at Annoroad (a sequencing service provider in China, http://www.annoroad.com/).

### PAIso-seq sequencing data processing

**Raw circular consensus sequencing reads conversion from subreads.** The sequencing data in .subreads.bam files off the PacBio sequencing instrument were provided by the sequencing service provider. Then, highly accurate single-molecule circular consensus sequencing (CCS) reads were generated using ccs software (https://github.com/PacificBiosciences/ccs, version 5.0.0) and converted to fastq format using bam2fastq software with the pbbam package (https://github.com/PacificBiosciences/pbbam, version 1.0.6). The number of passes for each of the raw CCS reads was generated using GetCCSpass.pl (https://github.com/Lulab-IGDB/polyA_analysis/blob/main/bin/).

**Clean circular consensus sequencing reads extraction from raw circular consensus sequencing reads.** The barcode split clean reads were extracted from CCS reads in fastq format using CCS_split_clean_end_extension_v1.py (https://github.com/Lulab-IGDB/polyA_analysis/blob/main/PAIso-seq/). The output file contains seven columns, including CCS ID (column 2, containing information of CCS name, barcode, and pass number with colon delimiter), clean CCS read sequence (column 6), and quality value (column 7), and was then converted into fastq format.

**Human genome alignment of clean circular consensus sequencing reads.** The clean CCS reads in fastq format were aligned to human reference genome (GRCh38) using minimap2 (https://github.com/lh3/minimap2, version v2.15)[86] with the following parameters '-ax splice -uf−secondary=no -t 40 -L−MD−cs−junc-bed human.genome.bed human.genome.mmi'. The reference file human.genome.bed was built from the gtf format annotation file of human genome (human.genome.

gtf, https://ftp.ebi.ac.uk/pub/databases/gencode/Gencode_human/release_36/gencode.v36.annotation.gtf.gz) using paftools.js, a script in minimap2 software; and the reference file human.genome.mmi was built from human genome sequences in fasta format (https://ftp.ebi.ac.uk/pub/databases/gencode/Gencode_human/release_36/GRCh38.primary_assembly.genome.fa.gz) using minimap2 software with the following parameters '-x splice -t 20'. Then, the files were filtered by samtools software (http://samtools.sourceforge.net, version 1.9) with the following parameters '-F 3844 -bS' (https://broadinstitute.github.io/picard/explain-flags.html). At this point, the mapped CCS reads in bam format were ready for poly(A) tail extraction. About 10,000–700,000 mapped reads were obtained from single human oocytes or preimplantation embryos (Supplementary Table 1). One of the samples among the 73 individual oocytes or embryos failed in PAIso-seq (MI-7, Supplementary Table 1), which contained only 92 extracted CCS reads or 86 mapped reads.

**Annotation of the mapped clean CCS reads.** The mapped clean CCS reads were assigned to annotated genes using featureCounts software in subread package (http://subread.sourceforge.net/) with the following parameters '-L -g gene_id -t exon -s 1 -R CORE human.genome.gtf'. The output file *.clean.filter.bam.featureCounts will be used for the following poly(A) tail annotation step, which contains four columns, including CCS ID (column 1) and gene ID (column 4).

**Poly(A) tail extraction for mapped clean CCS reads.** The poly(A) tail of each mapped clean CCS read in bam format was extracted using PolyA_trim_V5.4.1.py (https://github.com/Lulab-IGDB/polyA_analysis/blob/main/bin/). Alignments with the 'SA' (supplementary alignment) tags were ignored, and the terminal clipped sequence of the aligned CCS reads was used as candidate poly(A) tail sequence. According to previous reports[26,31,39], the majority of the residues in poly(A) tails were A residues, and if the poly(A) tails contain non-A residues, they were randomly distributed without a defined pattern. Therefore, we call a poly(A) tail taking the proportion of U (presented as T in CCS reads, the same goes for the following), C, and G residues and the distribution pattern of U, C, and G residues within a tail into account. If the proportion of U, C, and G were all no less than 0.1 in the 3'-soft clip sequence, this read would be marked as 'HIGH_TCG'. We defined a continuous score based on the transitions between the two adjacent nucleotide residues throughout the 3'-soft clip sequences. From the 5'-end to the 3'-end of the 3'-soft clip sequence, a transition from one residue to the same residue was scored as 0, and a transition from one residue to a different residue scored as 1, and the sum was the score for the read. The reads would be marked as 'FALSE_score_12+' if they scored more than 12. After the above two steps, the 3'-soft clips classified as neither 'HIGH_TCG' nor 'FALSE_score_12+' were marked as 'TRUE'. Reads with 'TRUE' tags were used for the following poly(A) tail annotation and analysis.

**Poly(A) tail annotation for mapped clean CCS reads.** After poly(A) tail extraction and read annotation, the information about the length and residue content of the poly(A) tail (including 0-nt tails) of each mapped clean CCS read was summarized using PolyA_note_V2.1.py (https://github.com/Lulab-IGDB/polyA_analysis/blob/main/bin/). Each *.polyA_note.txt output file contains the following 13 columns of information: barcode, CCS ID, gene ID, pass number, '1', number of residue A, number of residue T, number of residue C, number of residue G, number of non-A residues, '0', poly(A) tail sequence, and average quality value of poly(A) tail bases. The *.polyA_note.txt files were ready for the analysis of poly(A) tail length and non-A residues.

In this manuscript, transcripts from protein-coding genes encoded in the nuclear genome (excluding genes encoding histones and histone variants) were used in all analyses, except for the saturation curve

analysis in Extended Data Fig. 3d, which used all the annotated genes. In this manuscript, gene expression and saturation curve analyses employed reads with CCS reads with at least one pass, while analyses involving poly(A) tails employed CCS reads with at least ten passes[87,88]. Reads with pass number at least 10 and poly(A) tail length at least 1 nt, were called poly(A)+ reads.

Each single oocyte or embryo was used as one replicate in the oocyte and embryo similarity analysis by uniform manifold approximation and projection (UMAP). The following pairs of oocytes were combined as one replicate in other analyses owing to shared barcode sequence: GV-1 and GV-6, GV-2 and GV-7, GV-3 and GV-8, GV-4 and GV-9, MI-1 and MI-7, as well as MI-2 and MI-8, resulting five replicates for the germinal vesicle and six replicates for metaphase I. Each single oocyte/embryo for the other PAIso-seq samples was used as one replicate in other analyses (replicate number: germinal vesicle, $n = 5$; metaphase I, $n = 6$; metaphase II, $n = 7$; one-cell stage, $n = 5$; two-cell stage, $n = 5$; four-cell stage, $n = 6$; eight-cell stage, $n = 5$; morula, $n = 5$; blastocyst, $n = 5$; siNC, $n = 4$; siTUT4/7, $n = 4$; siTENT4A/B, $n = 6$; siBTG4, $n = 4$). For the gene expression analyses, the above numbers of replicates were used. For gene-level analyses of poly(A) tail length, non-A residues, and PDIs, the replicates were combined for each stage. As at least 20 poly(A)+ reads were required for genes to be included in these analyses, if individual replicates were filtered with this cutoff, very few genes could be retained for analysis.

## PAIso-seq2 sequencing data processing
**Raw CCS reads conversion from subreads.** The steps were as described in the 'PAIso-seq sequencing data processing' section.

**Clean CCS reads extraction from raw CCS reads.** The barcode split clean reads were extracted from CCS reads in fastq format using CCS_split_clean_UMI_V4.py (https://github.com/Lulab-IGDB/polyA_analysis/blob/main/PAIso-seq2/). The output file was in the same format as described in the 'PAIso-seq sequencing data processing' section, and was then converted into fastq format.

Human genome alignment of clean CCS reads and annotation of the mapped clean CCS Reads were performed in the same way as described in the 'PAIso-seq sequencing data processing' section. In general, about 100,000–700,000 mapped reads were generated from each of the PAIso-seq2 replicates for 3–5 human oocytes or preimplantation embryos (Supplementary Table 15).

**Poly(A) tail sequence extraction for mapped clean CCS reads.** The poly(A) tail of each clean CCS read was extracted via PolyA_trim_last_exon_V5.5.2.1.py (https://github.com/Lulab-IGDB/polyA_analysis/blob/main/bin/). Reads were also added tags with 'TRUE', 'HIGH_TCG', or 'FALSE_score_12+' as described in the 'PAIso-seq sequencing data processing' section. The PAIso-seq2 data contain unique molecular identifiers (UMIs), which can be used to remove the PCR duplicates by pickOne_V2.py (https://github.com/Lulab-IGDB/polyA_analysis/blob/main/PAIso-seq2/). The UMI deduplicated reads were used for downstream PAIso-seq2 analysis. In addition, reads whose polyadenylation site was located in the last exon of its assigned gene are marked as 'Last_'. For PAIso-seq2 data for human oocyte and embryo, reads were frequently found to end at the middle of annotated transcripts, which were not polyadenylated, indicating that RNA fragmentation happened during sample collection, transportation, or preparation. Therefore, we focused on reads that end at the last annotated exons for poly(A) tail analysis for all the PAIso-seq2 data analysis of human oocytes and embryos.

**Poly(A) tail annotation for mapped clean CCS reads.** The steps were as described in the 'PAIso-seq sequencing data processing' section, but the output file was named *.polyA_note.UMI_uniq.txt to convey the information of UMI deduplication in the prior step.

**rRNA reads analysis.** The clean CCS reads were aligned to the human nuclear rRNA sequences (RNA45SN1, 45S pre-ribosomal N1, gene ID: 106631777) with bwa software (https://github.com/lh3/bwa) with the '-x pacbio' option. Then, the CCS ID of the reads aligned to nuclear rRNA were used for counting of rRNA reads or for filtering out rRNA reads. Reads assigned to ENSG00000211459.2 (MT-RNR1, mitochondrial 12S RNA) and ENSG00000210082.2 (MT-RNR2, mitochondrial 16S RNA) by minimap2 and featureCounts in the human genome alignment of clean CCS reads and annotation of the mapped clean CCS reads steps were considered mitochondrial rRNA reads. The proportion of rRNA remaining in the PAIso-seq2 datasets were calculated by dividing the number of nuclear rRNA reads or mitochondrial rRNA reads to the number of clean CCS reads. We found that about 20% of nuclear rRNA and about 3% of mitochondrial rRNA remained in the PAIso-seq2 datasets (Extended Data Fig. 3c), suggesting that most of the rRNA is removed in our PAIso-seq2 procedures.

## FLAM-seq sequencing data processing
The raw subreads data of FLAM-seq on Hela S3 cells, the human iPSCs, and human cerebral organoids were kindly provided by the authors of ref. [31].

**Raw CCS reads conversion from subreads.** The steps were as described in the 'PAIso-seq sequencing data processing' section.

**Clean CCS reads extraction from raw CCS reads.** The barcode split clean reads were extracted from CCS reads in fastq format using CCS_split_clean_UMI_FLAM-seq_V2.py (https://github.com/Lulab-IGDB/polyA_analysis/blob/main/bin/). Because GI tailing was used in library preparation of FLAM-seq[31], about nine residues (Gs) would be added after the poly(A) tails and before UMI and barcode sequences, thus the reads with fewer than seven Gs before UMI and barcode sequences were discarded. For each read with 7–16 Gs before the UMI and barcode sequence, the sequence before the Gs would be extracted as a clean CCS read. Although the ratio was very low (for example, 133 out of 49,307 in sample hiPSC_rep1), for each read with ≥17 Gs before UMI and barcode sequences, the sequences before 17 Gs would be extracted as a clean CCS read. The output file was in the same format as described in the 'PAIso-seq sequencing data processing' section, and was then converted into fastq format.

Human genome alignment of clean CCS reads, annotation of the mapped clean CCS reads, poly(A) tail extraction for mapped clean CCS reads, and CCS readspoly(A) tail annotation for mapped clean CCS reads were performed in a same way as described in the 'PAIso-seq sequencing data processing' section with the additional step of removing PCR duplicates based on UMI by pickOne_V2.py before poly(A) tail extraction and naming the output file *.polyA_note.UMI_uniq.txt to convey the UMI deduplication. At this point, the FLAM-seq data were ready for downstream analysis in the same way as PAIso-seq and PAIso-seq2 data.

## Measurement of poly(A) tail length
The PAIso-seq, PAIso-seq2, and FLAM-seq datasets were processed in the same way. The poly(A) tail length of a poly(A)+ transcript was the number of all bases within the poly(A) tail. The poly(A) tail length of a given gene was calculated by the geometric mean poly(A) tail length of all the transcripts assigned to it, because poly(A)-tail-length distribution of a gene follows a lognormal-like distribution[10].

## Detection of non-A residues in poly(A) tails
The PAIso-seq, PAIso-seq2, and FLAM-seq datasets were processed in the same way. The number of U, C, and G residues in the poly(A) tail of each CCS read were summarized in column 7, 8, and 9 of the output *.polyA_note.txt or *.polyA_note.UMI_uniq.txt file, respectively, and the complete poly(A) tail sequences were in column 12.

The transcript-level proportion of non-A residues was the number of transcripts with non-A residues divided by the total number of poly(A)+ transcripts. The gene-level proportion of non-A transcripts (CCS reads containing the given non-A residues) of a gene was calculated as the number of transcripts containing the given non-A residues divided by the total number of poly(A)+ transcripts for the gene.

U$n$ refers to poly(A) tails with an $n$ number of consecutive U residues. For grouping poly(A) tails based on longest consecutive non-A residues, U1 refers to poly(A) tails which contain a single U, U2 refers to poly(A) tails that contain UU, U≥3 refers to poly(A) tails that contain three or more consecutive Us, U2–5 refers to poly(A) tails which contain 2–5 consecutive Us, and U ≥ 6 refers to poly(A) tails which contain 6 or more consecutive Us. C and G residues were analyzed in the same way.

For assigning the poly(A) tails based on the positions of U residues in poly(A) tails, a given poly(A) tail was first scanned for 3′-end U residues. If 3′-end U residues were found, the poly(A) tail would be assigned to poly(A) tails with 3′-end U residues. If no 3′-end U residues were found, the poly(A) tail was further scanned for U residues and assigned to poly(A) tails with internal U residues if found. C and G residues were analyzed in the same way.

For calculating the N length for U residues, a given poly(A) tail was first searched for the longest consecutive U residues within it. The length of sequence located 5′ upstream of this longest consecutive stretch of U residues was considered the N length. If a given poly(A) tail contained multiple stretches of consecutive U residues of the same length which was longest, then the N length for this tail could not be determined and thus discarded from the N length analysis.

The average U length per tail in Fig. 6b is the sum of the number of U residues from all the transcripts for a given gene divided by the total number of reads for the given gene. The average U length per tail of PDI in Fig. 6c is the sum of the number of U residues from all the PDIs for a given gene divided by the total number of PDIs for the given gene.

## Polyadenylation site calling
To call the PASs for each gene, poly(A)+ transcripts in *.polyA_note.txt or *.polyA_note.UMI_uniq.txt files were analyzed via findAPA_v7.1.py (https://github.com/Lulab-IGDB/polyA_analysis/blob/main/bin/) following the rules of calling polyadenylation sites in the TAPIS package[89]. In brief, for each gene, the site with the most (≥10) reads in each 5 bp upstream and downstream was considered a PAS site, then the site with second most (≥10) reads in each 5 bp upstream and downstream and not within 20 nt from a called PAS was considered a PAS site, until no site with ≥10 reads in each 5 bp upstream and downstream existed. For genes with two PASs, the PAS site proximal to the transcription start site was called pPAS, and the PAS site distal to the transcription start site was called dPAS. The output *.APAsites.csv file contains the information of PASs called which can serve as a reference for the following analysis involving PAS.

## Polyadenylated intact transcripts and polyadenylated degradation intermediates assignment
The PASs called in GV oocytes were used as reference PASs for assigning the reads to them. Poly(A)+ reads in the *.polyA_note.txt or *.polyA_note.UMI_uniq.txt files for each of the stages can then be assigned to the PASs called in GV stage as described above using readsOnAPA_v4.py (https://github.com/Lulab-IGDB/polyA_analysis/blob/main/bin/). For genes with called PASs, one read was assigned to a PAS of the gene if it was located within 5 nt around this PAS, and was called a PIT. Reads from the gene that ended outside 5 nt of all its PASs and 5′upstream of at least one PAS were then considered as polyadenylated degradation intermediates (PDIs). For Hela S3, iPSC, and organoid samples, the PASs called in PAIso-seq data of GV oocytes were used as reference PASs for PDI and PIT analysis. The transcript-level proportion of PDIs was the number of PDIs divided by the sum of PDIs and PITs. The gene-level proportion of PDIs of a gene was calculated as the number of PDIs divided

by the sum (genes with the sum number at least 20 were included in the analysis) of the number of PDIs and PITs for the gene. Note that there were a small number of reads that could not be assigned as either PDI or PIT, which were discarded in the PDI- and PIT-related analysis. Therefore, the number of reads in the PDI- and PIT-related analysis was less than total CCS reads.

## Gene expression analysis

Gene expression was quantified as counts per million (CPM) according to previous studies[90,91]. CPM was calculated as: CPM = 1,000,000 × (number of reads assigned to a gene) / (sum of all reads). Differential gene expression between negative control and knockdown samples were statistically analyzed with Student's $t$-test.

## Uniform manifold approximation and projection analysis

CCS reads in *.polyA_note.txt for each of the oocyte and embryo PAIso-seq data were included in this analysis. The gene count matrix for protein-coding genes of these single oocytes or embryos were generated across the 73 oocytes or embryos, and cells with fewer than 600 detected genes were filtered out (the MI-7 sample was excluded due to only 41 genes detected). We performed standard preprocessing with Seurat_3.2.0 software[42] on 72 samples with 13,341 genes, including highly variable gene selection and scaling. Top 10 components were selected after performing principal component analysis (PCA). Based on the top 10 components, we performed UMAP analysis with Seurat_3.2.0 to present the distance of cells. Finally, we added the original label of human oocytes and embryos to the UMAP plot.

## Visualization of aligned PAIso-seq reads

To prepare files for visualization of poly(A) tails in the IGV genome browser, we extracted the CCS reads with at least ten passes[87,88] for genome browser view. Clean CCS reads with at least 10 passes that mapped to the genome was extracted from the minimap2 mapped reads in bam format, and data from individual oocytes or embryos from the same stage were combined to a single file in bam format (these bam files were available in a public data repository, see 'Data availability' for details). Then, the bam files for each of the stages were indexed using samtools to generate the index file in bam.bai format ($ samtools index input.bam). At this stage, the bam files for each of the stages were ready to be loaded into the integrative genomics viewer (IGV, https://software.broadinstitute.org/software/igv/) for visualization of the poly(A) tails for each of the stages with the human hg38 reference genomes.

For the IGV views shown in Fig. 2, to better present the polyadenylation events around the last exon of a given gene, the sequences from the beginning of the last annotated exon to the end of the clean CCS reads for the given gene for each of the stages were extracted and realigned to the genome using minimap2 with the same parameters as as described in the 'PAIso-seq sequencing data processing' section. Then, the mapped reads in bam format were indexed using samtools to generate the index file in bam.bai format ($ samtools index input. bam). Next, the bam files for each stage for the given gene were loaded into the IGV for visualization of the poly(A) tails for each of the stages with the human hg38 reference genomes. The IGV views here used the default setting, which would display all reads if there were no more than 100 reads or 100 random reads if there were more than 100. The visualization of reads in this way could represent the pattern of all reads. The sequences that match the reference genome are shown in gray (the dispersed colored regions in the mapped genomic regions indicated unmatched residues caused by single-nucleotide polymorphisms or sequencing errors); the poly(A) tails (soft clip sequences) that cannot map to the genome were shown in colors (forward strand: A residues, cyan; U residues, magenta; C residues, blue; G residues, dark yellow). Note that we combined all replicates of each stage for analysis here.

## Gene Ontology analysis

GO analysis was performed through the online analysis tool g:Profiler (https://biit.cs.ut.ee/gprofiler/gost) using Ensembl gene ID as the input.

## Statistics and reproducibility

The number of single oocytes and embryos sequenced by PAIso-seq was shown in Fig. 1b, and two replicates were performed for each of the PAIso-seq2 samples. The oocytes and embryos were randomly assigned to each experimental group. For data collection and analysis, researchers were not blinded to the conditions of the experiments. For the oocyte and embryo images in Fig. 1a, all the oocytes and embryos used were checked under microscope to confirm they had correct morphology, and one of them for each stage were photographed as an example of the oocyte or embryo morphology. MI-7 in the PAIso-seq dataset was excluded in the analysis of individual MI replicates due to low number of reads. One 3'-dA replicate for PAIso-seq2 dataset was lost during library preparation and was not included in the analysis. Levels of significance were calculated with Student's $t$-test if not specified otherwise in the figure legend. Levels of significance in Extended Data Fig. 5b–d were calculated with the $\chi^2$ test. The correlation coefficient in Extended Data Fig. 3b,e was Pearson's correlation coefficient. The regression lines in Extended Data Fig. 3e are linear regressions.

Data distribution for the statistical tests was assumed to be normal, but this was not formally tested.

## Genome and gene annotation

The genome sequence used in this study is from the following links: http://ftp.ebi.ac.uk/pub/databases/gencode/Gencode_human/release_36/GRCh38.primary_assembly.genome.fa.gz. The genome annotation used in this study is from the following links: http://ftp.ebi.ac.uk/pub/databases/gencode/Gencode_human/release_36/gencode.v36.primary_assembly.annotation.gtf.gz

## Reporting summary

Further information on research design is available in the Nature Portfolio Reporting Summary linked to this article.

# Data availability

The CCS data for human oocytes and embryos in bam format from PAIso-seq and PAIso-seq2 are available at Genome Sequence Archive for Human (GSA-Human: https://ngdc.cncb.ac.cn/gsa-human/) hosted by National Genomics Data Center (PAIso-seq: HRA001288, PAIso-seq2: HRA001289). Details for samples in HRA001288 and HRA001289 are shown in Supplementary Table 18. The bam files for visualization of the mapped reads in IGV are available at GSA-human (PAIso-seq: HRA001911, PAIso-seq2: HRA001912). The raw subreads data of FLAM-seq of Hela S3 cells, the human iPSCs, and human cerebral organoids were kindly provided by the authors of ref. [31].

# Code availability

Custom scripts used for data analysis are available in github: https://github.com/Lulab-IGDB/polyA_analysis.

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

## Acknowledgements

This work was supported by the National Key Research and Development Program of China (2018YFA0107001 and 2020YFA0804000 to F. L.), the Strategic Priority Research Program of the Chinese Academy of Sciences (XDA24020203 to F. L., XDA16010113 to B. Z.), the CAS Project for Young Scientists in Basic Research (YSBR-012 to F. L.), National Natural Science Foundation of China (31970588 to J. W., 32170606 to F. L., 81871168 to K. W., 32201060 to Y. L., 82192874 to H. Z.), Natural Science Foundation of Heilongjiang Province (YQ2020C003 to J. W.), the China Postdoctoral Science Foundation (2020M670516 and 2020T130687 to Y. L.), Shandong Provincial Key Research and Development Program (2020ZLYS02 to Z.-J. C.), Innovative Research Team of High-Level Local Universities in Shanghai (SHSMU-ZLCX20210201 to Z.-J. C.), the State Key Laboratory of Molecular Developmental Biology, and the Fundamental Research Funds of Shandong University.

## Author contributions

Y. L., J. W., and F. L. conceived the project and designed the study. Y. L. constructed the PAIso-seq and PAIso-seq2 libraries. Y. L., F. S., Y. Z., H. N., J. W., B. Z., and F. L. analyzed the sequencing data. H. Z., J. Z., C. L., Z. H., Z.-J. C., and K. W. collected human oocytes and embryos and performed drug treatment on human embryos and siRNA-mediated knock-down in human oocytes and embryos. Y. L. and J. W. organized all figures. Y. L., J. W., and F. L. supervised the project. Y. L., J. W., and F. L. wrote the manuscript, with input from the other authors.

## Competing interests

The Authors declare no competing interests.

## Additional information

**Extended data** is available for this paper at https://doi.org/10.1038/s41594-022-00908-2.

**Correspondence and requests for materials** should be addressed to Jiaqiang Wang, Bing Zhou, Keliang Wu or Falong Lu.

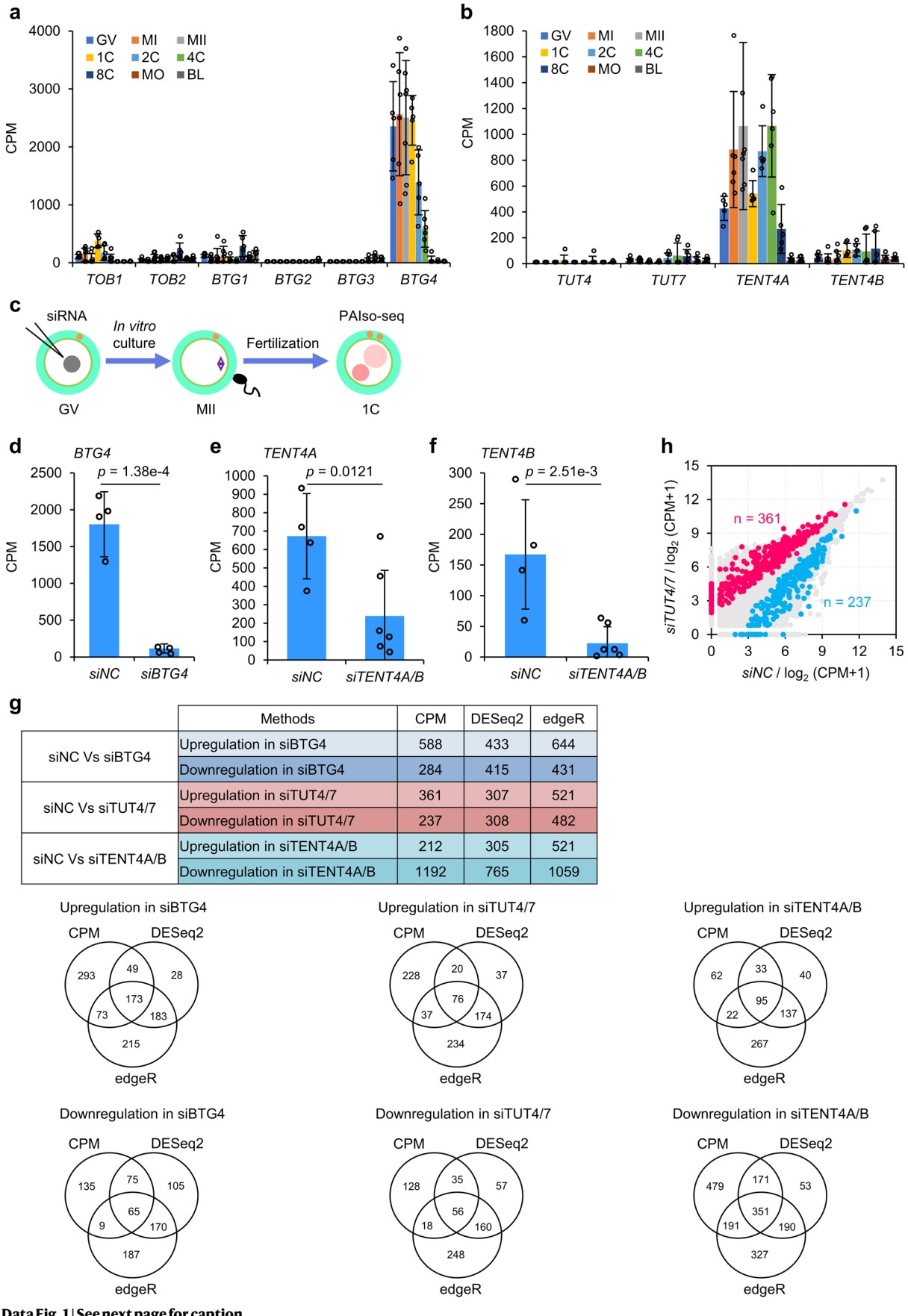

**Extended Data Fig. 1 | See next page for caption.**

**Extended Data Fig. 1 | Knockdown of candidate factors involved in poly(A) tail regulation and non-A residue distribution within poly(A) tails. a, b**, PAIso-seq data showing the expression level of *TOB1*, *TOB2*, *BTG1*, *BTG2*, *BTG3*, and *BTG4* genes (**a**), as well as *TUT4*, *TUT7*, *TENT4A*, and *TENT4B* genes (**b**) in human oocytes and early embryos (Replicates: GV, 5; MI, 6; MII, 7; 1C, 5; 2C, 5; 4C, 6; 8C, 5; MO, 5; BL, 5). **c**, Illustration of siRNA microinjection followed by in vitro maturation and fertilization through intracytoplasmic sperm injection (ICSI) to get knockdown human 1-cell embryos for PAIso-seq analysis. **d**, PAIso-seq data showing the expression level of *BTG4* in *siNC* (n = 4) and *siBTG4* (n = 4) human 1-cell embryos. **e, f**, PAIso-seq data showing the expression level of *TENT4A* (**e**) and *TENT4B* (**f**) in *siNC* (n = 4) and *siTENT4A/B* (n = 6) human 1-cell embryos. **g**, Number of differentially expressed genes (DEGs) in *siBTG4*, *siTUT4/7*, or *siTENT4A/B* human

1-cell embryos as determined by CPM, DESeq2, or edgeR methods (top). The same criteria ($|\log_2(\text{fold change})| \geq 0.5$, $P < 0.05$) are used for calling DEGs for all three methods. The *P* values are calculated by one-tailed Student's *t*-test for the CPM method, while the *P* values are from the built-in output for DESeq2 and edgeR. Venn diagrams showing the overlap of DEGs determined by CPM, DESeq2, or edgeR methods (bottom). **h**, Gene expression level change in *siTUT4/7* (n = 4) compares to *siNC* (n = 4) human 1-cell embryos. The upregulated (361) or downregulated (237) genes ($|\log_2(\text{fold change})| \geq 0.5$, $P < 0.05$) upon *TUT4/7* knockdown are shown in pink or wathet. Data lists for **a-b, d-f, h** are provided in Supplementary Table 4. CPM: counts per million. Error bars indicate the SEM. The *P* values are calculated by one tailed Student's *t*-test if not specified.

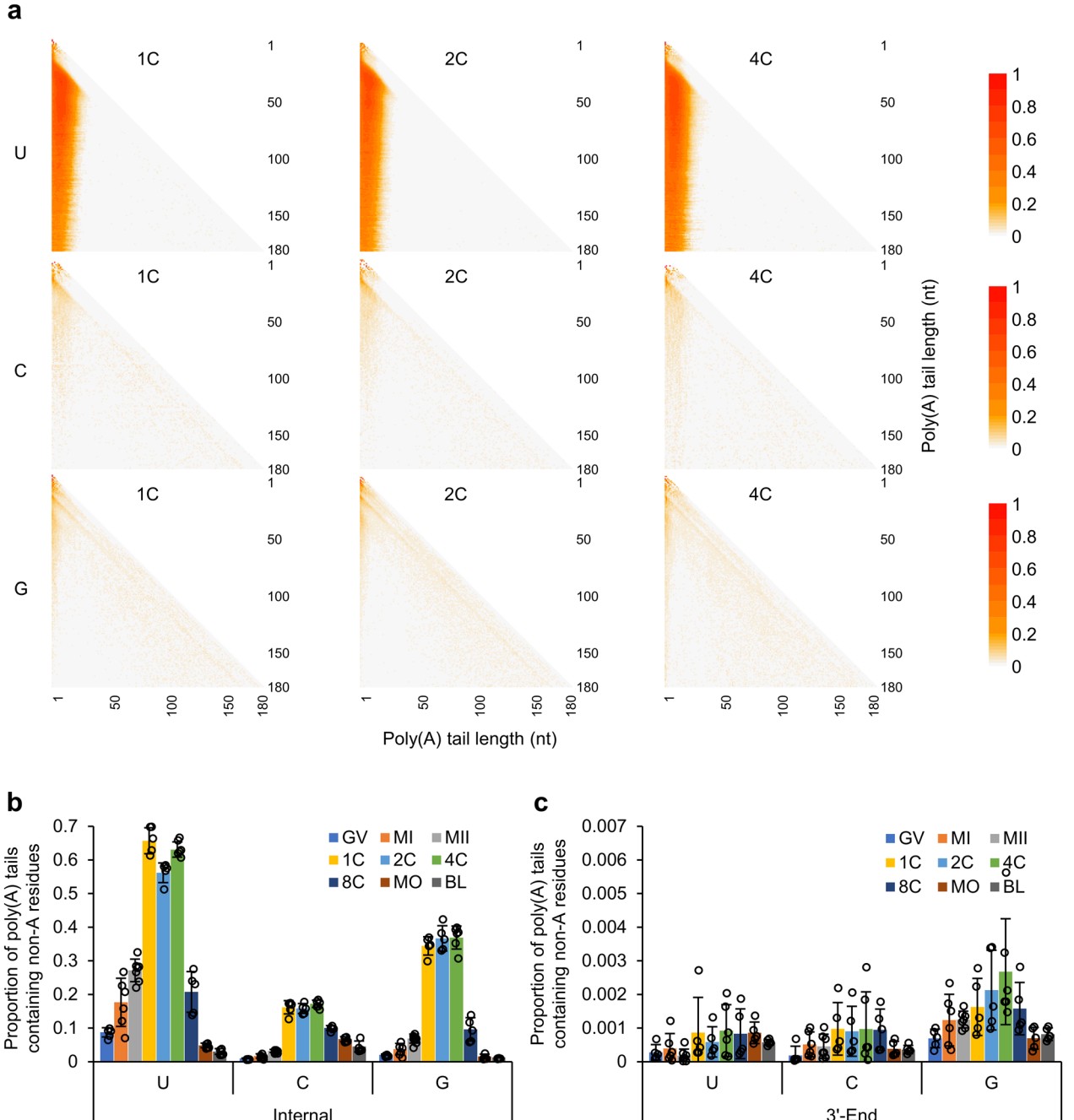

**Extended Data Fig. 2 | Non-A residues within poly(A) tails measured by PAIso-seq. a**, Distribution patterns of U (top), C (middle), or G (bottom) residues within poly(A) tails in human 1-cell (1C), 2-cell (2C), and 4-cell (4C) embryos as measured by PAIso-seq. Poly(A) tails with indicated non-A residues of a given length are collapsed to one line. The relative abundance of non-A residues at each position is calculated and visualized by a color scale. Poly(A) tails with length between 1–180 nt are included and ranked in the heatmap from 1–180 (top to bottom). **b, c**, Frequency of non-A residues (U, C, or G) of transcripts separated by Internal (**b**) and 3'-end positions (**c**) in human oocytes and early embryos measured by

PAIso-seq (GV, 5; MI, 6; MII, 7; 1C, 5; 2C, 5; 4C, 6; 8C, 5; MO, 5; BL, 5). The proportion (y axis) shows the number of poly(A) tails containing the non-A residues with indicated position divided by the total number of poly(A)+ reads for each stage. Reads with pass number at least 10 and poly(A) tail length at least 1 nt are called poly(A)+ reads (see Methods). The 3'-end non-A residues refer to non-A residues at the 3' terminal of poly(A) tails. Non-A residues apart from the 3'-end positions of poly(A) tails are considered as internal non-A residues. The separation of non-A residues based on positions all follow the above rule in this manuscript. Error bars indicate the SEM. Data list is shown in Supplementary Table 8.

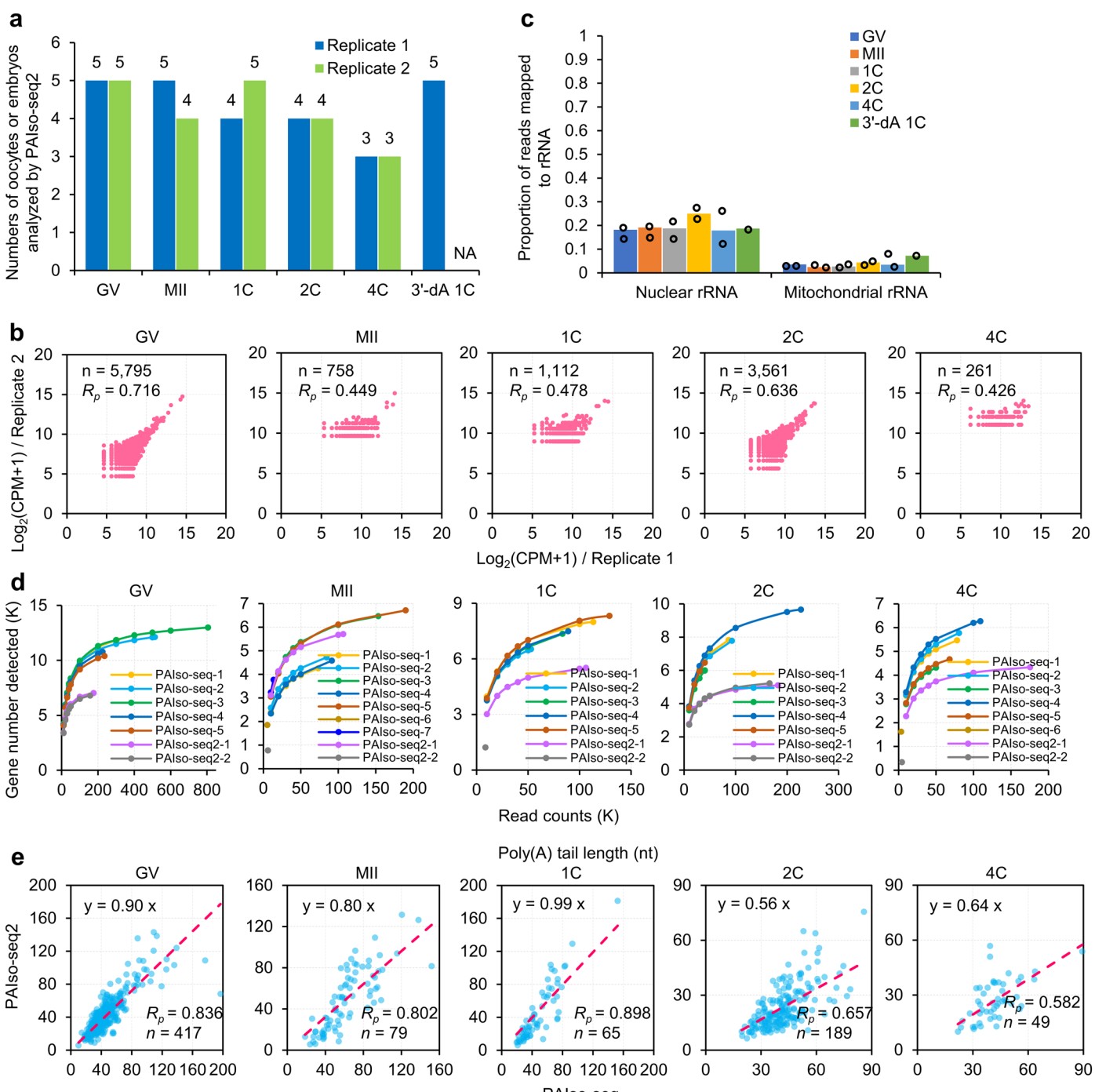

**Extended Data Fig. 3 | Information about all PAIso-seq2 datasets. a**, Numbers of human oocytes and early embryos analyzed by PAIso-seq2 for each of the replicate at different stages. Numbers are shown on top of columns. NA, not applicable. **b**, Gene expression correlation between two replicates for human oocytes and early embryos at different stages measured by PAIso-seq2. Pearson's correlation coefficient (*Rp*) and number of genes are shown on the top. CPM: counts per million. **c**, rRNA levels in PAIso-seq2 libraries (replicates: GV, 2; MII, 2; 1C, 2; 2C, 2; 4C, 2; 3'-dA 1C, 1). The proportion (y axis) shows the number of rRNA reads divided by the total number of reads for each stage. Nuclear rRNA and mitochondrial rRNA are encoded in nuclear and mitochondrial genome respectively. Error bars indicate the SEM. **d**, Saturation curves of PAIso-seq (replicates: GV, 5; MII, 7; 1C, 5; 2C, 5; 4C, 6) and PAIso-seq2 (two replicates each)

data of different stages. The PAIso-seq data use single oocytes or embryos per replicate except for GV stage samples (sample 1, 2, 3, and 4 include two oocytes, while sample 5 includes a single oocyte). The PAIso-seq2 data use 3–5 oocytes or embryos (see **a**) as input. All annotated coding and non-coding genes with ≥ 20 reads are used in this analysis. **e**, Scatterplots showing poly(A) tail length (geometric mean) of individual genes measured by PAIso-seq and PAIso-seq2 for human oocytes and embryos at different stages. Pearson's correlation coefficient (*Rp*) and number of genes are shown at the bottom right of each graph. The dotted line in red represents linear regression line with the linear regression equations on the top. Genes with ≥ 20 poly(A)⁺ reads in both PAIso-seq and PAIso-seq2 datasets of the same stage are included. Data lists for **b** and **e** are provided in Supplementary Table 4 and 16.

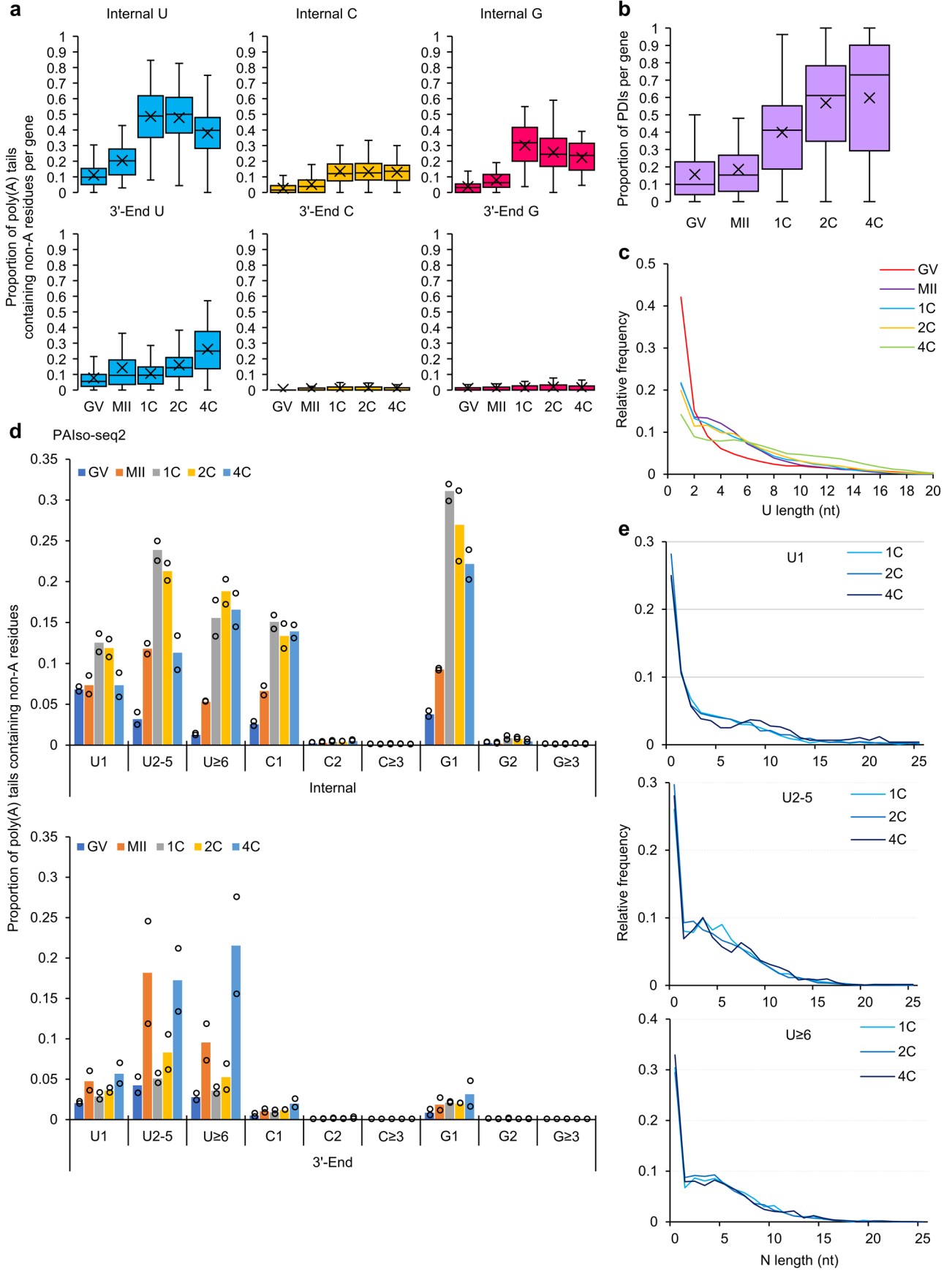

**Extended Data Fig. 4 | See next page for caption.**

**Extended Data Fig. 4 | Validation of PAIso-seq results by PAIso-seq2. a**, Box plots showing frequency of U (left), C (middle), or G (right) residues of individual genes (GV, 417; MII, 80; 1C, 65; 2C, 198; 4C, 49) separated by the positions (Internal, top; 3′-end, bottom) in human oocytes and early embryos. **b**, Box plots of the PDI level of individual genes (GV, 348; MII, 72; 1C, 58; 2C, 149; 4C, 44) in human oocytes and early embryos measured by PAIso-seq2. **c**, Distribution of U length of mRNAs in human oocytes and early embryos measured by PAIso-seq2. The relative frequency (y axis) is the number of the poly(A) tails with the given length of longest consecutive U residues divided by the total number of poly(A) tails with U residues for each sample. **d**, Frequency of U, C, or G residues grouped by the non-A length (1, 2–5, and ≥6 for U; 1, 2, and ≥3 for C and G) separated by the positions (internal, top; 3′-end, bottom) in poly(A) tails in human oocytes and early embryos (2 replicates each) measured by PAIso-seq2. **e**, Distribution of N length of mRNAs separated into U1, U2-5, U≥6 groups in human 1C, 2C, and 4C embryos measured by PAIso-seq2. Relative frequency (y axis) is the number of reads with the given N length divided by the total number of reads for each group (U1, U2-5, U≥6) of each stage. Data lists for **a-e** are provided in Supplementary Table 7–10, 13. For all box plots, the '×' indicates the mean value, the horizontal bars show the median value, and the top and bottom of the box represent the value of 25th and 75th percentile, respectively.

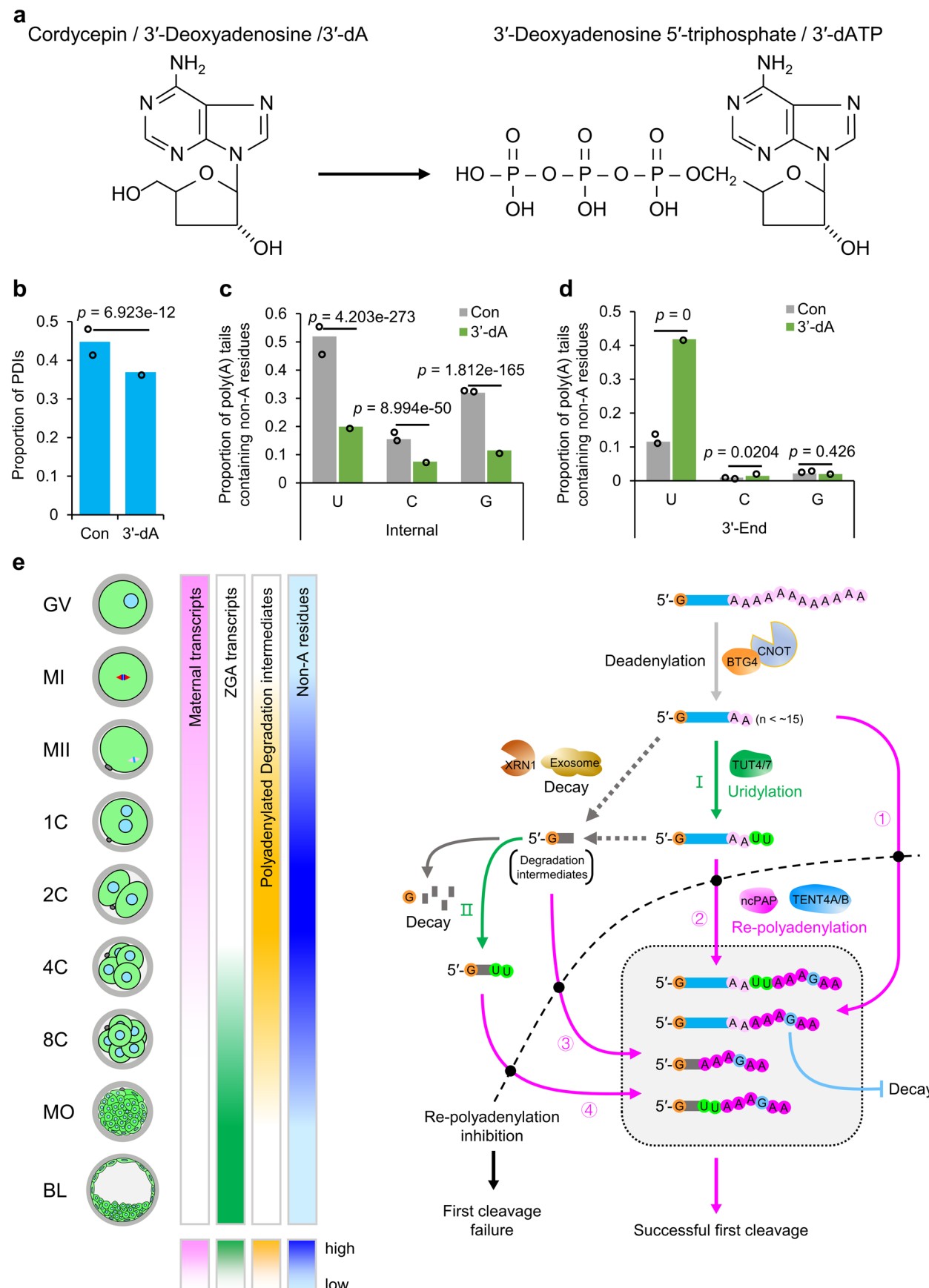

**Extended Data Fig. 5 | See next page for caption.**

**Extended Data Fig. 5 | Human 1-cell embryos treated with 3′-dA. a**, Chemical structural formula of 3′-dA and its conversion to 3′-dATP in cells. **b**, The PDI level of mRNAs in Con (n = 2) and 3′-dA (n = 1) human 1-cell embryos measured by PAIso-seq2. **c, d**, Frequency of non-A residues (U, C, or G) of mRNAs in control (Con, n = 2) and 3′-dA-treated (3′-dA, n = 1) human 1-cell embryos measured by PAIso-seq2 separated by Internal (**c**) and 3′-end (**d**) positions. The proportion (y axis) shows the number of reads with poly(A) tails containing the non-A residues with indicated position divided by the total number of reads with poly(A) tails for each sample. The *P* values are calculated by Chi-squared test. Data lists for **b-d** are provided in Supplementary Table 6 and 8. **e**, Summary of poly(A) tail dynamics during human OET. Left: During the human OET, maternal transcripts are gradually degraded (pink-white gradient), zygotic genome activation (ZGA, green-white gradient) begins in human late 4C embryos, polyadenylated degradation intermediates (PDIs, orange-white gradient) and non-A (U, C, and G) residues (dark blue-light blue gradient) are highly dynamic, both of which peak at 1C to 4C stages. Right: After BTG4-mediated deadenylation, the maternal mRNAs can be decayed by exonuclease (including XRN1 and Exosome). However, uridylation can happen to deadenylated maternal mRNAs with very short poly(A) tail (I) and 3′-UTR partially degraded maternal mRNAs (II). In addition, re-polyadenylation can happen to deadenylated maternal mRNAs with very short poly(A) tail (①), the uridylated mRNAs from I (②), 3′-UTR partially degraded maternal mRNAs (③), and the uridylated mRNAs from II (④). Inhibition of the cytoplasmic polyadenylation (black dotted line) results in the first cleavage failure. CNOT, CCR4-NOT complexes; ncPAP, non-canonical poly(A) polymerase.

# Reporting Summary

## Statistics

For all statistical analyses, confirm that the following items are present in the figure legend, table legend, main text, or Methods section.

| n/a | Confirmed | |
|---|---|---|
| ☐ | ☒ | The exact sample size (*n*) for each experimental group/condition, given as a discrete number and unit of measurement |
| ☒ | ☐ | A statement on whether measurements were taken from distinct samples or whether the same sample was measured repeatedly |
| ☐ | ☒ | The statistical test(s) used AND whether they are one- or two-sided *Only common tests should be described solely by name; describe more complex techniques in the Methods section.* |
| ☒ | ☐ | A description of all covariates tested |
| ☐ | ☒ | A description of any assumptions or corrections, such as tests of normality and adjustment for multiple comparisons |
| ☐ | ☒ | A full description of the statistical parameters including central tendency (e.g. means) or other basic estimates (e.g. regression coefficient) AND variation (e.g. standard deviation) or associated estimates of uncertainty (e.g. confidence intervals) |
| ☐ | ☒ | For null hypothesis testing, the test statistic (e.g. *F*, *t*, *r*) with confidence intervals, effect sizes, degrees of freedom and *P* value noted *Give P values as exact values whenever suitable.* |
| ☒ | ☐ | For Bayesian analysis, information on the choice of priors and Markov chain Monte Carlo settings |
| ☒ | ☐ | For hierarchical and complex designs, identification of the appropriate level for tests and full reporting of outcomes |
| ☐ | ☒ | Estimates of effect sizes (e.g. Cohen's *d*, Pearson's *r*), indicating how they were calculated |

*Our web collection on statistics for biologists contains articles on many of the points above.*

## Software and code

Policy information about availability of computer code

| Data collection | The sequencing data was acquired from Pacific BioSciences (PacBio) sequencing platform sequel II HiFi mode with the default parameters. |
|---|---|
| Data analysis | Ccs (version 5.0.0) was used to process subreads generated by PacBio SMRTbell sequencing.<br>The bam format reads were converted to fastq format using bam2fastq software in pbbam package (https://github.com/PacificBiosciences/pbbam, version 1.0.6).<br>The number of passes for each of the raw CCS read was generated using GetCCSpass.pl (https://github.com/Lulab-IGDB/polyA_analysis/blob/main/bin/).<br>samtools software (http://samtools.sourceforge.net, version 1.9).<br>Minimap2 (version v2.15) was used to align ccs reads to reference genome.<br>Read counts of each gene and gene assignments of each CCS reads were summarized by featureCounts v2.0.0.<br>Custom scripts used for data analysis are available in github: https://github.com/Lulab-IGDB/polyA_analysis. |

For manuscripts utilizing custom algorithms or software that are central to the research but not yet described in published literature, software must be made available to editors and reviewers. We strongly encourage code deposition in a community repository (e.g. GitHub). See the Nature Portfolio guidelines for submitting code & software for further information.

## Data

Policy information about <u>availability of data</u>
All manuscripts must include a <u>data availability statement</u>. This statement should provide the following information, where applicable:

- Accession codes, unique identifiers, or web links for publicly available datasets
- A description of any restrictions on data availability
- For clinical datasets or third party data, please ensure that the statement adheres to our <u>policy</u>

---

Data availability
The CCS data for human oocytes and embryos in bam format from PAIso-seq and PAIso-seq2 are available at Genome Sequence Archive for Human (GSA-Human: https://ngdc.cncb.ac.cn/gsa-human/) hosted by National Genomics Data Center (PAIso-seq: HRA001288, PAIso-seq2: HRA001289). Details for samples in HRA001288 and HRA001289 are shown in Supplementary Table 18. The bam files for visualization of the mapped reads in IGV are available at GSA-human (PAIso-seq: HRA001911, PAIso-seq2: HRA001912). The raw subreads data of FLAM-seq of Hela S3 cells, the human induced pluripotent stem cells (iPSCs), and human cerebral organoids were kindly provided by the original authors.
Genome and gene annotation
The genome sequence used in this study is from the following links: http://ftp.ebi.ac.uk/pub/databases/gencode/Gencode_human/release_36/GRCh38.primary_assembly.genome.fa.gz. The genome annotation used in this study is from the following links: http://ftp.ebi.ac.uk/pub/databases/gencode/Gencode_human/release_36/gencode.v36.primary_assembly.annotation.gtf.gz

---

## Human research participants

Policy information about <u>studies involving human research participants and Sex and Gender in Research.</u>

| | |
|---|---|
| Reporting on sex and gender | The oocytes used this study are from female donors, and the sperms used in this study are from male donors. The sex information for pre-implantation embryos has not been collected, as pre-implantation development is not likely affected by different sex. |
| Population characteristics | The donor women for oocytes are 25–38 years old with tubal-factor infertility and their partners have healthy semen. The donor men for sperms are men of normal semen of no more than 35 years old. |
| Recruitment | Immature oocytes in either the germinal vesicle (GV) or metaphase I (MI) stage were donated by patients taking intracytoplasmic sperm injection (ICSI) treatments, and these immature oocytes were not used in regular clinical practice. The donor women are 25–38 years old with tubal-factor infertility and their partners have healthy semen. In general, immature oocytes obtained in controlled ovarian hyperstimulation cycles were not used for subsequent clinical practice, because the development efficiency of embryos from immature oocytes was low, and patients generally had enough mature oocytes. Therefore, patients with a large number of follicles are communicated clearly about the research purpose in advance before oocyte retrieval to see if they are willing to donate immature oocytes for scientific research with no compensation, and also be ensured that the donated oocytes will be used for research only but not any clinical purposes. Written informed consent is signed by all the donors. When obviously immature GV or MI oocytes are identified by an embryologist during oocyte denuding, another embryologist will confirm the oocyte maturity and then check whether the patient has signed the informed consent for donation. The oocytes meeting the requirements will be collected for subsequent scientific research. The sperms are cryopreserved normal semen donated for research purpose from men of no more than 35 years old with written informed consent signed. |
| Ethics oversight | Institutional Review Board of Reproductive Medicine, Shandong University |

Note that full information on the approval of the study protocol must also be provided in the manuscript.

# Field-specific reporting

Please select the one below that is the best fit for your research. If you are not sure, read the appropriate sections before making your selection.

☒ Life sciences   ☐ Behavioural & social sciences   ☐ Ecological, evolutionary & environmental sciences

For a reference copy of the document with all sections, see <u>nature.com/documents/nr-reporting-summary-flat.pdf</u>

# Life sciences study design

All studies must disclose on these points even when the disclosure is negative.

| | |
|---|---|
| Sample size | No statistical methods were used to pre-determine sample sizes but our sample sizes are similar to those used in previous publications (DOI: 10.1038/nsmb.2660, 10.1038/nature12364). |
| Data exclusions | MI-7 in the PAIso-seq dataset was excluded in the analysis of individual MI replicates due to low number of reads. One of the 3'-dA replicate for PAIso-seq2 dataset was lost during library preparation and was not included in the analysis. |

| Replication | At least four biological replicates were performed for each PAIso-seq experiments. Two biological replicates were performed for each PAIso-seq2 experiments. However, one 3'-dA treated embryo PAIso-seq2 replicate was lost during library preparation. To minimize the use of human embryos and the one replicate of 3'-dA treated sample gives compelling results, we proceed with one replicate for 3'-dA treated human 1-cell embryos. |
|---|---|
| Randomization | The oocytes and embryos were randomly assigned to each experimental groups. |
| Blinding | Data collection and analysis were not performed blind to the conditions of the experiments. We are investigating transcriptome-wide molecular profiling of RNA poly(A) tails which could not be observed before the analysis of the collected data. Therefore, blinding is not necessary. |

# Reporting for specific materials, systems and methods

We require information from authors about some types of materials, experimental systems and methods used in many studies. Here, indicate whether each material, system or method listed is relevant to your study. If you are not sure if a list item applies to your research, read the appropriate section before selecting a response.

## Materials & experimental systems

| n/a | Involved in the study |
|---|---|
| ☒ | ☐ Antibodies |
| ☒ | ☐ Eukaryotic cell lines |
| ☒ | ☐ Palaeontology and archaeology |
| ☒ | ☐ Animals and other organisms |
| ☒ | ☐ Clinical data |
| ☒ | ☐ Dual use research of concern |

## Methods

| n/a | Involved in the study |
|---|---|
| ☒ | ☐ ChIP-seq |
| ☒ | ☐ Flow cytometry |
| ☒ | ☐ MRI-based neuroimaging |

