## [Peer Review File · Nature Structural & Molecular Biology]

Peer Review Information

Manuscript Title: Remodeling of maternal mRNA through poly(A) tail orchestrates human oocyte-to-embryo transition

Corresponding author name(s): Jiaqiang Wang, Bing Zhou, Keliang Wu, Falong Lu

Reviewer Comments & Decisions:

Decision Letter, initial version:

Message: 8th Jul 2022

Dear Dr. Lu,

Thank you again for submitting your manuscript "Remodeling of maternal mRNA through poly(A) tail orchestrates human oocyte-to-embryo transition". We now have comments (please see below) from the 2 reviewers who evaluated your paper. In light of those reports, we remain interested in your study and would like to see your response to the comments of the referees, in the form of a revised manuscript.

You will see that reviewer #1 is positive about the resource value but questions the validity of some of the analyses (e.g. using the the CPM method). Reviewer #2 is supportive of eventual publication after a suitable revision. Their requests are for additional analyses; no new data. There are no major technical concerns from reviewer #2.

Please be sure to address/respond to all concerns of the referees in full in a point-by-point response and highlight all changes in the revised manuscript text file. If you have comments that are intended for editors only, please include those in a separate cover letter.

We are committed to providing a fair and constructive peer-review process. Do not hesitate to contact me if there are specific requests from the reviewers that you believe are technically impossible or unlikely to yield a meaningful outcome.

We expect to see your revised manuscript within 3 months. If you cannot send it within this time, please contact us to discuss an extension; we would still consider your revision, provided that no similar work has been accepted for publication at NSMB or published elsewhere.

As you already know, we put great emphasis on ensuring that the methods and statistics

reported in our papers are correct and accurate. As such, if there are any changes that should be reported, please submit an updated version of the Reporting Summary along with your revision.

Reporting Summary:

Please note that all key data shown in the main figures as cropped gels or blots should be presented in uncropped form, with molecular weight markers. These data can be aggregated into a single supplementary figure item. While these data can be displayed in a relatively informal style, they must refer back to the relevant figures. These data should be submitted with the final revision, as source data, prior to acceptance, but you may want to start putting it together at this point.

Data availability: this journal strongly supports public availability of data. All data used in accepted papers should be available via a public data repository, or alternatively, as Supplementary Information. If data can only be shared on request, please explain why in your Data Availability Statement, and also in the correspondence with your editor. Please note that for some data types, deposition in a public repository is mandatory - more

information on our data deposition policies and available repositories can be found below:
<https://www.nature.com/nature-research/editorial-policies/reporting-standards#availability-of-data>

[Redacted]

Sincerely,

Tiago

Tiago Faial, PhD
Consulting Editor
Nature Structural & Molecular Biology

Reviewers' Comments:

Reviewer #1:

Remarks to the Author:

This study reports interesting features of the human OET, including PDI, re-adenylation and incorporation of non-A bases. To explore the regulation of poly(A) dynamics during OET, the authors performed knockdown of important regulators including Btg4, Tent, and Tut. PAIso-seq and PAIso-seq2 are used to profile poly(A) length, base composition, and changes. Overall, the techniques and the sequencing data provide a valuable resource to the field.

The authors should address the following points:

1. The differential analyses of the PAIso-seq (e.g., Fig. 1g, 7c) use CPM values. Understandably, the read number of each gene is much smaller than regular RNA-seq which complicates differential analyses by DESeq2 or EdgeR. However, is comparison of CPM values of individual genes (Student's t-test without adjusting for false positive results) a reasonable alternative? First, the variation of each gene's CPM across samples is very large (i.e., huge error bars in Fig. 1i). Second, the number of counts of each gene is very small. This weakens the differential analyses and may account for the closeness of the siRNA groups to the siNC in Fig. 1c UMAP. It would be important to systematically compare the differential analysis results from PAIso-seq and regular RNA-seq to justify the CPM method used in this study.
2. In mouse, maternal Btg4 knockout oocytes/embryos have global accumulation of RNA (doi: 10.1038/nsmb.3204). If this also happens in human oocytes/embryos, the global accumulation may not be reflected using CPM.
3. The interpretation of the Fig. 1j seems misleading. Line 227 suggests that the accumulation of the three RNA (PADI6, MOS, ZP2) preferentially selects 20-40 nt poly(A) length. However, if plotting the density instead of the true read counts in Fig. 1j, the siBTG4 and siNC could have similar distribution of poly(A) length. It appears that the RNAs of these genes uniformly accumulate independent of poly(A) length.
4. Quite a few genes have more than two PAS, but the manuscript acknowledges only two (pPAS, dPAS). If there are more than two PAS called in a transcript, is the dPAS the most distal or the second proximal? Based on Fig. 1f, would you expect a positive correlation of poly(A) tail length with the distance of PAS from proximal to distal?
5. What are the empty regions in the poly(A) tail in Fig.2b ZAR1 GV, MI and 1C?
6. Fig. 4a-c are unnecessarily complex. It is generally agreed that most genes are maternal rather than zygotic in 1C embryos as documented by the low percentage of the counts from the 117 genes (~1%). Also, the total number of sequenced genes should be included, rather than only those with read counts >20.
7. A better explanation of Fig. 6 should be provided. The percentage of transcripts having U>10 is very small, probably no more than 5%. Considering the large variation between samples, it may be better to merge all the transcripts of U>10 into one group to increase the sample size and power of analysis.

8. In Fig. 7, the 1,192 down-regulated genes have high internal non-A bases and PDI, which show the same patterns as Fig. 3b, c. The heights of the bars in 7d are generally higher than 3b. Similarly, comparing Fig. 7e and 3c, the mean/median values in 7e are generally larger than 3c, suggesting a higher PDI in the 1,192 genes. Why is there is a decrease of PDI in Fig. 7h? Also, y values in 7g are generally higher than 3f, suggesting a higher non-A bases per gene in the 1,192 genes. Why is there a decrease of non-A bases in Fig. 7i? The explanation of the figure should be clarified.

9. In Fig. 8c, the PAIso-seq2 method identified few transcripts with 3' C and G which is many fewer than 3' U. However, in the tail-seq results of mouse GV oocytes, the terminal C, G modification happens roughly at the same level as terminal U (doi: 10.1038/nature23318.). Is this because of the different organisms or is that possible for PAIso-seq2 to have some internal insufficiency in capturing terminal C and G?

Other points that the authors may wish to consider:

1. Can the authors discuss why the poly(A) length generally has two peaks (Fig. 1d)?
2. The PDI/PIT ratio defined in current study is not quite convincing. It is known that the cleavage and polyadenylation from a PAS are very imprecise (doi: 10.1093/nar/gki158). Therefore, it seems questionable to use 5 nt surrounding a PAS as a precise threshold value to define PDI and PIT. In addition, almost all 'partial degradation' happens at 3'UTR and never in the CDS. The versatile polyadenylation sites are more likely to happen due to the multiple cleavage sites of a PAS rather than 'partial degradation'. On the other hand, it is agreed that the differences of PDI upon different treatment may be evidence of dynamic poly(A) regulation. The authors should clarify these points in the Discussion.

Reviewer #2:

Remarks to the Author:

In this manuscript, Liu, Zhao, Shao et al. investigate the dynamics of mRNA polyadenylation during the oocyte-to-embryo transition (OET) in human embryos. Coupling a very specific library cloning method with long-read sequencing (PacBio), the authors can analyze the length and nucleotide composition of the polyA tail for a number of transcripts during the OET. This universal developmental process is characterized by the clearance of maternal mRNA and the de novo synthesis and processing of zygotic mRNA. The maternal mRNAs that were dormant become competent for translation thanks to the elongation of their polyA tails. These concepts are clear from previous studies in model organisms such as *Drosophila*, *Xenopus*, and zebrafish, but they were largely unexplored in human embryos, precisely the focus of this manuscript. Interestingly, the authors uncover that multiple deadenylation and mRNA degradation events are aborted, resulting in polyadenylation degradation intermediates (PDIs) and that the majority of these transcripts incorporate non-A nucleotides in their tails. The authors not only conduct observational studies across multiple timepoints of the OET but also dissect the mechanisms underlying mRNA degradation and incorporation of non-canonical nucleotides in the polyA tails. The authors demonstrate that BTG4 is involved in the mRNA deadenylation and its loss of function via RNAi leads to a significant reduction of PDIs as a result of less dynamic 3'-end metabolism. The authors followed a similar approach to identify Tut4/Tut7 as the main uridylyl transferase and TENT4 as the enzyme responsible for adding guanosines. Finally, the authors complement their analysis with a new cloning and

sequencing strategy that allows them to identify transcripts with very short or absent polyA tail and show how stalling 3'UTR dynamics is detrimental to proper development via dideoxyadenine injection.

Overall, the authors present a very comprehensive analysis of polyadenylation dynamics during human embryogenesis with an unprecedented level of detail and mechanistic insights. Despite the major post-transcriptional events were already described in previous work in other organisms, the value of this manuscript lies on the relevance for human development and the sequencing approaches that allow nucleotide resolution of the polyadenylation events. One of their major findings is the discovery of the prevalence of PDIs and their doping with non-canonical nucleotides. Albeit these PDIs could be the results of alternative cleavage and polyadenylation signals the authors rationalize effectively based on the analysis of zygotic genes that these events are indeed the product of aborted degradation events.

In summary, the data of this manuscript is accurate, well presented, and supports the conclusions and mechanistic insights of the authors that are relevant to the RNA community. Therefore, this manuscript is suitable for the audience and rigor expected of Nature Structural Molecular Biology if the authors address the following minor comments:

1.- In figure 2, the authors clearly convey the prevalence of non-templated uridylation and 3'UTR degradation in two transcripts, TLE6 and ZAR1. This this GVI visualization it is clear that uridylation occurs close to the end of the 3' UTR. The authors should expand this analysis of non-Adenosine additions to all the detected transcripts and present the data as metaplot, centered at the end of the 3'UTR. This will facilitate to have a global view of the position and length of polyuridylation.

2.- The authors knock down the expression of TENT4A/B and show that it leads to the significant down-regulation of 1,192 transcripts. The authors argue that the lack of stability comes from the loss of internal guanosines in the polyA tail. Since the authors know the exact composition of the polyA tail in the sequenced transcripts, the authors should indicate which proportion of the destabilized transcripts have internal guanosines in the untreated embryos.

3.- Thanks to the combination of PAIso-Seq and PAIso-Seq2, the authors have a good overview of all the possible 3'-ends of the sequenced mRNAs. Using this data, the authors should plot for each detected gene, the percentage of transcripts with detectable degradation and readenylation, or PDI events. The resulting plot can be a box plot or a cumulative plot. This representation will help to visualize the prevalence of PDI across the transcriptome.

Author Rebuttal to Initial comments

Point-to-point Response to referees' comments

We thank the reviewers for their thoughtful and insightful comments on our manuscript. Following the reviewers' suggestions, we have performed additional analyses and thoroughly addressed the reviewers' concerns point-by-point. The manuscript has been revised accordingly. The main changes in the revised

manuscript are highlighted in blue. The following is our point-by-point response to the reviewers' comments.

The responses are colored blue to separate them from the reviewers' comments in black. We have included the updated figures as well as newly generated figures here directly in the response letter. To better cite the figures in the response letter, if the figure is in the revised manuscript, it is named following the name in the revised manuscript. If the figure is composed of panels from different figures, or contains panels for reviewers only, the figures are numbered from **Figure R1** to **Figure R4** with each panel additionally explained in the legends. There is also one table for reviewers only named **Table R1**.

Reviewers' Comments:

Reviewer #1:

Remarks to the Author:

This study reports interesting features of the human OET, including PDI, re-adenylation and incorporation of non-A bases. To explore the regulation of poly(A) dynamics during OET, the authors performed knockdown of important regulators including Btg4, Tent, and Tut. PAIso-seq and PAIso-seq2 are used to profile poly(A) length, base composition, and changes. Overall, the techniques and the sequencing data provide a valuable resource to the field.

The authors should address the following points:

1. The differential analyses of the PAIso-seq (e.g., Fig. 1g, 7c) use CPM values. Understandably, the read number of each gene is much smaller than regular RNA-seq which complicates differential analyses by DESeq2 or EdgeR. However, is comparison of CPM values of individual genes (Student's t-test without adjusting for false positive results) a reasonable alternative? First, the variation of each gene's CPM across samples is very large (i.e., huge error bars in Fig. 1i). Second, the number of counts of each gene is very small. This weakens the differential analyses and may account for the closeness of the siRNA groups to the siNC in Fig. 1c UMAP. It would be important to systematically compare the differential analysis results from PAIso-seq and regular RNA-seq to justify the CPM method used in this study.

Response 1-1:

We thank the reviewer for pointing out this technical issue. We agree with the reviewer that DESeq2 or EdgeR is not specifically designed for low-input full-length 3rd generation transcriptome, the smaller read counts of which may theoretically compromise their performance. We searched the literatures, however, find no established methods that can specifically address this issue. Alternatively, for example, one recent FLEP-seq study presents the difference in transcript quantification in scatter plots without statistic tests (Jia et al., 2020). Therefore, we turned to compare our analysis here with DESeq2 and edgeR which are commonly used tools for RNA-seq analysis of 2nd generation transcriptome as suggested by the reviewer.

The two limitations for the PAIso-seq data as pointed out by the reviewer, variation between samples and small number of counts for each gene, impose difficulty for all three methods: the method used in this study based on student's *t*-test of the CPM data (CPM for short hereafter), DESeq2, and edgeR. There are comparable numbers of differentially expressed genes (DEGs) called for the three methods (**Fig. R1a**). We notice that there are overlapped and method-specific DEGs (**Fig. R1b**). Despite a good proportion of overlapped DEGs between methods, the difference between the results from DESeq2 and edgeR is comparable to that between CPM and DESeq2 or that between CPM and edgeR. These results suggest that these three methods might perform comparably for the analysis of the current datasets. The main limitation might come from the availability of the extremely precious human embryos that leads to the low number of read counts.

Figure R1: Differentially expressed genes by PAIso-seq.

(a). Number of differentially expressed genes (DEGs) in *siBTG4*, *siTUT4/7*, or *siTENT4A/B* human 1-cell embryos as determined by CPM, DESeq2, or edgeR methods. The same criteria ($|\log_2(\text{fold_change})| \geq 0.5$ and $p < 0.05$) are used for calling DEGs for all three methods. The p values are calculated by one-tailed Student's t -test for the CPM method, while the p values are from the built-in output for DESeq2 and edgeR. (b). Venn diagrams showing the overlap of DEGs determined by CPM, DESeq2, or edgeR methods in *siBTG4*, *siTUT4/7*, or *siTENT4A/B* human 1-cell embryos.

The molecular function of BTG4 and TUT4/7 are linked to global mRNA decay in the previous studies (Lim et al., 2014; Liu et al., 2016; Morgan et al., 2017; Yu et al., 2016), which is consistent with our results showing that more genes upregulated upon knockdown of *BTG4* or *TUT4/7*. In addition, the molecular function of TENT4A/B is linked to mRNA stabilization (Lim et al., 2018), which is also consistent with our result showing that more genes are downregulated upon knockdown of *TENT4A/B*. Moreover, our PAIso-seq measures fulllength transcripts, by default, each read count is one transcript. Therefore, the counts per million (CPM) here equals to transcripts per million (TPM), which does not require normalization to account for the length of the transcripts. In this case, CPM is a preferred measurement for the quantification of full-length transcript over FPKM or RPKM which involves normalization over transcript length. Therefore, we'd prefer to keep the current methods in DEG analysis here. Nevertheless, we agree with the reviewer that a better statistical analysis method dealing with low input 3rd generation transcriptome data may help improve the analysis, which is out of the scope of the current study. We wish the reviewer can understand the current situation of the data analysis methods.

2. In mouse, maternal Btg4 knockout oocytes/embryos have global accumulation of RNA (doi: 10.1038/nsmb.3204). If this also happens in human oocytes/embryos, the global accumulation may not be reflected using CPM.

Response 1-2:

We agree with the reviewer that *BTG4* knockdown might cause global accumulation of RNA in human oocytes and embryos. Our analysis here assumes that the global transcripts within a cell are comparable between *siNC* and *siBTG4* which is also the common assumption for regular RNA-seq analysis. Therefore, the global accumulation effect indeed can only be partly reflected using CPM or other RNA-seq analysis methods. Indeed, there are only several hundred genes significantly upregulated in *siBTG4* human embryos for all three data analysis methods as discussed in **Response 1-1**. This is the intrinsic caveat for the current PAIso-seq data. Nevertheless, if the global accumulation of RNA in *BTG4* knockdown human embryos is taken into account, we'd expect many more upregulated genes in *BTG4* knockdown human embryos in the analysis in Fig. 1g and an even larger difference in the analysis in Fig. 1i, and the upregulated genes identified in the current analysis shall have a higher fold of upregulation, which will further strengthen rather than compromise our conclusion that human BTG4 regulates global maternal mRNA deadenylation.

The designed purpose of PAIso-seq data is to uncover the dynamic changes of RNA poly(A) tails rather than differential gene expression. As this caveat does not affect the conclusion that BTG4 regulates maternal mRNA deadenylation, we hope the reviewer can agree that our current analysis based on the CPM serves the main purpose.

3. The interpretation of the Fig. 1j seems misleading. Line 227 suggests that the accumulation of the three RNA (PADI6, MOS, ZP2) preferentially selects 20-40 nt poly(A) length. However, if plotting the density instead of the true read counts in Fig. 1j, the siBTG4 and siNC could have similar distribution of poly(A) length. It appears that the RNAs of these genes uniformly accumulate independent of poly(A) length.

Response 1-3:

We thank the reviewer for pointing out this potentially misleading interpretation. To avoid this potential confusion, we removed the description about “accumulation” in the sentence. We have changed the original words “*We found these three genes also showed increased mRNA level (Fig. 1i and Supplementary Table 3), and accumulated poly(A) tails largely in the range of 20 – 40 nt in BTG4-knockdown human 1-cell embryos (Fig. 1j)*” to “*We found these three genes also showed increased mRNA level with detectable poly(A) tails in the range of 20 – 40 nt in BTG4-knockdown human 1-cell embryos (Fig. 1i, j, and Supplementary Table 3)*”.

4. Quite a few genes have more than two PAS, but the manuscript acknowledges only two (pPAS, dPAS). If there are more than two PAS called in a transcript, is the dPAS the most distal or the second proximal? Based on Fig. 1f, would you expect a positive correlation of poly(A) tail length with the distance of PAS from proximal to distal?

Response 1-4:

We apologize that we did not describe which genes were analyzed in Fig. 1f. In Fig. 1f, we analyzed only genes with two PASs. Therefore, the pPAS and dPAS labeled in the figure mean exactly the proximal and distal PAS.

There are indeed 1,567 genes with more than two PASs in our dataset. However, our analysis requires at least

20 reads for each of the PASs. There are many PASs with less than 20 reads for genes with multiple PASs. Therefore, very few genes with more than 2 PASs pass this criterion. To better convey the above information we have added the following words in the figure legend of the revised manuscript: “*Genes with 2 polyadenylation sites (PASs) are of enough coverage and are used for poly(A) tail length analysis between isoforms.*” In addition, we have changed “*If two PASs were called for one gene*” to “*For genes with two PASs*” in the methods.

The reviewer raised a very interesting possibility that a positive correlation of poly(A) tail length with the distance of PAS from proximal to distal. To test this idea, we only employed genes ($n = 185$) with 3 PASs in GV stage (**Fig. R2**). The number of genes that meet our criteria is too small for the other stages or genes with more than 3 PASs, which can not be used for such statistical analyses. The result shows that indeed there is a nice trend of positive correlation of poly(A) tail length with the distance of PAS from proximal to distal (**Fig. R2**). The statistical results show that the length difference between PAS2 and PAS3, as well as between PAS1 and PAS3, is significant, while the length difference between PAS1 and PAS2 is not significant. Therefore, the trend is as what the reviewer predicted; however, the differences are not all statistically significant, possibly due to the number of genes being too small. In this case, we prefer not to include this result in the manuscript, and we present this piece of data here for reviewer only.

Figure R2: Poly(A) tail length of APA isoforms for genes with three APA sites.

Box plots (left) for geometric means of poly(A) tail length of APA isoforms in human GV oocytes measured by PAIsoseq. PAS1 represents the APA isoform with the polyadenylation site most proximal to the stop codon, PAS3 represents the APA isoform with the polyadenylation site most distal to the stop codon, and PAS2 represents the APA isoform with the polyadenylation site between the proximal and distal ones. Genes ($n = 185$) with ≥ 20 poly(A)⁺ reads for all the three APA isoforms. Reads with pass number ≥ 10 and poly(A) tail length ≥ 1 nt are called poly(A)⁺ reads. The p values are calculated by one-tailed Student's t -test and shown on the right. Note that we combined all replicates of GV samples for analysis here.

5. What are the empty regions in the poly(A) tail in Fig.2b ZAR1 GV, MI and 1C?

Response 1-5:

We apologize that we did not describe it clearly in the figure legend. The regions colored in gray represent the segment of the alternative 3'-UTR sequence in the isoform with long 3'-UTR. To avoid this confusion, we have added "and gray lines between the black and violet dotted line represent alternative 3'-UTR sequences included in the long isoform" in the figure legend of Fig. 2b.

6. Fig. 4a-c are unnecessarily complex. It is generally agreed that most genes are maternal rather than zygotic in 1C embryos as documented by the low percentage of the counts from the 117 genes (~1%). Also, the total number of sequenced genes should be included, rather than only those with read counts >20.

Response 1-6:

We apologize that we did not make it clear why we presented Fig. 4a-c. Yes, we totally agree with the reviewer that most genes are maternal rather than zygotic in 1C embryos. These three panels are presented to make it clear that zygotic transcripts represent a very small amount of the data in the following analysis in Fig. 4d-j. Fig. 4c shows the number of genes with read counts >20, because the subsequent analysis in Fig. 4d-j all use this read count threshold. To avoid potential confusion to the readers, we have added “*To distinguish the contributions from the zygotic transcripts from maternal transcripts, we first examined the amount of potentially zygotically transcribed mRNAs*” in the main text.

7. A better explanation of Fig. 6 should be provided. The percentage of transcripts having U>10 is very small, probably no more than 5%. Considering the large variation between samples, it may be better to merge all the transcripts of U>10 into one group to increase the sample size and power of analysis.

Response 1-7:

We thank the reviewer for this good suggestion. We have merged all the transcripts of U>10 into one group and revised **Fig. 6a** according to the suggestion from the reviewer. Indeed, the figure looks better now. The decrease of the level of U residues upon *TUT4/7* knockdown is mainly seen in the U>10 group which represents about 8% of poly(A) tails with U residues. The expression levels of *TUT4* and *TUT7* are very low in human oocytes and embryos from GV to blastocyst stages, suggesting that a low level of *TUT4* and *TUT7* is already enough to catalyze the U residues we see here. We performed siRNA-mediated knockdown to try to reduce the level of *TUT4* and *TUT7*. As the *TUT4* and *TUT7* levels are already low, the effect of knockdown can not be as clean as knockout. Therefore, the remaining *TUT4* and *TUT7* after siRNA-mediated knockdown are still sufficient to catalyze a good level of U residues, leading to the effect of knockdown only seen in tails with very long U residues. In addition, *TUT4* and *TUT7* protein stored in GV oocytes may account for part of the uridylation activity, which further masked part of the effect of siRNA-mediated knockdown.

Figure 6a: Distribution of U length of mRNAs in control (*siNC*, $n = 4$) and *TUT4/7* knockdown (*siTUT4/7*, $n = 4$) human 1-cell embryos measured by PAIso-seq. Relative frequency (y axis) is the number of reads with poly(A) tails containing the given length of consecutive U residues divided by the total number of reads with poly(A) tails containing U residues for each sample. The U length (x axis) is the number of the longest consecutive U residues within poly(A) tail. The p values from U1 to U10 successively are 0.374; 0.120; 0.214; 0.171; 0.397; 0.410; 0.196; 0.174; 0.229; and 0.129. The p value for $U \geq 11$ is 0.031. Error bars indicate the SEM. The p values are calculated by one-tailed Student's t -test.

8. In Fig. 7, the 1,192 down-regulated genes have high internal non-A bases and PDI, which show the same patterns as Fig. 3b, c. The heights of the bars in 7d are generally higher than 3b. Similarly, comparing Fig. 7e and 3c, the mean/median values in 7e are generally larger than 3c, suggesting a higher PDI in the 1,192 genes. Why is there is a decrease of PDI in Fig. 7h? Also, y values in 7g are generally higher than 3f, suggesting a higher non-A bases per gene in the 1,192 genes. Why is there a decrease of non-A bases in Fig. 7i? The explanation of the figure should be clarified.

Response 1-8:

We apologize that our writing leads to a misunderstanding here. Fig. 7d-g shows the data from transcripts of these 1,192 genes from WT oocytes and embryos. Fig. 3b, c, e, f, and Fig. 7d-g are from the same dataset of WT oocytes and embryos. The difference between Fig. 3b, c, e, f, and Fig. 7d-g is that Fig. 3b, c, e, f uses data from all detected genes, while Fig. 7d-g used data from the selected 1,192 genes. In somatic cells, genes with high G content in poly(A) tails are known to be affected more by Tent4A/B (Lim *et al.*, 2018). Therefore, indeed the 1,192 genes harbor higher PDIs and non-A residues at human 1-cell stage. Figure 7h, i show that the amount of PDIs and non-A residues decrease upon *TENT4A/B* knockdown. This means that these 1,192 genes are preferentially regulated by *TENT4A/B*, knockdown of which leads to decrease of PDIs, non-A residues, and the transcript level.

9. In Fig. 8c, the PAIso-seq2 method identified few transcripts with 3' C and G which is many fewer than 3' U. However, in the tail-seq results of mouse GV oocytes, the terminal C, G modification happens roughly at the same level as terminal U (doi: 10.1038/nature23318.). Is this because of the different organisms or is that possible for PAIso-seq2 to have some internal insufficiency in capturing terminal C and G?

Response 1-9:

First, the observation that the transcripts with 3'-end U residues are many more than 3'-end C and G residues is not likely to be human-specific. We looked into our unpublished PAIso-seq2 data from mice, rats, and pigs, and had similar observations in these species (**Table R1**).

Second, it is also not likely due to the PAIso-seq2 with insufficiency in capturing terminal C and G. PAIso-seq2 used here and TAIL-seq used in the reference the reviewer mentioned employ exactly the same strategy to add pre-adenylated 3'-adaptor by direct ligation using T4 RNA ligase truncated.

We examined the data and the methods from the mouse GV oocyte TAIL-seq reference again (**Table R1**). We realized that it is not because our data has much fewer transcripts with 3'-end C and G residues. Our PAIsoseq2 data actually harbors slightly more transcripts with 3' terminal C or G residues than the TAIL-seq data. Moreover, the TAIL-seq data (~0.48%) has a much lower level of 3' U residues compared to our PAIso-seq2 data (~9.07% in humans, and 8.23% in mice). One very likely possibility is that very short tails (poly(A) tails shorter than 8 nt) were discarded in the TAIL-seq analysis (Chang et al., 2014; Morgan *et al.*, 2017) which presented tails of length in the range between 8 and 79 nt (Morgan *et al.*, 2017), while our PAIso-seq2 data includes tails of length from 1 nt and above without upper limit. The 3'-end U residues are known to readily occur on deadenylated mRNAs (Lim *et al.*, 2014). Therefore, a large amount of very short tails with 3'-end U residues was likely discarded in the TAIL-seq analysis in the reference the reviewer mentioned.

Proportion of poly(A) tails containing 3'-End non-A residues					
GV oocyte	PAIso-seq2				TAIL-seq
	Human	Pig	Rat	Mouse	Mouse
U	9.07%	6.74%	6.48%	8.23%	0.48%
C	0.53%	0.38%	0.64%	0.50%	0.24%
G	0.91%	0.35%	0.51%	0.74%	0.57%

Table R1: Proportion of poly(A) tails containing 3'-End non-A residues in GV oocytes.

Proportion of poly(A) tails containing 3'-End non-A residues in GV oocytes measured by PAIso-seq2 or TAIL-seq (Morgan *et al.*, 2017). The human PAIso-seq2 data is from this study. The pig, rat, and mouse PAIso-seq2 data are our unpublished data.

Other points that the authors may wish to consider:

1. Can the authors discuss why the poly(A) length generally has two peaks (Fig. 1d)?

Response 1-10:

It is an interesting question that we do not have good answer yet. Many genes are of different isoforms. Previous reports show that alternative polyadenylation (APA) isoforms of the same gene can harbor poly(A) tails with different length of peak size (Chang *et al.*, 2014; Liu *et al.*, 2019). One possibility is that the poly(A) tail length for different isoforms contributes to the two peaks. Another possibility might be related to the PABP binding that PABP binds around 20 nt poly(A) sequence to stabilize it, which has been suggested by FLEP-seq2 data (<https://www.biorxiv.org/content/10.1101/2022.01.21.477033v1>). Nevertheless, other potential reasons for this remain an interesting question to explore. Therefore, we have added the following words in the directions to be investigated in the future in the discussion: “*Why the global poly(A) tail length distribution has two peaks in general?*”

2. The PDI/PIT ratio defined in current study is not quite convincing. It is known that the cleavage and polyadenylation from a PAS are very imprecise (doi: 10.1093/nar/gki158). Therefore, it seems questionable to use 5 nt surrounding a PAS as a precise threshold value to define PDI and PIT. In addition, almost all ‘partial degradation’ happens at 3’UTR and never in the CDS. The versatile polyadenylation sites are more likely to happen due to the multiple cleavage sites of a PAS rather than ‘partial degradation’. On the other hand, it is agreed that the differences of PDI upon different treatment may be evidence of dynamic poly(A) regulation.

The authors should clarify these points in the Discussion.

Response 1-11:

Using 5 nt as a threshold is because 5 nt is used in calling the polyadenylated sites (Abdel-Ghany *et al.*, 2016). Indeed, as the reviewer noted, there might be some overestimation of PDIs using this stringent threshold. However, in both cell lines and GV oocytes, we see PDIs around 10%, indicating that the overestimation is very modest which must be less than 10%. We observed more than 60% PDIs in 1C, 2C, and 4C, which is much higher than the potential overestimation, confirming the dynamic generation of the high amount of PDIs in human OET.

The statement that “almost all ‘partial degradation’ happens at 3’UTR and never in the CDS” is not accurate. The polyadenylation sites of PDIs can be found in both 3’-UTR and CDS. To have a global view of the distribution between 3’-UTR and CDS, we have analyzed the proportion of the polyadenylation sites of PDIs in

CDS. The results show that there are around 10% of transcripts with their polyadenylation sites within CDS in 1C, 2C, and 4C embryos which has a high level of PDIs, while only around 1% in oocytes or embryos with a regular level of PDIs (**Fig. R3**).

Figure R3: Proportion of PDIs with polyadenylation sites located in the coding sequence.

Proportion of PDIs with polyadenylation sites located in the coding sequence (CDS). The proportion is calculated by dividing the number of PDIs with polyadenylation sites located in CDS with the number of all the PDIs and PITs. Genes with one unique annotated stop codon are included in the analysis here. In addition, genes with PITs that are polyadenylated at sites located 5' upstream of the stop codon are excluded in the analysis, possibly due to the unannotated isoforms with called PAS upstream of the annotated stop codon.

For the polyadenylation on partially degraded transcripts versus normal PAS cleavage coupled polyadenylation, we prefer polyadenylation on partially degraded transcripts mainly for the following four reasons. First, the above-mentioned high level of transcripts with their polyadenylation sites within CDS support that the PDIs are from partially degraded transcripts but not likely from normal cleavage and polyadenylation. Second, if the new

PASs are generated through canonical cleavage couple polyadenylation, we'd expect that the cleavage sites will cluster near the polyadenylation signal sites, however, we do not see such a pattern, and we see largely even distribution upstream of the original PAS. Third, it is agreed that new transcription is minimal in human 1C, 2C, and 4C embryos, therefore, it is very unlikely that there are highly abundant cleavage and polyadenylation events which are generally thought to be transcription-coupled. Forth, the results from knockdown of *BTG4*, *TUT4/7*, or *TENT4A/B* support dynamic post-transcriptional regulation of poly(A) tails by these factors which is also agreed by the reviewer. We have added further discussion about the above point in the discussion of the revised manuscript which is also included below for your convenience.

"We are confident that these PDIs are generated by polyadenylation on partially degraded transcripts, but not normal PAS cleavage coupled polyadenylation, for the following four reasons. First, high levels of PDIs are with their polyadenylation sites within CDS (around 10% in PAIso-seq data of 1C, 2C, and 4C stages, while only around 1% in GV or BL stages). Second, if the new PASs are generated through canonical cleavage couple polyadenylation, we'd expect that the cleavage sites will cluster near the polyadenylation signal sites, however, we do not see such a pattern, and we see largely even distribution upstream of original PAS. Third, it is agreed that new transcription is minimal in human 1C, 2C, and 4C

embryos, therefore, it is very unlikely that there are highly abundant cleavage and polyadenylation events which are generally thought to be transcription coupled. Forth, the results from knockdown of BTG4, TUT4/7, or TENT4A/B support dynamic post-transcriptional regulation of poly(A) tails by these factors.”

Reviewer #2:

Remarks to the Author:

In this manuscript, Liu, Zhao, Shao et al. investigate the dynamics of mRNA polyadenylation during the oocyte-to-embryo transition (OET) in human embryos. Coupling a very specific library cloning method with long-read sequencing (PacBio), the authors can analyze the length and nucleotide composition of the polyA tail for a number of transcripts during the OET. This universal developmental process is characterized by the clearance of maternal mRNA and the de novo synthesis and processing of zygotic mRNA. The maternal mRNAs that were dormant become competent for translation thanks to the elongation of their polyA tails. These concepts are clear from previous studies in model organisms such as *Drosophila*, *Xenopus*, and zebrafish, but they were largely unexplored in human embryos, precisely the focus of this manuscript. Interestingly, the authors uncover that multiple deadenylation and mRNA degradation events are aborted, resulting in polyadenylation degradation intermediates (PDIs) and that the majority of these transcripts incorporate non-A nucleotides in their tails. The authors not only conduct observational studies across multiple timepoints of the OET but also dissect the mechanisms underlying mRNA degradation and incorporation of non-canonical nucleotides in the polyA tails. The authors demonstrate that BTG4 is involved in the mRNA deadenylation and its loss of function via RNAi leads to a significant reduction of PDIs as a result of less dynamic 3'-end metabolism. The authors followed a similar approach to identify Tut4/Tut7 as the main uridyl transferase and TENT4 as the enzyme responsible for adding guanosines. Finally, the authors complement their analysis with a new cloning and sequencing strategy that allows them to identify transcripts with very short or absent polyA tail and show how stalling 3'UTR dynamics is detrimental to proper development via dideoxyadenine injection.

Overall, the authors present a very comprehensive analysis of polyadenylation dynamics during human embryogenesis with an unprecedented level of detail and mechanistic insights. Despite the major posttranscriptional events were already described in previous work in other organisms, the value of this manuscript lies on the relevance for human development and the sequencing approaches that allow nucleotide resolution of the polyadenylation events. One of their major findings is the discovery of the prevalence of PDIs and their doping with non-canonical nucleotides. Albeit these PDIs could be the results of alternative cleavage and polyadenylation signals the authors rationalize effectively based on the analysis of zygotic genes that these events are indeed the product of aborted degradation events.

In summary, the data of this manuscript is accurate, well presented, and supports the conclusions and mechanistic insights of the authors that are relevant to the RNA community. Therefore, this manuscript is suitable for the audience and rigor expected of Nature Structural Molecular Biology if the authors address the following minor comments:

1.- In figure 2, the authors clearly convey the prevalence of non-templated uridylation and 3'UTR degradation in two transcripts, TLE6 and ZAR1. This this GVI visualization it is clear that uridylation occurs close to the end of the 3' UTR. The authors should expand this analysis of non-Adenosine additions to all the detected transcripts and present the data as metaplot, centered at the end of the 3'UTR. This will facilitate to have a global view of the position and length of polyuridylation. **Response 2-1:**

Yes, the uridylation occurs close to the end of the 3'-UTR. We thank the reviewer for this great suggestion to provide a global view of the distribution of non-A residues. We plotted the global distribution of U, C, and G residues as suggested by the reviewer. We can see that U residues are highly enriched at the start of the poly(A) tails, while C and G residues are distributed relatively uniformly across the poly(A) tails (**Extended Data Fig. 2h**). We have included this new result as **Extended Data Fig. 2h** in the revised manuscript.

Extended Data Fig. 2h: The overall distribution of non-A residues in the poly(A) tails.

Distribution patterns of U (**top**), C (**middle**), or G (**bottom**) residues within poly(A) tails in human 1-cell (1C), 2-cell (2C), and 4-cell (4C) embryos as measured by PAlso-seq. Poly(A) tails with indicated non-A residues of a given length are collapsed to one line. The relative abundance of non-A residues at each position is calculated and visualized by a color scale. Poly(A) tails with length between 1-180 nt are included and ranked in the heatmap from 1-180 (top to bottom).

2.- The authors knock down the expression of TENT4A/B and show that it leads to the significant downregulation of 1,192 transcripts. The authors argue that the lack of stability comes from the loss of internal guanosines in the polyA tail. Since the authors know the exact composition of the polyA tail in the sequenced transcripts, the authors should indicate which proportion of the destabilized transcripts have internal guanosines in the untreated embryos.

Response 2-2:

There are 30.24% transcripts with G residues in the poly(A) tails for the 1,192 destabilized genes in the untreated 1-cell embryos (*siNC*), which is decreased to 25.44% after *TENT4A/B* knockdown. To convey

this information, we have added “*At the transcript level, the decreased of poly(A) tails with G residues for these 1,192 genes after TENT4A/B knockdown (30.24% in siNC and 25.44% in siTENT4A/B) was similar to the trend seen at the gene level (Fig. 7i).*” in the main text of the revised manuscript.

3.- Thanks to the combination of PAIso-Seq and PAIso-Seq2, the authors have a good overview of all the possible 3'-ends of the sequenced mRNAs. Using this data, the authors should plot for each detected gene, the percentage of transcripts with detectable degradation and readenylation, or PDI events. The resulting plot can be a box plot or a cumulative plot. This representation will help to visualize the prevalence of PDI across the transcriptome. **Response 2-3:**

Yes, it is a good idea to present the percentage of transcripts with PDI events. The box plots for the gene level proportion of PDIs are presented in **Fig. 3c** and **Fig. S5b (Fig. R4a, b** here for convenience). In addition, we also performed the cumulative distribution function (CDF) plot as the reviewer suggested (**Fig. R4c, d**). Both the box plots and CDF plots visualize clearly the prevalence and dynamic changes of the level of PDIs across the transcriptome during human OET. We think the box plots might be more friendly to readers. Therefore, we present the box plots in the manuscript.

Figure R4: Proportion of transcripts with PDIs for each detected gene.

Box plots (**top**) or cumulative distribution function (CDF) plots (**bottom**) of the PDI level of individual genes in human oocytes and early embryos measured by PAIso-seq (**left**) or PAIso-seq2 (**right**). The proportion (y axis in the box plots and x axis in the CDF plots) is the number of PDIs divided by the total number of reads (the sum of the number of PDIs and PITs) for each gene of each stage. Genes with ≥ 20 reads (the sum of the number of PDIs and PITs) in each stage are included. Note that we combined all replicates of each stage for analysis here.

Again, we would like to express our highest appreciation to the reviewers for their insightful and constructive comments. We hope now the reviewers are satisfied with the point-to-point answers to their concerns and the revision of the manuscript.

Best wishes,

Falong Lu, Ph.D.

Investigator,

State Key Laboratory of Molecular Developmental Biology,

Institute of Genetics and Developmental Biology, Chinese Academy of Sciences

Referneces

- Abdel-Ghany, S.E., Hamilton, M., Jacobi, J.L., Ngam, P., Devitt, N., Schilkey, F., Ben-Hur, A., and Reddy, A.S. (2016). A survey of the sorghum transcriptome using single-molecule long reads. *Nat Commun* 7, 11706. 10.1038/ncomms11706.
- Chang, H., Lim, J., Ha, M., and Kim, V.N. (2014). TAIL-seq: genome-wide determination of poly(A) tail length and 3' end modifications. *Mol Cell* 53, 1044-1052. 10.1016/j.molcel.2014.02.007.
- Jia, J., Long, Y., Zhang, H., Li, Z., Liu, Z., Zhao, Y., Lu, D., Jin, X., Deng, X., Xia, R., et al. (2020). Posttranscriptional splicing of nascent RNA contributes to widespread intron retention in plants. *Nat Plants* 6, 780788. 10.1038/s41477-020-0688-1.
- Lim, J., Ha, M., Chang, H., Kwon, S.C., Simanshu, D.K., Patel, D.J., and Kim, V.N. (2014). Uridylation by TUT4 and TUT7 marks mRNA for degradation. *Cell* 159, 1365-1376. 10.1016/j.cell.2014.10.055.
- Lim, J., Kim, D., Lee, Y.S., Ha, M., Lee, M., Yeo, J., Chang, H., Song, J., Ahn, K., and Kim, V.N. (2018). Mixed tailing by TENT4A and TENT4B shields mRNA from rapid deadenylation. *Science* 361, 701-704. 10.1126/science.aam5794.
- Liu, Y., Lu, X., Shi, J., Yu, X., Zhang, X., Zhu, K., Yi, Z., Duan, E., and Li, L. (2016). BTG4 is a key regulator for maternal mRNA clearance during mouse early embryogenesis. *J Mol Cell Biol* 8, 366-368. 10.1093/jmcb/mjw023.
- Liu, Y., Nie, H., Liu, H., and Lu, F. (2019). Poly(A) inclusive RNA isoform sequencing (PAIso-seq) reveals wide-spread non-adenosine residues within RNA poly(A) tails. *Nat Commun* 10, 5292. 10.1038/s41467-01913228-9.
- Morgan, M., Much, C., DiGiacomo, M., Azzi, C., Ivanova, I., Vitsios, D.M., Pistolic, J., Collier, P., Moreira, P.N., Benes, V., et al. (2017). mRNA 3' uridylation and poly(A) tail length sculpt the mammalian maternal transcriptome. *Nature* 548, 347-351. 10.1038/nature23318.
- Yu, C., Ji, S.Y., Sha, Q.Q., Dang, Y., Zhou, J.J., Zhang, Y.L., Liu, Y., Wang, Z.W., Hu, B., Sun, Q.Y., et al. (2016). BTG4 is a meiotic cell cycle-coupled maternal-zygotic-transition licensing factor in oocytes. *Nat Struct Mol Biol* 23, 387-394. 10.1038/nsmb.3204.

Decision Letter, first revision:

Message: Our ref: NSMB-A46363A

24th Aug 2022

Dear Dr. Lu,

Thank you for submitting your revised manuscript "Remodeling of maternal mRNA through poly(A) tail orchestrates human oocyte-to-embryo transition" (NSMB-A46363A). It has now been seen by the original referees and their comments are below. The reviewers find that the paper has improved in revision, and therefore we'll be happy in principle to publish it in Nature Structural & Molecular Biology, pending minor revisions to satisfy the referees' final requests and to comply with our editorial and formatting guidelines.

[EDITOR: REMOVE IF WORD DOCUMENT IS AVAILABLE]

To facilitate our work at this stage, we would appreciate if you could send us the main text as a word file. Please make sure to copy the NSMB account (cc'ed above).

Sincerely,

Tiago Faial
Editor
Nature Structural & Molecular Biology

Reviewer #1 (Remarks to the Author):

The manuscript has been nicely revised. I have but one SUGGESTION:

The authors' explanation of using CPM to do the differential analysis is reasonable. It's better to include the comparison (Figure R1) as a supplemental figure in the manuscript, because this could be a reference for people using PacBio sequencing without regular RNA-seq to get the differential genes until a standard statistical method has been established.

Reviewer #2 (Remarks to the Author):

In this revised version of the manuscript from Liu, Zhao and Shao et al., the authors address the comments raised by me and another reviewer. The authors satisfactorily address my comments as follows:

1.- The authors show now the non-A nucleotide additions to the polyA tail for all detected mRNAs at different stages. The new figure, that will be included in the manuscript as extended data, captures the extent and position of uridylation, making it very clear that i) uridylation is prevalent, ii) that most RNAs are modified and iii) that the modification occurs near the end of the 3'UTR.

2.- The authors incorporated a text specifying the percentage of mRNAs with Gs in their polyA tails, before and after knock-down of TENT4A/B.

3.- The authors have incorporated a new figure that shows for each time point the proportion of mRNAs that show polyadenylated degradation intermediates. Now with this box plot is clear that the majority of mRNAs undergo some sort of deadenylation and re-adenylation, during the 1C-4C stages of development.

The additional replies to the other reviewer strengthen the manuscript, too. Overall, the quality of data, rigor of analysis and the novelty of studying the oocyte-to-embryo transition in humans makes the data valuable for the community and the manuscript should be accepted for publication at NSMB.

Decision Letter, author guidance

Message: Our ref: NSMB-A46363A

28th Sep 2022

Dear Dr. Lu,

Thank you for your patience as we've prepared the guidelines for final submission of your Nature Structural & Molecular Biology manuscript, "Remodeling of maternal mRNA through poly(A) tail orchestrates human oocyte-to-embryo transition" (NSMB-A46363A). Please carefully follow the step-by-step instructions provided in the attached file, and add a response in each row of the table to indicate the changes that you have made. Ensuring that each point is addressed will help to ensure that your revised manuscript can be swiftly handed over to our production team.

In recognition of the time and expertise our reviewers provide to Nature Structural & Molecular Biology's editorial process, we would like to formally acknowledge their contribution to the external peer review of your manuscript entitled "Remodeling of

maternal mRNA through poly(A) tail orchestrates human oocyte-to-embryo transition". For those reviewers who give their assent, we will be publishing their names alongside the published article.

Nature Structural & Molecular Biology offers a Transparent Peer Review option for new original research manuscripts submitted after December 1st, 2019. As part of this initiative, we encourage our authors to support increased transparency into the peer review process by agreeing to have the reviewer comments, author rebuttal letters, and editorial decision letters published as a Supplementary item. When you submit your final files please clearly state in your cover letter whether or not you would like to participate in this initiative. Please note that failure to state your preference will result in delays in accepting your manuscript for publication.

Cover suggestions

As you prepare your final files we encourage you to consider whether you have any images or illustrations that may be appropriate for use on the cover of Nature Structural & Molecular Biology.

Nature Structural & Molecular Biology has now transitioned to a unified Rights Collection system which will allow our Author Services team to quickly and easily collect the rights and permissions required to publish your work. Approximately 10 days after your paper is formally accepted, you will receive an email in providing you with a link to complete the grant of rights. If your paper is eligible for Open Access, our Author Services team will also be in touch regarding any additional information that may be required to arrange payment for your article.

Please note that *Nature Structural & Molecular Biology* is a Transformative Journal (TJ). Authors may publish their research with us through the traditional subscription access route or make their paper immediately open access through payment of an article-processing charge (APC). Authors will not be required to make a final decision about access to their article until it has been accepted. [Find out more about Transformative Journals](https://www.springernature.com/gp/open-research/transformative-journals)

Authors may need to take specific actions to achieve [compliance](https://www.springernature.com/gp/open-research/funding/policy-compliance-faqs) with funder and institutional open access mandates. If your research is supported by a funder that requires immediate open access (e.g. according to [Plan S principles](https://www.springernature.com/gp/open-research/plan-s-compliance)) then you should select the gold OA route, and we will direct you to the compliant route where possible. For authors selecting the subscription publication route, the journal's standard licensing terms will need to be accepted, including [self-archiving policies](https://www.nature.com/nature-portfolio/editorial-policies/self-archiving-and-license-to-publish). Those licensing terms will supersede any other terms that the author or any third party may assert apply to any version of the manuscript.

Please use the following link for uploading these materials:
[Redacted]

Best regards,

Sophia Frank
Editorial Assistant
Nature Structural & Molecular Biology
nsmb@us.nature.com

On behalf of

Florian Ullrich, Ph.D.
Associate Editor
Nature Structural & Molecular Biology
ORCID 0000-0002-1153-2040

Reviewer #1:

Remarks to the Author:

The manuscript has been nicely revised. I have but one SUGGESTION:

The authors' explanation of using CPM to do the differential analysis is reasonable. It's better to include the comparison (Figure R1) as a supplemental figure in the manuscript,

because this could be a reference for people using PacBio sequencing without regular RNA-seq to get the differential genes until a standard statistical method has been established.

Reviewer #2:

Remarks to the Author:

In this revised version of the manuscript from Liu, Zhao and Shao et al., the authors address the comments raised by me and another reviewer. The authors satisfactorily address my comments as follows:

1.- The authors show now the non-A nucleotide additions to the polyA tail for all detected mRNAs at different stages. The new figure, that will be included in the manuscript as extended data, captures the extent and position of uridylation, making it very clear that i) uridylation is prevalent, ii) that most RNAs are modified and iii) that the modification occurs near the end of the 3'UTR.

2.- The authors incorporated a text specifying the percentage of mRNAs with Gs in their polyA tails, before and after knock-down of TENT4A/B.

3.- The authors have incorporated a new figure that shows for each time point the proportion of mRNAs that show polyadenylated degradation intermediates. Now with this box plot is clear that the majority of mRNAs undergo some sort of deadenylation and re-adenylation, during the 1C-4C stages of development.

The additional replies to the other reviewer strengthen the manuscript, too. Overall, the quality of data, rigor of analysis and the novelty of studying the oocyte-to-embryo transition in humans makes the data valuable for the community and the manuscript should be accepted for publication at NSMB.

Author Rebuttal, second version

Point-to-point Response to referees' comments

Reviewer #1 (Remarks to the Author):

The manuscript has been nicely revised. I have but one SUGGESTION:

The authors' explanation of using CPM to do the differential analysis is reasonable. It's better to include the comparison (Figure R1) as a supplemental figure in the manuscript, because this could be a reference for people using PacBio sequencing without regular RNA-seq to get the differential genes until a standard statistical method has been established.

Response:

We are glad to hear that this reviewer considers our manuscript been nicely revised.

We thank the reviewer for the one additional great suggestion to include the comparison between different methods (Figure R1) in the manuscript that will be helpful for the reader. We have now included the Figure R1 in the previous version of point-to-point response to referees' comments as Extended Data Fig. 1g in the revised manuscript.

Reviewer #2 (Remarks to the Author):

In this revised version of the manuscript from Liu, Zhao and Shao et al., the authors address the comments raised by me and another reviewer. The authors satisfactorily address my comments as follows:

- 1.- The authors show now the non-A nucleotide additions to the polyA tail for all detected mRNAs at different stages. The new figure, that will be included in the manuscript as extended data, captures the extent and position of uridylation, making it very clear that i) uridylation is prevalent, ii) that most RNAs are modified and iii) that the modification occurs near the end of the 3'UTR.
- 2.- The authors incorporated a text specifying the percentage of mRNAs with Gs in their polyA tails, before and after knock-down of TENT4A/B.
- 3.- The authors have incorporated a new figure that shows for each time point the proportion of mRNAs that show polyadenylated degradation intermediates. Now with this box plot is clear that the majority of mRNAs undergo some sort of deadenylation and re-adenylation, during the 1C-4C stages of development.

The additional replies to the other reviewer strengthen the manuscript, too. Overall, the quality of data, rigor of analysis and the novelty of studying the oocyte-to-embryo transition in humans makes the data valuable for the community and the manuscript should be accepted for publication at NSMB.

Response:

We are glad to hear that this reviewer is satisfied with our responses to the comments. We highly appreciate that this reviewer considers our study novel and valuable for the community.

Again, we would like to express our highest appreciation to the reviewers for their insightful and constructive comments that have helped us to improve the manuscript greatly.

Best wishes,

Falong Lu, Ph.D.

Investigator,

State Key Laboratory of Molecular Developmental Biology,

Institute of Genetics and Developmental Biology, Chinese Academy of Sciences

Final Decision Letter:

Message 6th Dec 2022

:
Dear Falong,

We are now happy to accept your revised paper "Remodeling of maternal mRNA through poly(A) tail orchestrates human oocyte-to-embryo transition" for publication as a Article in Nature Structural & Molecular Biology.

As soon as your article is published, you can generate your shareable link by entering the DOI of your article here: http://authors.springernature.com/share. Corresponding authors will also receive an automated email with the shareable link

Your paper will be published online soon after we receive proof corrections and will appear in print in the next available issue. You can find out your date of online publication by contacting the production team shortly after sending your proof corrections. Content is published online weekly on Mondays and Thursdays, and the embargo is set at 16:00 London time (GMT)/11:00 am US Eastern time (EST) on the day of publication. Now is the time to inform your Public Relations or Press Office about your paper, as they might be interested in promoting its publication. This will allow them time to prepare an accurate and satisfactory press release. Include your manuscript tracking number (NSMB-A46363B) and our journal name, which they will need when they contact our press office.

About one week before your paper is published online, we shall be distributing a press release to news organizations worldwide, which may very well include details of your work. We are happy for your institution or funding agency to prepare its own press release, but it must mention the embargo date and Nature Structural & Molecular Biology. If you or your Press Office have any enquiries in the meantime, please contact press@nature.com.

An online order form for reprints of your paper is available at https://www.nature.com/reprints/author-reprints.html. Please let your coauthors and your institutions' public affairs office know that they are also welcome to order reprints by this method.

Please note that *Nature Structural & Molecular Biology* is a Transformative Journal (TJ). Authors may publish their research with us through the traditional subscription access route or make their paper immediately open access through payment of an article-processing charge (APC). Authors will not be required to make a final decision about access to their article until it has been accepted. [Find out more about Transformative Journals](https://www.springernature.com/gp/open-research/transformative-journals)

Authors may need to take specific actions to achieve [compliance with funder and institutional open access mandates](https://www.springernature.com/gp/open-research/funding/policy-compliance-faqs). If your research is supported by a funder that requires immediate open access (e.g. according to [Plan S principles](https://www.springernature.com/gp/open-research/plan-s-compliance)) then you should select the gold OA route, and we will direct you to the compliant route where possible. For authors selecting the subscription publication route, the journal's standard licensing terms will need to be accepted, including [self-archiving policies](https://www.springernature.com/gp/open-research/policies/journal-policies). Those licensing terms will supersede any other terms that the author or any third party may assert apply to any version of the manuscript.

Kind regards,
Florian

Dr Florian Ullrich
Associate Editor, Nature
Consulting Editor, Nature Structural & Molecular Biology
ORCID 0000-0002-1153-2040
